# Near-Optimal Algorithms for Gaussians with Huber Contamination: Mean Estimation and Linear Regression

**Ilias Diakonikolas**
University of Wisconsin–Madison
ilias@cs.wisc.edu

**Daniel M. Kane**
University of California, San Diego
dakane@cs.ucsd.edu

**Ankit Pensia**
IBM Research
ankit@ibm.com

**Thanasis Pittas**
University of Wisconsin–Madison
pittas@wisc.edu

## Abstract

We study the fundamental problems of Gaussian mean estimation and linear regression with Gaussian covariates in the presence of Huber contamination. Our main contribution is the design of the first sample near-optimal and almost linear-time algorithms with optimal error guarantees for both these problems. Specifically, for Gaussian robust mean estimation on $\mathbb{R}^d$ with contamination parameter $\epsilon \in (0, \epsilon_0)$ for a small absolute constant $\epsilon_0$, we give an algorithm with sample complexity $n = \tilde{O}(d/\epsilon^2)$ and almost linear runtime that approximates the target mean within $\ell_2$-error $O(\epsilon)$. This improves on prior work that achieved this error guarantee with polynomially suboptimal sample and time complexity. For robust linear regression, we give the first algorithm with sample complexity $n = \tilde{O}(d/\epsilon^2)$ and almost linear runtime that approximates the target regressor within $\ell_2$-error $O(\epsilon)$. This is the first polynomial sample and time algorithm achieving the optimal error guarantee, answering an open question in the literature. At the technical level, we develop a methodology that yields almost-linear time algorithms for multi-directional filtering that may be of broader interest.

## 1 Introduction

Modern machine learning systems operate with vast amounts of training data, which are difficult to carefully curate. Consequently, outliers have become a fixture of modern training datasets. This contradicts the standard i.i.d. assumption of classical statistical theory and has spurred the development of robust statistics, which seeks to develop algorithms that perform well in the presence of outliers [HR09; DK23]. The standard contamination model of outliers was first formalized by Huber [Hub64]:

**Definition 1.1** (Huber Contamination Model). *Given $0 < \epsilon < 1/2$ and a distribution family $\mathcal{D}$, the algorithm specifies $n \in \mathbb{N}$ and observes $n$ i.i.d. samples from a distribution $P = (1 - \epsilon)G + \epsilon B$, where $G$ is an unknown distribution in $\mathcal{D}$, and $B$ is an arbitrary distribution. We say $G$ is the distribution of inliers, $B$ is the distribution of outliers, and $P$ is an $\epsilon$-corrupted version of $G$.*

The Huber contamination model has since served as a bedrock for the development and evaluation of robust algorithms. Given that estimating properties of Gaussian distributions is a prototypical task in statistics, Gaussian estimation under Huber contamination is, analogously, a central problem in robust statistics. The *univariate* version of this problem is addressed in Huber's work [Hub64]. In this paper, we focus on estimating *high-dimensional* Gaussian distributions under Huber contamination.

---

Authors are in alphabetical order.

37th Conference on Neural Information Processing Systems (NeurIPS 2023).

Consider the fundamental problem of estimating the mean $\mu$ of a $d$-dimensional isotropic Gaussian $\mathcal{N}(\mu, \mathbf{I})$ given samples from an $\epsilon$-corrupted distribution (Definition 1.1). [DKKLMS18] gave an $n = \text{poly}(d/\epsilon)$ sample and $\text{poly}(n)$-time algorithm for this problem, achieving the information-theoretically optimal error of $\Theta(\epsilon)$. Despite being polynomial in sample and time complexity, these guarantees are far from the linear sample complexity and linear runtime of the sample mean on uncontaminated data. While a number of works [CDG19; DHL19; DKPP22] have developed near-linear time Gaussian robust mean estimation algorithms, all these prior methods *inherently* suffer a sub-optimal error guarantee of $\Omega(\epsilon\sqrt{\log(1/\epsilon)})$.[1] Given the fundamental nature of Gaussian robust mean estimation [DM22], these contrasting sets of results raise the question of whether it is possible to achieve the best of both worlds. In other words:

*Can we obtain $O(\epsilon)$ error for Gaussian mean estimation with Huber contamination in near-linear time and sample complexity?*

While mean estimation is the most basic unsupervised learning task, linear regression is arguably the most basic supervised learning task. Here we study the basic case of Gaussian covariates.

**Definition 1.2** (Gaussian Linear Regression). *Fix $\sigma > 0$ and $\beta \in \mathbb{R}^d$. Let $G$ be the joint distribution of pairs $(X, y)$, with $X \in \mathbb{R}^d$, $y \in \mathbb{R}$, such that $X \sim \mathcal{N}(0, \mathbf{I}_d)$ and $y = \beta^\top X + \eta$, where $\eta \sim \mathcal{N}(0, \sigma^2)$ independently of $X$. The goal of the algorithm is to compute $\widehat{\beta}$ such that $\|\beta - \widehat{\beta}\|_2$ is small.*

The information-theoretic error for robust Gaussian linear regression with Huber contamination is $\Theta(\sigma\epsilon)$. However, all known polynomial time algorithms for this task incur a higher error of $\Omega(\sigma\epsilon\sqrt{\log(1/\epsilon)})$ [DKS19; PJL20; CATJFB20; BP21], raising the following open question in [BP21]:

*Can we obtain $O(\sigma\epsilon)$ error for Gaussian linear regression with Huber contamination in polynomial time and sample complexity?*

Perhaps surprisingly, despite the extensive algorithmic progress on robust statistics over the past years [DKKLMS16; LRV16; DK19; DK23], these basic questions have remained open.

## 1.1 Our Results

**Robust Mean Estimation** We begin by stating our result for Gaussian robust mean estimation:

**Theorem 1.3** (Almost Linear-Time Algorithm for Robust Mean Estimation). *Let $\epsilon_0$ be a sufficiently small positive constant. There is an algorithm that, given parameters $\epsilon \in (0, \epsilon_0)$, $c \in (0, 1)$, $\delta \in (0, 1)$, and $n \gg \frac{1}{\epsilon^2}(d + \log(1/\delta))\,\text{polylog}(d/\epsilon)$ $\epsilon$-corrupted points from $\mathcal{N}(\mu, \mathbf{I}_d)$ (in the Huber contamination model per Definition 1.1), computes an estimate $\widehat{\mu}$ such that $\|\widehat{\mu} - \mu\|_2 = O(\epsilon/c)$ with probability at least $1 - \delta$. Moreover, the algorithm runs in time $(nd + 1/\epsilon^{2+c})\text{polylog}(d/\epsilon\delta)$.*

Taking $c$ to be a small positive constant, our algorithm achieves the optimal asymptotic error (see, e.g., [CGR18]) up to constant factor with near-optimal sample complexity [CGR18] and almost-linear runtime. Moreover, the runtime of our algorithm is near-optimal in the regime when $\epsilon \geq d^{-2/c}$, which is the main regime of interest.[2] Moreover, we note that the algorithm continues to work for a wider family of distributions than Gaussians; See Remark 2.2.

Our techniques are also amenable to the streaming framework of [DKPP22], and Theorem 1.3 can be extended to a streaming algorithm (with polynomial time and sample complexity) that uses only $\tilde{O}(d + \text{poly}(1/\epsilon))$-memory, which is optimal up to the additive $\text{poly}(1/\epsilon)$ term (see Appendix E).

**Robust Linear Regression** We next consider robust Gaussian linear regression (Definition 1.2). As existing polynomial-time algorithms for robust Gaussian linear regression can achieve error of $O(\sigma\epsilon\log(1/\epsilon))$, we assume without loss of generality that the true regressor satisfies $\|\beta\|_2 = O(\sigma\epsilon\log(1/\epsilon))$ (in fact, it can be ensured in nearly-linear time; see, e.g., [CATJFB20, Theorem 2.5]).

Using a novel reduction of linear regression to robust mean estimation for Gaussians, we prove:

---

[1]These prior algorithms continue to work in the total variation/strong contamination model with error $O(\epsilon\sqrt{\log(1/\epsilon)})$. Still, their underlying algorithmic approaches provably cannot obtain better error in the Huber model — the fundamental contamination model and the focus of our work; See Section 1.3 for details.

[2]Using the expression $n = d/\epsilon^2$, the runtime is dominated by $nd$ when $\epsilon > n^{-2/(4+c)}$. Note that the sample complexity requirement directly implies that $\epsilon > 1/\sqrt{n}$.

**Theorem 1.4** (Almost Linear-Time Algorithm for Robust Linear Regression)**.** *Let $\epsilon_0$ be a sufficiently small positive constant. Let $G$ be the joint distribution over $(X, y)$ following the Gaussian linear regression model (Definition 1.2) with unknown parameters $\sigma > 0$ and $\beta \in \mathbb{R}^d$ satisfying $\|\beta\|_2 = O(\sigma\epsilon \log(1/\epsilon))$ There is an algorithm that, given $\epsilon \in (0, \epsilon_0)$, $c \in (0, 1)$ and $n \gg (d/\epsilon^2)\mathrm{polylog}(d/\epsilon)$ i.i.d. labeled examples $(x, y)$ from an $\epsilon$-corrupted version of $G$, the algorithm returns a $\widehat{\beta}$ such that $\|\widehat{\beta} - \beta\|_2 = O(\sigma\epsilon/c)$, with probability at least $0.9$. Moreover, the algorithm runs in time $(nd + 1/\epsilon^{2+c})\mathrm{polylog}(d/\epsilon)$.*

For a constant $c > 0$, Theorem 1.4 has optimal asymptotic error [CGR16], near-optimal sample complexity [CGR16] for constant failure probabilities, and near-optimal runtime (up to an additive factor of $1/\epsilon^c$). Thus, we provide the first polynomial time and sample algorithm (in fact, *near-optimal* time and sample complexity) for robust Gaussian linear regression with Huber contamination.

## 1.2 Our Techniques

Our first major result is a near-optimal sample and almost-linear time algorithm for learning a Gaussian mean to error $O(\epsilon)$ in the Huber contamination model. At a high level, our technique borrows ideas from the $O(\epsilon)$ error algorithm of [DKKLMS18] and the fast robust estimation techniques of [DKPP22]. We emphasize that a number of challenges need to be overcome in order to achieve this result, as elaborated below.

Roughly speaking, the fast algorithm of [DKPP22] works via a careful implementation of the standard filtering algorithm [DKKLMS16; DKKLMS17; DK19]. If the empirical covariance matrix has no direction of large variance, this serves as a certificate that the remaining outliers do not substantially affect the mean. If the empirical covariance matrix has any large eigenvalue, the algorithm projects all the points onto a (randomly chosen) direction of large variance and removes outliers in this direction.[3] Moreover, if there is such a direction, a careful analysis can be used to show that the filtering removes more outliers than inliers and that it improves a carefully chosen potential function — based on the trace of an appropriate power of the (translated) empirical covariance matrix $\Sigma'$, roughly $\mathrm{tr}((\mathbf{\Sigma}' - \mathbf{I})^{\log d})$. Repeating this procedure only $\mathrm{polylog}(d/\epsilon)$ many times eventually yields a sample set with no directions of large variance, whose sample mean is guaranteed to be close to the true mean. Crucially, to achieve this fast runtime, [DKPP22] must consider *random* directions of large variance obtained by applying a suitable power of the (translated) empirical covariance matrix to a random vector. This is to ensure that the outliers cannot be arranged so that they will end up orthogonal to the directions considered.

Conceptually, the $O(\epsilon)$-error algorithm in Huber's model [DKKLMS18] works via a more complicated filter whereby if the covariance matrix has $k$ (for $k = \Omega(\log(1/\epsilon))$ moderately large eigenvalues $1 + \Omega(\epsilon)$, the algorithm projects all points onto the subspace spanned by these $k$ directions and removes those points whose projections are too far from the mean. The usage of multiple orthogonal directions permits better concentration bounds of Gaussians (namely, the Hanson-Wright inequality) and achieves stronger filtering that translates to a smaller final error. In particular, the standard filter cannot reduce the leading eigenvalue to $1 + o(\epsilon \log(1/\epsilon))$, while this improved multi-directional filter can reach the threshold $1 + O(\epsilon)$ for the $k$-th largest eigenvalue. A brute-force approach is then used to learn the projection of the mean onto the subspace of the $k$ largest eigenvalues, while the sample mean is used as an approximation to the mean in the orthogonal directions.[4] However, [DKKLMS18] takes quadratic time, $nd^2$, to run because the subspace used to remove points is deterministic and outliers may be arranged in a way that filtering does not remove the outliers that lie in the orthogonal subspace. A natural idea is to use insights from [DKPP22] to improve the algorithm in [DKKLMS18].

Unfortunately, combining these techniques is highly non-trivial. While [DKPP22] filters based on a single random direction, [DKKLMS18] requires a multi-directional filter with $\Omega(\log(1/\epsilon))$ many (nearly-) orthogonal directions. If the technique of [DKPP22] for producing random directions is called several times in order to produce the directions necessary, it may (if the sample covariance matrix is dominated by only a few large eigenvalues) produce essentially the same vectors over and over. We deal with this by considering two complementary cases. In the first case, where the random

---

[3]We note that [DKPP22] uses a JL-sketch matrix of $\mathrm{polylog}(d/\epsilon)$ size instead of a single direction. However, that is a superficial difference, and a random direction also works; see, for example [DK23; DKPP23].

[4]The brute-force approach takes $2^{\Omega(k)}$ time. Hence, $k$ is chosen to be $\Theta(\log(1/\epsilon))$ to satisfy the filter requirement while ensuring the brute-force approach runs in polynomial time.

directions selected by the technique of [DKPP22] have small inner product, we show that we can simply take $k = \Theta(\log(d/\epsilon))$ many random directions and perform the multi-directional filter of [DKKLMS18] in these directions. When this is not the case, the covariance matrix must have a small number of dominant eigenvectors. We take one of these eigenvectors and put it aside (for now) and instead consider the projections of our original points on the orthogonal subspace. Via a careful analysis, it can be shown that either of these steps will lead to a substantial drop in the potential function of [DKPP22]. This guarantees that after a small number of filter steps, we are left with a matrix with small eigenvalues, allowing us to use the sample mean to approximate the true mean, at least in the directions orthogonal to those put aside.

In the directions that have been put aside, we still need to compute the sample mean. Fortunately, this reduces to the case where the dimension is $\text{polylog}(d/\epsilon)$. Although a brute-force approach will not run in polynomial time for such a large subspace, a careful analysis of [DKKLMS18] can be made to yield an appropriate runtime, because the "effective" dimension is now only $\text{polylog}(d/\epsilon)$.

We now sketch our algorithm for robust linear regression with Gaussian covariates. Our algorithm for robust linear regression in the Huber model achieves near-optimal error of $O(\epsilon\sigma)$ by leveraging our above-described robust mean estimation algorithm. Prior to our work, no polynomial time algorithm was known to achieve this optimal error guarantee. As an additional bonus, our algorithm has near-optimal sample complexity and runs in almost-linear time. The basic idea here is to consider the distribution of $X$ conditioned on the value of $y$ (or more precisely, after rejection sampling based on $y$ in a carefully chosen way). We show that this conditional distribution will be a nearly-spherical Gaussian whose mean is proportional to $\beta$, the true regressor. By using our robust mean estimation algorithm on this conditional distribution, we can obtain our final estimate of $\beta$.

## 1.3  Related Work

Our work lies in the field of algorithmic robust statistics, an active area of research since [DKKLMS16; LRV16]; see [DK23] for a recent book on this topic and Appendix A for additional discussion.

**Contamination Models**  The *strong contamination model* works as follows: a computationally unbounded adversary inspects all the samples and can replace $\epsilon$-fraction of samples with points of their choice [DKKLMS16; DK19]. In contrast, the Huber model is additive and preserves independence among data points. In the rest of this paragraph, we focus on the task of robust (isotropic) Gaussian mean estimation. Despite these differences between the contamination models, the information-theoretic error in both of these models is $\Theta(\epsilon)$. However, the computational landscape of the problem changes considerably. First, [DKKLMS18] was able to achieve this error with polynomial samples and time. However, under the strong contamination model, there is evidence in terms of statistical query lower bounds (and low-degree hardness using [BBHLT20]) that all polynomial time algorithms must incur a larger error of $\Omega(\epsilon\sqrt{\log(1/\epsilon)})$ [DKS17], matching the existing polynomial time algorithms [DKKLMS16], which necessitates that we consider Huber's model.[5]

**Gaussianity and Identity Covariance assumptions**  First, even for Gaussians, knowing the covariance matrix is crucial for computationally-efficient algorithms.[6] [DKKLMS18] that gets $O(\epsilon)$ error runs in polynomial time for isotropic covariances, while the algorithm for unknown covariance matrix runs in quasi-polynomial time [DKKLMS18].[7] Thus, we focus our attention to (nearly) isotropic distributions in this work and improve the runtime of the algorithm from [DKKLMS18] from a large polynomial to almost-linear. Looking beyond Gaussianity assumption, it is impossible (information-theoretically) to obtain $O(\epsilon)$ error for arbitrary isotropic subgaussian distributions, even in a single dimension and Huber contamination model [DK23]. Finally, our results can be extended to a subset of symmetric isotropic subgaussian distributions; see Remark 2.2.

---

[5]The lower bound from [DKS17] continues to hold for total variation corruption model, which is stronger than the Huber model but weaker than the strong corruption model.

[6]For example, all polynomial-time (statistical query and low-degree testing) algorithms for Gaussian mean estimation must use $\tilde{\Omega}(d^2)$ samples to get $o(\sqrt{\epsilon})$ error [DKS19; DKKPP22] as opposed to $O(\epsilon)$ error achievable with $O_\epsilon(d)$ samples using a computationally-inefficient algorithm.

[7][DKKLMS16] gets a larger error $O(\epsilon\sqrt{\log \epsilon^{-1}})$ for unknown covariance in polynomial time and samples.

**Outlier-robust Estimators in Nearly-Linear Time**  Several recent works have developed nearly-linear time algorithms for problems in robust statistics: mean estimation [CDG19; DL22; DHL19; LLVZ20; DKKLT21; DKPP22; CMY20], linear regression [CATJFB20], and PCA [JLT20; DKPP23].

## 2   Preliminaries

**Notation**   We denote $[n]:=\{1,\ldots,n\}$. For $w: \mathbb{R}^d \to [0,1]$ and a distribution $P$, we use $P_w$ to denote the weighted by $w$ version of $P$, i.e., the distribution with pdf $P_w(x) = w(x)P(x)/\mathbf{E}_{X \sim P}[w(X)]$. We use $\mu_P, \mathbf{\Sigma}_P$ for the mean and covariance of $P$. When the vector $\mu$ is clear from the context, we use $\overline{\mathbf{\Sigma}}_P$ to denote the second moment matrix of $P$ centered with respect to $\mu$, i.e., $\overline{\mathbf{\Sigma}}_P := \mathbf{E}_{X \sim P}[(X - \mu)(X - \mu)^\top]$. We use $\|\cdot\|_2$ for $\ell_2$ norm of vectors and $\mathrm{tr}(\cdot)$ and $\|\cdot\|_{\mathrm{op}}$ for trace and operator norm of matrices, respectively. For a subspace $\mathcal{V}$, we use $\mathbf{\Pi}_\mathcal{V}$ to denote the orthogonal projection matrix that projects to $\mathcal{V}$, use $\mathrm{Proj}_\mathcal{V}(x)$ to denote the *orthogonal projection* of $x$ onto $\mathcal{V}$, and use $\mathcal{V}^\perp$ for the subspace orthogonal to $\mathcal{V}$. We use $x \lesssim y$ to denote that $x \leq Cy$ for some absolute constant $C$. We use the notation $a \gg b$ to mean that $a > Cb$ where $C$ is some sufficiently large constant. We use $\mathrm{polylog}()$ to denote a quantity that is poly-logarithmic in its arguments and use $\tilde{O}, \tilde{\Omega}()$, and $\tilde{\Theta}$ to hide such factors.

### 2.1   Goodness Condition

Our algorithm builds on the following (slightly modified) goodness condition from [DKKLMS18]

**Definition 2.1** ( $(\epsilon, \alpha, k)$-Goodness)**.** *We say that a distribution $G$ on $\mathbb{R}^d$ is $(\epsilon, \alpha, k)$-good with respect to $\mu \in \mathbb{R}^d$, if the following conditions are satisfied:*

*(1) (Median) For all $v \in \mathcal{S}^{d-1}$, $\mathbf{Pr}_{X \sim G}[|v^\top(X - \mu)| \gtrsim \epsilon] < 1/2$.*

*(2) For every weight function $w$ with $\mathbf{E}_{X \sim G}[w(X)] \geq 1 - \alpha$, the following hold:*

  *(2.a) (Mean) $\|\mu_{G_w} - \mu\|_2 \lesssim \alpha\sqrt{\log(1/\alpha)}$,*

  *(2.b) (Covariance) $\|\overline{\mathbf{\Sigma}}_{G_w} - \mathbf{I}_d\|_{\mathrm{op}} \lesssim \alpha\log(1/\alpha)$,*

  *(2.c) (Concentration along nearly-orthogonal vectors)  For any $\mathbf{U} \in \mathbb{R}^{k \times d}$ with $\mathrm{tr}(\mathbf{U}^\top\mathbf{U}) = k$ and $\|\mathbf{U}^\top\mathbf{U}\|_{\mathrm{op}} \leq 2k/\log(k/\epsilon)$, the degree-2 polynomial $p(x) := \|\mathbf{U}(x - \mu)\|_2^2$ satisfies $\mathbf{E}_{X \sim G_w}[p(X)\mathbb{1}(p(x) > 100k)] \leq \epsilon/\log(1/\epsilon)$.*

The first condition, (1), states that the median is a good estimate in each direction. The next two conditions, (2.a) and (2.b), state that the mean and covariance of the inliers change by at most $\alpha\sqrt{\log(1/\alpha)}$ and $\alpha\log(1/\alpha)$, respectively, when $\alpha$-fraction of the dataset is deleted; these two conditions with $\alpha = \epsilon$ have been extensively used in the strong contamination model for (sub)-Gaussian distributions but the resulting algorithms necessarily get stuck at error $\epsilon\sqrt{\log(1/\epsilon)}$ error. Condition (2.c), is crucial in obtaining $O(\epsilon)$ error because it provides stronger concentration for projections along $k$-dimensional (nearly-orthogonal) projections using Hanson-Wright inequality. In particular, it implies that at most $O(\frac{\epsilon}{k\log(1/\epsilon)})$-fraction of inliers have $p(x) > 10k$. In contrast, using only Conditions (2.a) and (2.b) would require the threshold to be much larger, $p(x) \gtrsim k\log(k/\epsilon)$, which is insufficient for our application. We show in Appendix B that if $G$ is a set of $n \gg (d + \log(1/\delta))\mathrm{polylog}(d/\epsilon)/\epsilon^2$ i.i.d. samples from $\mathcal{N}(\mu, \mathbf{I})$, then $G$ satisfies the goodness condition mentioned above with probability $1 - \delta$; Prior work [DKKLMS18] had only shown $\mathrm{poly}(d/\epsilon)$ bound for the sample complexity.

**Remark 2.2.**  Our proof for the sample complexity directly extends to all distributions satisfying the following: (i) their linear projections are subgaussian and centrally symmetric with constant density around the median, and (ii) their quadratic projections satisfy Hanson-Wright inequality. In particular, this extension is crucial for deriving our results for robust linear regression.

Throughout the algorithm, we will assume that the inliers and the outliers satisfy the following:

**Setting 2.3.**  Let $\epsilon \in (0, \epsilon_0)$ for a sufficiently small constant $\epsilon_0$, $C > 0$ be a large enough constant, $P$ be a uniform distribution on $n$ points of $\mathbb{R}^d$ that has the mixture form $(1 - \epsilon)G + \epsilon B$ for some $G$ that is $(\epsilon, \alpha, k)$-good (cf. Definition 2.1) with respect to $\mu \in \mathbb{R}^d$, where $\alpha = \epsilon/\log(1/\epsilon)$ and $k = \Theta(\log^2(d/\epsilon))$. Let $w : \mathbb{R}^d \to [0, 1]$ be a weight function with $\mathbf{E}_{X \sim G}[w(X)] \geq 1 - \alpha$ such that $-C\epsilon\mathbf{I} \preceq \mathbf{\Sigma}_{P_w} - \mathbf{I} \preceq C\epsilon\log^2(1/\epsilon)\mathbf{I}$.

The condition on $\mathbf{\Sigma}_{P_w}$ in Setting 2.3 amounts to a "warm start" for our algorithm, which can be obtained in nearly-linear time by running the algorithm from [DKPP22] first (see Appendix B).

The following result gives a sufficient condition for the (weighted) sample mean using (2.a) and (2.b):

**Lemma 2.4** (Certificate Lemma). *In Setting 2.3, if $\|\mathbf{\Sigma}_{P_w}\|_{\mathrm{op}} \leq 1 + \lambda$, then $\|\mu_{P_w} - \mu\|_2 \lesssim \epsilon + \sqrt{\lambda\epsilon}$.*

In light of the certificate lemma, our goal will be to downweigh outliers until the top eigenvalue (in the subspace of interest) is at most $1 + O(\epsilon)$. That would make the bound above $O(\epsilon)$.

We now explain the filtering procedure that we use. A filtering algorithm takes weights $w(x)$ for each point and scores $\tau(x)$ (that capture how much of an outlier the point is) the procedure updates the weights to $w'(x)$ such that: (i) it removes much more mass from outliers than inliers, and (ii) $\epsilon \mathbf{E}_{X \sim B}[w'(x)\tau(x)]$ (the contribution to the weighted average of scores by the outliers) is small after filtering. By considering appropriate scores $\tau(x)$ that use multi-directional projections of the form of Condition (2.c) of Definition 2.1, we show the following in Appendix B.

**Lemma 2.5** (Multi-Directional Filtering). *Consider Setting 2.3. Given a nearly-orthogonal matrix $\mathbf{U} \in \mathbb{R}^{k \times d}$ satisfying $\|\mathbf{U}^\top \mathbf{U}\|_{\mathrm{op}} \leq 2\operatorname{tr}(\mathbf{U}^\top \mathbf{U})/\log(1/\epsilon)$, there is an algorithm that reads $\epsilon$, the $n$ points, their weights $w(x)$, and returns weights $w'$ in time $ndk + \operatorname{polylog}(d/\epsilon, \|\mathbf{U}\|_F^2)$ such that*

*(i)* $\mathbf{E}_{X \sim G}[w(X) - w'(X)] < (\epsilon/\log(1/\epsilon)) \mathbf{E}_{X \sim B}[w(X) - w'(X)]$.

*(ii)* $\epsilon \mathbf{E}_{X \sim B}[w'(X)\|\mathbf{U}(X - \mu_{P_w})\|_2^2] \lesssim \epsilon \operatorname{tr}(\mathbf{U}^\top \mathbf{U})$.

## 2.2 Polynomial Time Algorithm

We now record a (refined) guarantee of the algorithm from [DKKLMS18], which would be useful later on. This algorithm uses the multi-directional filter from Lemma B.27 with $\mathbf{U}$ set to be the top-$k$ eigenvectors of the covariance matrix until the top-$k$ eigenvalues are $1 + O(\epsilon)$ for $k = \Theta(\log(1/\epsilon))$. Lemma 2.4 then guarantees that the empirical mean in the subspace of the $(d - k)$ last eigenvalues is $O(\epsilon)$-close to $\mu$. Finally, the algorithm employs a brute force procedure that uses median and (Condition (1)) to estimate the mean in the remaining $k$-dimensional subspace.

**Lemma 2.6** (Adapted From [DKKLMS18]). *Let $T$ be a set of $n$ i.i.d. samples from an $\epsilon$-corrupted version of $\mathcal{N}(0, \mathbf{I}_d)$ and let $r, \delta \in (0, 1)$ be parameters. If $n \gg (d + \log(1/\delta))/\epsilon^2)\operatorname{polylog}(d/\epsilon)$, then there is an algorithm that, when having as input $T$, $\epsilon$, and $r$, after time $\tilde{O}(nd^2 + \frac{1}{\epsilon^{2+r}})$, it outputs a vector $\widehat{\mu} \in \mathbb{R}^d$ such that $\|\widehat{\mu} - \mu\|_2 \lesssim \epsilon/r$ with probability $1 - \delta$.*

Since the refined runtime above does not appear in [DKKLMS18], we provide a proof in Appendix B. In particular, it is important that the runtime is $nd^2 + \frac{1}{\epsilon^{2+r}}$. Even though this is super-linear in the size of the input, $nd$, our main algorithm in the next section will apply Lemma 2.6 only after projecting the entire dataset to a subspace of a much smaller dimension ($\operatorname{polylog}(d/\epsilon)$ in place of $d$ before).

# 3 Almost-Linear Time Algorithm for Robust Mean Estimation

In this section, we describe the main ideas of the almost-linear time algorithm for Gaussian robust mean estimation that achieves Theorem 1.3.

As described earlier, our algorithm runs in two stages, both of which will run in almost-linear time. In the first stage, our goal is to find a low-dimensional subspace $\mathcal{V}$ (of dimension at most $\operatorname{polylog}(d/\epsilon)$) and a vector that is $O(\epsilon)$ close to $\mu$ when projected in subspace $\mathcal{V}^\perp$. In the second stage, we deploy the basic $O(\epsilon)$ algorithm from [DKKLMS18] on the input data after projecting it onto $\mathcal{V}$ and estimate $\mu$ in that subspace. By a refined analysis of their argument (Lemma 2.6), the second stage runs fast because $\dim(V) = O(\operatorname{polylog}(d/\epsilon))$. By combining these two estimates we get an approximation of $\mu$ in the whole $\mathbb{R}^d$. Since the second stage algorithm's analysis closely follows [DKKLMS18], in the reminder we focus on the first stage, below: (the probability of success can be boosted by standard arguments)

**Theorem 3.1** (First Stage in Almost-Linear Time). *Given a set of $n$ samples, the uniform distribution on which has the form $(1 - \epsilon)G + \epsilon B$ for some $G$ satisfying Conditions (2.a) to (2.c) of Definition 2.1 with respect to $\mu \in \mathbb{R}^d$ and parameters $\alpha = \epsilon/\log(1/\epsilon)$ and $k = \Theta(\log^2(n + d))$. Algorithm 1 takes as input the $n$ points and $\epsilon$, and with probability 0.99, returns a vector $\widehat{\mu}$ and a subspace $\mathcal{V}$ such that $\|\operatorname{Proj}_{\mathcal{V}^\perp}(\widehat{\mu} - \mu)\|_2 \lesssim \epsilon$ and $\dim(\mathcal{V}) = \operatorname{polylog}(d/\epsilon)$. The algorithm runs in time $O(nd\operatorname{polylog}(d/\epsilon))$.*

Theorem 3.1 is realized by Algorithm 1. Without loss of generality, we assume Setting 2.3 holds at the beginning of the algorithm. Recall the high-level strategy that was outlined in Section 1.2: We want to iteratively downweigh points and add directions to $\mathcal{V}$ so that (i) the weight removed

from inliers is at most $O(\epsilon/\log(1/\epsilon))$, (ii) the downweighted dataset along every direction in $\mathcal{V}^\perp$ has variance at most $1 + O(\epsilon)$, (iii) and $\dim(\mathcal{V}) \leq \mathrm{polylog}(d/\epsilon)$. Having the first two, the certificate lemma (Lemma 2.4) implies that the empirical (weighted) mean is $O(\epsilon)$-close to $\mu$ along $\mathcal{V}^\perp$. The way to achieve (i) and (ii) in [DKKLMS18] was by using the matrix $\mathbf{U}$ to be the top-$k$ eigenvectors of the covariance matrix (also, the algorithm of [DKKLMS18] does not add directions to $\mathcal{V}$ until the very end; see Section 2.2)). As mentioned earlier, this approach runs in quadratic time. Our main technical insight is to (a) randomize the choice of $\mathbf{U}$ when safe to do so, and (b) when it is not safe, allow the algorithm to remove a direction by adding it to $\mathcal{V}$ at any time (as opposed to waiting until the end). We first describe the notation below for a more detailed overview.

**Notation for Algorithm 1.** For each round $t \in [t_{\max}]$ of our algorithm, we maintain weights $w_t : \mathbb{R}^d \to [0,1]$ over the dataset, capturing the confidence in the points being inliers, i.e., $w(x) = 0$ represents outliers and $w(x) = 1$ represents inliers. We also maintain a (low-dimensional) subspace $\mathcal{V}_t$ with the goal of making the covariance of the projections of the data on $\mathcal{V}_t^\perp$ be small at the end. Let $\mu_t^\perp, \Sigma_t^\perp$ be the sample mean and covariance of the data after projected on $\mathcal{V}_t^\perp$ and weighted by $w_t$, and define $\mathbf{B}_t^\perp \approx \Sigma_t^\perp - \mathbf{\Pi}_{\mathcal{V}_t^\perp}$ (see Lines 5 to 7 and 4 for precise definitions). We use the potential function $\phi_t := \mathrm{tr}((\mathbf{M}_t^\perp)^\top(\mathbf{M}_t^\perp)) = \|\mathbf{M}_t^\perp\|_{\mathrm{F}}^2$ to track the progress of our algorithm, where $\mathbf{M}_t^\perp = (\mathbf{B}_t^\perp)^p$ for $p = \log d$. Observe that the potential function $\phi_t$ ignores the contribution from the directions in $\mathcal{V}_t$.

---

**Algorithm 1** Robust Mean Estimation Under Huber Contamination (Stage 1)

> **Input**: Parameter $\epsilon \in (0, 1/2)$, uniform distribution over $n$ points that can be written as $P = (1-\epsilon)G + \epsilon B$ where $G$ satisfies Definition 2.1 with appropriate parameters.
> **Output**: An approximation of the mean in a subspace $\mathcal{V}^\perp$ and the orthogonal subspace $\mathcal{V}$.

1: Let $C$ be a sufficiently large constant, $k = C\log^2(n+d)$, $t_{\max} = (\log(d/\epsilon))^C$.
2: Initialize $V_1 \leftarrow \emptyset$ and $w_1(x) = 1$ for all $x \in \mathbb{R}^d$.
3: **for** $t = 1, \ldots, t_{\max}$ **do**
4:      Let $\mathcal{V}_t$ be the subspace spanned by the vectors in $V_t$, and $\mathcal{V}_t^\perp$ be the perpendicular subspace.
5:      Let $P_t$ be the distribution $P$ re-weighted by $w_t$, i.e., $P_t(x) = w_t(x)P(x)/\mathbf{E}_{X \sim P}[w(X)]$.
6:      Let $\mu_t^\perp, \Sigma_t^\perp$ be the mean and covariance of $\mathrm{Proj}_{\mathcal{V}_t}(X)$ when $X \sim P_t$.
7:      Define $\mathbf{B}_t^\perp = (\mathbf{E}_{X \sim P}[w(X)])^2\Sigma_t^\perp - (1 - C_1\epsilon)\mathbf{\Pi}_{\mathcal{V}_t^\perp}$, where $\mathbf{\Pi}_{\mathcal{V}_t^\perp}$ is the orthogonal projection matrix for $\mathcal{V}_t^\perp$, and $\mathbf{M}_t^\perp := (\mathbf{B}_t^\perp)^p$ for $p = \log(d)$.
8:      Calculate $\widehat{\lambda}_t$ such that $\widehat{\lambda}_t/\|\mathbf{B}_t^\perp\|_{\mathrm{op}} \in [0.1/10]$ using power iteration.      ▷ cf. Appendix B
9:      **If** $\widehat{\lambda}_t \leq C\epsilon$ **then return** $\mu_t$ and $V_t$.
10:      Let $q_t := \mathbf{Pr}_{z,z' \sim \mathcal{N}(0,\mathbf{I})}[|\langle \mathbf{M}_t^\perp z, \mathbf{M}_t^\perp z'\rangle| > \|\mathbf{M}_t^\perp z\|_2 \|\mathbf{M}_t^\perp z'\|_2/k^2]$.
11:      Calculate an estimate $\widehat{q}_t$ such that $|\widehat{q}_t - q_t| \leq \frac{1}{10(k^2 t_{\max})}$.
12:      **if** $\widehat{q}_t \leq 1/(k^2 t_{\max})$ **then**      ▷ Case 1 (cf. Section 3.1)
13:          **for** $j \in [k]$ **do**
14:              $v_{t,j} \leftarrow \mathbf{M}_t^\perp z_{t,j}$ for $z_{t,j} \sim \mathcal{N}(0, \mathbf{I})$.
15:          $\mathbf{U}_t \leftarrow [v_{t,1}, \ldots, v_{t,k}]^\top$ i.e., the matrix with rows $v_{t,j}$ for $j \in [k]$.
16:          $w_{t+1} \leftarrow \text{MULTI-DIRECTIONALFILTER}(P, w, \epsilon, \mathbf{U}_t)$      ▷ cf. Lemma B.27
17:      **else**      ▷ Case 2 (cf. Section 3.2)
18:          $u_t \leftarrow (\mathbf{B}_t^\perp)^{p'}z/\|(\mathbf{B}_t^\perp)^{p'}z\|_2$ for $p' := C\log^2(dt_{\max}), z \sim \mathcal{N}(0, \mathbf{I})$.      ▷ Power iteration
19:          $V_{t+1} \leftarrow V_t \cup \{u_t\}$.
20: Let $\mu_{\mathcal{V}_t} = \mathbf{E}_{X \sim P_t}[\mathrm{Proj}_{\mathcal{V}_t}(X)]$ be the mean of $P_t$ after projection to $\mathcal{V}_t$.
21: **return** $\mu_{\mathcal{V}_t}$ and $V_t$.

---

We will show that in every iteration, the potential function decreases multiplicatively, i.e., $\phi_{t+1} \leq (1 - \mathrm{polylog}(\epsilon/d))\phi_t$ while removing at most $O(\epsilon/\log(1/\epsilon))$ fraction of inliers throughout the algorithm (so that we do not fall outside of Setting 2.3). Since $\phi_t$ at $t = 0$ is at most $\mathrm{poly}((d/\epsilon)^p)$ and the algorithm necessarily terminates when it reaches below $\epsilon^p$ (because this implies that $\|\mathbf{B}_t^\perp\|_{\mathrm{op}} \leq (\phi_t)^{1/p} = O(\epsilon)$ which would cause Line 9 to activate), then after $t_{\max} = \mathrm{polylog}(d/\epsilon)$ iterations the algorithm yields a $\mu_{P_w}$ such that $\|\mathrm{Proj}_{\mathcal{V}^\perp}(\mu_{P_w} - \mu)\|_2 = O(\epsilon)$ by Lemma 2.4. Since each iteration is implementable in $\tilde{O}(nd)$ time, the whole algorithm terminates in $\tilde{O}(nd)$ time.

In the next two subsections, we explain how the algorithm decides whether to expand the subspace $\mathcal{V}_t$ or to remove outliers. This decision is based on Line 12, which checks if two random $\mathbf{M}_t^\perp z, \mathbf{M}_t^\perp z'$ for $z, z' \sim \mathcal{N}(0, \mathbf{I})$ are nearly-orthogonal with reasonable probability.

## 3.1 Case 1: Many Large Eigenvalues

By construction, "Case 1" corresponds to the case where the rows of $\mathbf{U}_t$ are nearly-orthogonal (with high probability); see Line 12. Thus, $\mathbf{U}_t$ will be nearly-orthogonal, permitting the use of the multi-directional filtering algorithm from Lemma B.27.

The explanation above ensures that the multi-directional filtering procedure with random $v_i$'s is correct. The reason that it is significantly faster than [DKKLMS18] is that the $v_i$'s are now randomized along the top eigenvalues of $\mathbf{B}_t^\perp$ because they are of the form $\mathbf{M}_t z / \|\mathbf{M}_t z\|$; In contrast, [DKKLMS18] sets $v_i$'s deterministically equal to the top-$k$ eigenvalues of $\mathbf{B}_t^\perp$. As outlined in the introduction, the random choice of $v_i$'s prevents the "adversary" to place the outliers in such a way that the outliers in the orthogonal subspace are unaffected during filtering. In particular, filtering with the random $v_i$'s reduces the contribution of outliers, not only along the exact top-$k$ eigenvectors of $\mathbf{B}_t^\perp$, but also along *all directions with variance comparable to the top eigenvector*. We use the technical insights from [DKPP22] to show that the potential function decreases multiplicatively:

**Claim 3.2.** *With high constant probability, for every round $t$ that Line 12 succeeds, $\phi_{t+1} \le 0.99\phi_t$. Moreover, $\mathbf{E}_{X \sim G}[w_t(X)] \ge 1 - \epsilon/\log(1/\epsilon)$ throughout the algorithm's execution.*

*Proof Sketch.* We first sketch how $\mathbf{U}_t$ is valid for the multi-directional filter, Lemma B.27. Since the check of Line 12 succeeds, with high probability, we have that for each $t \in [t_{\max}]$ rounds, the angles of every pair of rows of $\mathbf{U}_t$ formed in line 15 have cosine at most $1/k^2$. Also, Hanson-Wright inequality implies that with high probability, $\|v_{t,j}\|_2^2 \lesssim \mathrm{tr}(\mathbf{U}_t^\top \mathbf{U}_t) \log(k)$. Combining both of these with Gershgorin Discs Theorem and the choice of $k$, we have that $\|\mathbf{U}_t^\top \mathbf{U}_t\|_{\mathrm{op}} \le 2\mathrm{tr}(\mathbf{U}_t^\top \mathbf{U}_t)/\log(1/\epsilon)$, satisfying the requirements in Lemma B.27 and ensuring $\mathbf{E}_{X \sim G}[w_{t+1}(X)] \ge 1 - \epsilon/\log(1/\epsilon)$.

To show that $\phi_{t+1}$ reduces, we first use the following linear-algebraic result to relate $\phi_{t+1}$ with $\mathrm{tr}(\mathbf{M}_t^\perp \mathbf{B}_{t+1}^\perp \mathbf{M}_t^\perp)$ from [DKPP22]: $\phi_{t+1} = \mathrm{tr}((\mathbf{M}_{t+1}^\perp)^2) \le d^{1/2p} \left(\mathrm{tr}(\mathbf{M}_t^\perp \mathbf{B}_{t+1}^\perp \mathbf{M}_t^\perp)\right)^{\frac{2p}{2p+1}}$. By the definition of $\mathbf{B}_t^\perp$, we have $\mathrm{tr}(\mathbf{M}_t^\perp \mathbf{B}_{t+1}^\perp \mathbf{M}_t^\perp) \approx \mathbf{E}_{P_{t+1}}[\|\mathbf{M}_t^\perp(x-\mu)\|_2^2 - \mathrm{tr}((\mathbf{M}_t^\perp)^2)]$. To upper bound this, we will use the guarantees of the multi-dimensional filter along with the goodness conditions and the fact that $\|\mathbf{M}_t^\perp(x-\mu)\|_2^2 \approx \|\mathbf{U}_t(x-\mu)\|_2^2$ (since $\mathbf{U}_t$ is a Johnson-Lindenstrauss sketch of $\mathbf{M}_t^\perp$ with $k \gtrsim \log(n)$; see Appendix B) as follows: we show in the appendix that the contribution from inliers is small by Definition 2.1 and the contribution from outliers, $\epsilon \, \mathbf{E}_{B_{t+1}}[\|\mathbf{U}_t(x-\mu)\|_2^2]$, is controlled by Lemma B.27. Combining the two aforementioned arguments with some algebraic manipulations we can formally show that:

**Lemma 3.3** (Filtering Implication). *After the filtering, $\mathrm{tr}(\mathbf{M}_t^\perp \mathbf{B}_{t+1}^\perp \mathbf{M}_t^\perp) \le 0.1\|\mathbf{B}_t^\perp\|_{\mathrm{op}}\mathrm{tr}((\mathbf{M}_t^\perp)^2)$.* Combining Lemma C.4 with $\phi_{t+1} \le d^{1/2p} \left(\mathrm{tr}(\mathbf{M}_t^\perp \mathbf{B}_{t+1}^\perp \mathbf{M}_t^\perp)\right)^{\frac{2p}{2p+1}}$ from earlier, and noting that $d^{1/2p} = d^{1/2\log d} \le 3$ and $\|\mathbf{B}_t^\perp\|_{\mathrm{op}}^{2p} \le \mathrm{tr}((\mathbf{M}_t^\perp)^2) = \phi_t$, we have $\phi_{t+1} \le 0.99\phi_t$, as desired. □

## 3.2 Case 2: A Few Large Eigenvalues

Consider now the alternate case where $q_t$ from Line 10 is large. Then the approach from the previous case is inapplicable because $\mathbf{M}_t^\perp z$'s will be highly correlated vectors. The following result formally proves that this happens only when the top eigenvalue of $\mathbf{M}_t^\perp$ contributes significantly to its spectrum.

**Claim 3.4.** *If $\mathbf{Pr}_{z,z' \sim \mathcal{N}(0,\mathbf{I})} \left[ \frac{|\langle \mathbf{M}_t^\perp z, \mathbf{M}_t^\perp z' \rangle|}{\|\mathbf{M}_t^\perp z\|_2 \|\mathbf{M}_t^\perp z'\|_2} > \gamma \right] \ge \alpha$, then $\frac{\|\mathbf{M}_t^\perp\|_{\mathrm{op}}^2}{\mathrm{tr}((\mathbf{M}_t^\perp)^2)} \ge \mathrm{poly}(\alpha, \gamma)$.*

*Proof Sketch.* Let $q_t$ be the probability from the statement. Pretend for simplicity that $\|\mathbf{M}_t^\perp z\|_2^2$ is equal to its expectation $\mathrm{tr}((\mathbf{M}_t^\perp)^2)$. Applying Markov's inequality, we see that $\alpha < q_t \le \gamma^{-2} \mathbf{E}[\langle \mathbf{M}_t^\perp z, \mathbf{M}_t^\perp z' \rangle^2]/\mathrm{tr}((\mathbf{M}_t^\perp)^2) = \gamma^{-2}\|(\mathbf{M}_t^\perp)^2\|_F^2/\mathrm{tr}((\mathbf{M}_t^\perp)^2)$. Further using the standard fact that $\|\mathbf{A}\|_F^2 \le \mathrm{tr}(\mathbf{A})\|\mathbf{A}\|_{\mathrm{op}}$ for a psd matrix $\mathbf{A}$ applied to $\mathbf{M}_t^\perp$ completes the proof. □

Since $\alpha, \gamma$ are $\mathrm{polylog}(d/\epsilon)$ in our setting (Line 12), Claim 3.4 reveals that the top eigenvalue of $(\mathbf{M}_t^\perp)^2$ is at least $\phi_t/\mathrm{polylog}(d/\epsilon)$. Thus, our algorithm can take a top eigenvector $u$ of $\mathbf{M}_t^\perp$ and add it

to the subspace of directions to ignore, $\mathcal{V}_t$, i.e., project all data on the subspace perpendicular to $u$ in the future iterations. Formally, we show that the potential decreases if $u$ has large projection along $\mathbf{M}_t^\perp$:

**Claim 3.5.** *Let $V_{t+1} = V_t \cup \{u\}$ for some unit vector $u \in V_t^\perp$, then $\phi_{t+1} \le \phi_t - u^\top(\mathbf{M}_t^\perp)^2 u$.*

*Proof Sketch.* Going from step $t$ to step $(t+1)$, the effect of adding $u$ in the subspace $V_{t+1}$ is that $\mathbf{B}_{t+1}^\perp = \boldsymbol{\Delta}\mathbf{B}_t^\perp\boldsymbol{\Delta}$ for the projection matrix $\boldsymbol{\Delta} = \mathbf{I} - uu^\top$. Then by properties of the trace operator and Lieb-Thirring inequality: $\phi_{t+1} = \operatorname{tr}((\mathbf{B}_{t+1}^\perp)^{2p}) = \operatorname{tr}((\boldsymbol{\Delta}\mathbf{B}_t^\perp\boldsymbol{\Delta})^{2p}) = \operatorname{tr}((\mathbf{B}_t^\perp\boldsymbol{\Delta})^{2p}) \le \operatorname{tr}((\mathbf{B}_t^\perp)^{2p}\boldsymbol{\Delta}^{2p}) = \operatorname{tr}((\mathbf{B}_t^\perp)^{2p}\boldsymbol{\Delta}) = \operatorname{tr}((\mathbf{B}_t^\perp)^{2p}) - u^\top(\mathbf{B}_t^\perp)^{2p}u = \phi_t - u^\top(\mathbf{M}_t^\perp)^2 u$. $\qquad\square$

Claims 3.4 and 3.5 imply that if we use $u$ equal to the top eigenvector of $\mathbf{M}_t^\perp$, then our proof will be completed. However, the algorithm cannot compute exact eigenvectors in almost-linear time. Fortunately, power iteration in Line 20 computes a fine enough approximation such that $u^\top(\mathbf{M}_t^\perp)^2 u$ is a constant fraction of $\|\mathbf{M}_t^\perp\|_{\mathrm{op}}^2$. Thus, we obtain $\phi_{t+1} \le (1 - 1/\operatorname{polylog}(d/\epsilon))\phi_t$ as desired.

## 4 Robust Linear Regression: Optimal Error In Almost-Linear Time

In this section, we focus our attention to linear regression in Huber contamination model and sketch the proof of Theorem 1.4. Recall that each inlier sample $(X, y)$ is distributed according to the Gaussian linear regression model of Definition 1.2.

---

**Algorithm 2** Robust Linear Regression Under Huber Contamination

---

    **Input**: $\epsilon > 0$, multiset $S$ of $\mathbb{R}^{d+1}$ containing pairs of the form $(x, y)$ with $x \in \mathbb{R}^d$, $y \in \mathbb{R}$.
    **Output**: A vector in $\mathbb{R}^d$.
1: Set $\widehat{\sigma}_y^2 \leftarrow \textsc{TrimmedMean}(\{y : (x, y) \in S\})$.          $\triangleright\ \widehat{\sigma}_y^2 = \sigma_y^2(1 \pm O(\epsilon \log(1/\epsilon)))$
2: Draw $a \in \mathbb{R}$ uniformly at random from $[-\widehat{\sigma}_y, \widehat{\sigma}_y]$.
3: Define the interval $I := [a - \ell, a + \ell]$ for $\ell := \widehat{\sigma}_y / \log(1/\epsilon)$.       $\triangleright$ Random choice of I
4: $S' \leftarrow \{x\ :\ (x, y) \in S, y \in I\}$.              $\triangleright$ Simulating $G_I$
5: $\widehat{\beta}_I \leftarrow \textsc{RobustMean}(S', 10\epsilon)$          $\triangleright$ Algorithm from Theorem 1.3
6: **return** $\beta = \widehat{(\widehat{\sigma}_y^2/a)}\widehat{\beta}_I$.                 $\triangleright$ Rescaling

---

Our main technical insight is a novel reduction to robust mean estimation. Existing reductions in the literature (see, e.g., [BDLS17]) rely on the fact that $\mathbf{E}[yX] = \mathbf{E}[XX^\top\beta + Xz] = \beta$. However, it is unclear how to estimate the mean of $yX$ up to error $O(\epsilon)$ under Huber contamination because $yX$ is very far from Gaussian; e.g., $yX$ is subexponential as opposed to (sub)-Gaussian, and it is not even symmetric around $\beta$, implying median might not even work along a direction. Thus, existing approaches using this methodology have error $\Omega(\sigma\epsilon\log(1/\epsilon))$. In contrast, our reduction reduces to robust (almost)-Gaussian mean estimation by using the conditional distribution of $X$ given $y = a$ under Definition 1.2.

**Claim 4.1** (Conditional Distribution). *Let $(X, y) \in \mathbb{R}^{d+1}$ follow Definition 1.2. Given $a \in \mathbb{R}$, denote by $G_a$ the distribution of $X$ given $y = a$. Similarly, given an interval $I \subset \mathbb{R}$, let $G_I$ represent the conditional distribution of $X$ given $y \in I$. Define $\sigma_y^2 := \sigma^2 + \|\beta\|_2^2$. Then, $G_a = \mathcal{N}\left(\frac{a}{\sigma_y^2}\beta, \mathbf{I}_d - \frac{1}{\sigma_y^2}\beta\beta^\top\right)$ and $G_I = \frac{1}{\mathbf{Pr}[y \in I]}\int_I \phi(a'; 0, \sigma_y^2)\mathcal{N}\left(\frac{a'}{\sigma_y^2}\beta, I_d - \frac{1}{\sigma_y^2}\beta\beta^\top\right)\mathrm{d}a'$, where $\phi(z; 0, \nu^2)$ denotes the pdf of the $\mathcal{N}(0, \nu^2)$ at $z \in \mathbb{R}$.*

Since the conditional distribution $G_a$ above is just a Gaussian whose mean is scaled version of $\beta$ (and roughly isotropic covariance), one would ideally like to get ($O(\epsilon)$-corrupted) samples from $G_a$, then use the robust mean estimator from Theorem 1.3, and finally scale the result back appropriately.

Obviously, it is impossible to simulate $G_a$ using $G$ since there is zero probability over the inliers that $y$ is exactly equal to $a$ in our dataset. Instead, we can simulate the conditional distribution given $y \in I$ for some interval $I$ centered around $a$. The length $\ell$ of $I$ needs to be carefully selected. On the one hand, $I$ should be sufficiently narrow to ensure that the mean of $G_I$ closely approximates the mean of $G_a$. On the other hand, it should not be excessively narrow to avoid rejecting a significant portion of our samples. As we show later, an interval of length $\Theta(\sigma/\log(1/\epsilon))$ meets both of these criteria as long as $\|\beta\|_2 \lesssim \sigma\epsilon\log(1/\epsilon)$. Also, a set of i.i.d. samples from $G_I$ for such a small interval satisfy the goodness condition with the correct parameters because $G_I$ will be roughly isotropic.

Finally, it is critical to ensure that the fraction of outliers in the simulated samples from $G_a$ remains $O(\epsilon)$. In fact, this may not be true depending on the choice of the interval $I$. For example, if the adversary knows $I$, then outliers could all happen to have their labels inside $I$, which would cause the contamination rate to blow up. To overcome this, we randomize the selection of $I$: Since the distributions of outliers is independent of $I$, choosing the center of $I$ to be random means that the bad event of the previous sentence is now a small probability event.

## 4.1 Proof Sketch of Theorem 1.4

The algorithm is given in Algorithm 2. We now give the proof sketch of Theorem 1.4: We claim that the following three events hold simultaneously with probability at least 0.9 ($\widehat{\beta}_0, \widehat{\beta}_1$ and $\widehat{\sigma}_y^2$ are defined in Algorithm 2):

$$\text{(i) } |\widehat{\sigma}_y^2 - \sigma_y^2| \lesssim \sigma_y^2 \epsilon \log(1/\epsilon) \qquad \text{(ii) } |a| > 0.0001\widehat{\sigma}_y \qquad \text{(iii) } \|\widehat{\beta}_I - \mu_{G_I}\|_2 \lesssim \epsilon. \tag{1}$$

Given the above, we first show that $\widehat{\beta}$ is $O(\sigma\epsilon)$-close to $\beta$. For simplicity, let us assume momentarily that $\widehat{\sigma}_y^2 = \sigma_y^2$. Then by triangle inequality $\|\widehat{\beta} - \beta\|_2 \leq \frac{\sigma_y^2}{|a|}(\|\widehat{\beta}_I - \mu_{G_I}\|_2 + \|\mu_{G_I} - \frac{a}{\sigma_y^2}\beta\|_2) \lesssim \sigma_y\epsilon + \sigma_y\|\mu_{G_I} - \frac{a}{\sigma_y^2}\beta\|_2$. The second term arises from the fact that we use $G_I$ instead of $G_a$ in the algorithm (since simulating samples from $G_a$ is algorithmically impossible). Using the expression for $G_I$ from Claim 4.1, we can upper bound this term by $\sigma_y\|\mu_{G_I} - \frac{a}{\sigma_y^2}\beta\|_2 \leq \frac{\ell\|\beta\|_2}{\sigma_y}$. Since $\|\beta\|_2 \lesssim \sigma\epsilon\log(1/\epsilon)$ and $\ell := \sigma_y/\log(1/\epsilon)$, this expression is overall $O(\sigma\epsilon)$.

Now, we show why the events in (1) above hold. The first is simply the guarantee of the trimmed mean on a subexponential distribution (cf. Appendix B) and the second holds because $a \sim \mathcal{U}([-\widehat{\sigma}_y, \widehat{\sigma}_y])$.

The third is significantly more involved. We first claim that Line 4 approximately preserves (within log factors) the size of the dataset and maintains the ratio of inliers to outliers (within constants). Recall that we use $G$ for the inlier distribution and $B$ for the distribution of outliers.

**Lemma 4.2.** *Consider the context of Theorem 1.4 and Algorithm 2. First, for every possible choice of $I$ that can be made in Line 3, $\mathbf{Pr}_{(X,y)\sim G}[y \in I] \gtrsim \ell/\sigma_y$. Second, $\mathbf{E}_I\left[\mathbf{Pr}_{(X,y)\sim B}[y \in I|I]\right] \lesssim \ell/\sigma_y$, where the outer expectation is taken with respect to the random choice of the center of $I$.*

*Proof Sketch.* For inliers, $y \sim \mathcal{N}(0, \sigma_y^2)$ and thus the fraction of inliers in $S$ that belong to $S'$ is at least $\Omega(\ell/\sigma_y)$ since $I \subset [-2\sigma_y, 2\sigma_y]$ with length $\ell$. For outliers, we may assume without loss of generality that $y \in [-\sigma_y - \ell, \sigma_y + \ell]$ always (since otherwise $y \notin I$). Since $I$ is independent of $(X, y)$, we may treat $(X, y)$ as fixed and only $I$ as random. Thus, the probability of $y \in I$ is at most the ratio of the length of $I$ to the length of $[-\sigma_y - \ell, \sigma_y + \ell]$, which is at most $O(\ell/\sigma_y)$ as claimed. $\square$

Observe that the set $S' = \{(x, y) : (x, y) \in S, y \in I\}$ in Line 4 consists of i.i.d. points from $(1 - \epsilon_I)G_I + \epsilon_I B_I$, where $G_I$ is the conditional distribution from Claim 4.1, $B_I$ is the conditional distribution of the outliers, and $\epsilon_I := \frac{\epsilon\,\mathrm{Pr}_{(X,y)\sim B}[y\in I]}{(1-\epsilon)\,\mathrm{Pr}_{(X,y)\sim D}[y\in I]+\epsilon\,\mathrm{Pr}_{(X,y)\sim B}[y\in I]} \leq \frac{\epsilon\,\mathrm{Pr}_{(X,y)\sim B}[y\in I]}{(1-\epsilon)\,\mathrm{Pr}_{(X,y)\sim D}[y\in I]}$. By Lemma 4.2, $\mathbf{E}_I[\epsilon_I] = O(\epsilon)$ and thus by Markov's inequality, with high constant probability we will have that $\epsilon_I = O(\epsilon)$. Finally, it remains to show that $S'$ satisfies the goodness conditions as dictated by Theorem 1.3. This requires technical effort and is deferred to Appendix D.2.

## 5 Discussion

In this paper, we provided fast algorithms for mean estimation and linear regression that achieve optimal error under Huber contamination model for isotropic Gaussian inlier distributions. Several open problems and avenues for improvement remain. First, the sample complexity of our linear regression algorithm is multiplicative in $\log(1/\delta)$, where $\delta$ is the failure probability, as opposed to additive in the information-theoretic sample complexity [CGR16]. More broadly, it is an open problem to design algorithms that achieve similar optimal guarantees for other fundamental tasks: robust principal component analysis, sparse mean estimation, and covariance estimation. In particular, the best known algorithm for the covariance estimation (or mean estimation with unknown covariance) achieving the optimal error runs in quasi-polynomial time [DKKLMS18].

## Acknowledgments

Ilias Diakonikolas is supported by NSF Medium Award CCF-2107079, NSF Award CCF-1652862 (CAREER), and a DARPA Learning with Less Labels (LwLL) grant. Daniel Kane is supported by NSF Medium Award CCF-2107547 and NSF Award CCF-1553288 (CAREER). Ankit Pensia was supported by NSF Awards CCF-1652862, and CCF-1841190, and CCF-2011255; The majority of this work was done while Ankit Pensia was at UW–Madison. Thanasis Pittas is supported by NSF Medium Award CCF-2107079 and NSF Award DMS-2023239 (TRIPODS

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

| | |
|---|---|
| [CTBJ22] | Y. Cherapanamjeri, N. Tripuraneni, P. L. Bartlett, and M. I. Jordan. "Optimal Mean Estimation without a Variance". In: *Proc. 35th Annual Conference on Learning Theory (COLT)*. 2022. |
| [Dep20] | J. Depersin. "A Spectral Algorithm for Robust Regression with Subgaussian Rates". In: *CoRR* abs/2007.06072 (2020). |
| [DHL19] | Y. Dong, S. B. Hopkins, and J. Li. "Quantum Entropy Scoring for Fast Robust Mean Estimation and Improved Outlier Detection". In: *Advances in Neural Information Processing Systems 32 (NeurIPS)*. 2019. |
| [DK19] | I. Diakonikolas and D. M. Kane. "Recent Advances in Algorithmic High-Dimensional Robust Statistics". In: *CoRR* abs/1911.05911 (2019). |
| [DK23] | I. Diakonikolas and D. M. Kane. *Algorithmic High-Dimensional Robust Statistics*. https://sites.google.com/view/ars-book/. Cambridge University Press, 2023. |
| [DKKLMS16] | I. Diakonikolas, G. Kamath, D. M. Kane, J. Li, A. Moitra, and A. Stewart. "Robust Estimators in High Dimensions without the Computational Intractability". In: *Proc. 57th IEEE Symposium on Foundations of Computer Science (FOCS)*. 2016, pp. 655–664. |
| [DKKLMS17] | I. Diakonikolas, G. Kamath, D. M. Kane, J. Li, A. Moitra, and A. Stewart. "Being Robust (in High Dimensions) Can Be Practical". In: *Proceedings of the 34th International Conference on Machine Learning, ICML 2017*. 2017, pp. 999–1008. |
| [DKKLMS18] | I. Diakonikolas, G. Kamath, D. M. Kane, J. Li, A. Moitra, and A. Stewart. "Robustly Learning a Gaussian: Getting Optimal Error, Efficiently". In: *Proceedings of the Twenty-Ninth Annual ACM-SIAM Symposium on Discrete Algorithms, SODA 2018*. Full version available at https://arxiv.org/abs/1704.03866. 2018, pp. 2683–2702. |
| [DKKLSS19] | I. Diakonikolas, G. Kamath, D. Kane, J. Li, J. Steinhardt, and A. Stewart. "Sever: A Robust Meta-Algorithm for Stochastic Optimization". In: *Proceedings of the 36th International Conference on Machine Learning, ICML 2019*. 2019, pp. 1596–1606. |
| [DKKLT21] | I. Diakonikolas, D. M. Kane, D. Kongsgaard, J. Li, and K. Tian. "List-decodable mean estimation in nearly-pca time". In: *Advances in Neural Information Processing Systems 34 (NeurIPS)*. Vol. 34. 2021. |
| [DKKLT22] | I. Diakonikolas, D. M. Kane, D. Kongsgaard, J. Li, and K. Tian. "Clustering Mixture Models in Almost-Linear Time via List-Decodable Mean Estimation". In: *Proc. 54th Annual ACM Symposium on Theory of Computing (STOC)*. 2022. |
| [DKKPP22] | I. Diakonikolas, D. M. Kane, S. Karmalkar, A. Pensia, and T. Pittas. "Robust Sparse Mean Estimation via Sum of Squares". In: *Proc. 35th Annual Conference on Learning Theory (COLT)*. 2022. |
| [DKKPS19] | I. Diakonikolas, D. M. Kane, S. Karmalkar, E. Price, and A. Stewart. "Outlier-Robust High-Dimensional Sparse Estimation via Iterative Filtering". In: *Advances in Neural Information Processing Systems 32 (NeurIPS)*. 2019. |
| [DKLP22] | I. Diakonikolas, D. M. Kane, J. C. H. Lee, and A. Pensia. "Outlier-Robust Sparse Mean Estimation for Heavy-Tailed Distributions". In: *Advances in Neural Information Processing Systems 35 (NeurIPS)*. 2022. |
| [DKP20] | I. Diakonikolas, D. M. Kane, and A. Pensia. "Outlier Robust Mean Estimation with Subgaussian Rates via Stability". In: *Advances in Neural Information Processing Systems 33 (NeurIPS)*. 2020. |
| [DKPP22] | I. Diakonikolas, D. M. Kane, A. Pensia, and T. Pittas. "Streaming algorithms for high-dimensional robust statistics". In: *International Conference on Machine Learning*. PMLR. 2022, pp. 5061–5117. |
| [DKPP23] | I. Diakonikolas, D. M. Kane, A. Pensia, and T. Pittas. "Nearly-Linear Time and Streaming Algorithms for Outlier-Robust PCA". In: *Proc. 40th International Conference on Machine Learning (ICML)*. 2023. |
| [DKS17] | I. Diakonikolas, D. M. Kane, and A. Stewart. "Statistical Query Lower Bounds for Robust Estimation of High-Dimensional Gaussians and Gaussian Mixtures". In: *58th IEEE Annual Symposium on Foundations of Computer Science, FOCS 2017*. Full version at http://arxiv.org/abs/1611.03473. 2017, pp. 73–84. |

[DKS19]     I. Diakonikolas, W. Kong, and A. Stewart. "Efficient Algorithms and Lower Bounds for Robust Linear Regression". In: *Proc. 30th Annual Symposium on Discrete Algorithms (SODA)*. 2019.

[DL22]      J. Depersin and G. Lecué. "Robust Sub-Gaussian Estimation of a Mean Vector in Nearly Linear Time". In: *The Annals of Statistics* 50.1 (Feb. 2022), pp. 511–536.

[DM22]      A. S. Dalalyan and A. Minasyan. "All-In-One Robust Estimator of the Gaussian Mean". In: *The Annals of Statistics* 50 (2022).

[HLZ20]     S. B. Hopkins, J. Li, and F. Zhang. "Robust and Heavy-Tailed Mean Estimation Made Simple, via Regret Minimization". In: *Advances in Neural Information Processing Systems 33 (NeurIPS)*. 2020.

[HR09]      P. J. Huber and E. M. Ronchetti. *Robust Statistics*. John Wiley & Sons, 2009.

[HR21]      L. Hu and O. Reingold. "Robust Mean Estimation on Highly Incomplete Data with Arbitrary Outliers". In: *Proc. 24th International Conference on Artificial Intelligence and Statistics (AISTATS)*. 2021.

[Hub64]     P. J. Huber. "Robust Estimation of a Location Parameter". In: *The Annals of Mathematical Statistics* 35.1 (Mar. 1964), pp. 73–101.

[JLT20]     A. Jambulapati, J. Li, and K. Tian. "Robust sub-gaussian principal component analysis and width-independent schatten packing". In: *Advances in Neural Information Processing Systems 33 (NeurIPS)* (2020). arxiv preprint at https://arxiv.org/abs/2006.06980.

[KKM18]     A. Klivans, P. K. Kothari, and R. Meka. "Efficient Algorithms for Outlier-Robust Regression". In: *Proc. 31st Annual Conference on Learning Theory (COLT)*. 2018.

[KSKO20]    W. Kong, R. Somani, S. Kakade, and S. Oh. "Robust Meta-learning for Mixed Linear Regression with Small Batches". In: *Advances in Neural Information Processing Systems 33 (NeurIPS)*. 2020.

[KSS18]     P. K. Kothari, J. Steinhardt, and D. Steurer. "Robust Moment Estimation and Improved Clustering via Sum of Squares". In: *Proc. 50th Annual ACM Symposium on Theory of Computing (STOC)*. ACM Press, 2018, pp. 1035–1046.

[LLVZ20]    Z. Lei, K. Luh, P. Venkat, and F. Zhang. "A Fast Spectral Algorithm for Mean Estimation with Sub-Gaussian Rates". In: *Proc. 33rd Annual Conference on Learning Theory (COLT)*. 2020.

[LM21]      G. Lugosi and S. Mendelson. "Robust Multivariate Mean Estimation: The Optimality of Trimmed Mean". In: *The Annals of Statistics* 49.1 (2021), pp. 393–410.

[LRV16]     K. A. Lai, A. B. Rao, and S. Vempala. "Agnostic Estimation of Mean and Covariance". In: *Proceedings of FOCS'16*. 2016.

[LY20]      J. Li and G. Ye. "Robust Gaussian Covariance Estimation in Nearly-Matrix Multiplication Time". In: *Advances in Neural Information Processing Systems 33 (NeurIPS)*. 2020.

[MM15]      C. Musco and C. Musco. "Randomized block krylov methods for stronger and faster approximate singular value decomposition". In: *Advances in neural information processing systems* 28 (2015).

[PBR19]     A. Prasad, S. Balakrishnan, and P. Ravikumar. "A Unified Approach to Robust Mean Estimation". In: *CoRR* abs/1907.00927 (July 2019).

[PJL20]     A. Pensia, V. Jog, and P. Loh. "Robust Regression with Covariate Filtering: Heavy Tails and Adversarial Contamination". In: *CoRR* abs/2009.12976 (Sept. 2020).

[Tuk60]     J. W. Tukey. "A survey of sampling from contaminated distributions". In: *Contributions to probability and statistics* 2 (1960), pp. 448–485.

[Ver18]     R. Vershynin. *High-Dimensional Probability: An Introduction with Applications in Data Science*. Cambridge University Press, 2018.

[Wai19]     M. J. Wainwright. *High-Dimensional Statistics: A Non-Asymptotic Viewpoint*. Cambridge University Press, 2019.

[XCM13]     H. Xu, C. Caramanis, and S. Mannor. "Outlier-Robust PCA: The High-Dimensional Case". In: *IEEE Transactions on Information Theory* 59.1 (2013), pp. 546–572.

[ZJS22]     B. Zhu, J. Jiao, and J. Steinhardt. "Robust Estimation via Generalized Quasi-Gradients". In: *Information and Inference: A Journal of the IMA* 11.2 (2022), pp. 581–636.

# Appendix

**Table of Contents**

## A   Additional Related Work

A systematic study of robust statistics was initiated in the 1960s ([Tuk60; Hub64; ABHHRT72]). We refer the reader to the book [HR09] for a classical treatment of this topic. The theoretical investigation in these studies often focused on the asymptotics of sample size going to infinity. A non-asymptotic analysis of the sample complexity can be found in, for example, [CGR18; CGR16; LM21].

Algorithmic high-dimensional robust statistics has been an active area of research since the publication of [DKKLMS16; LRV16]. Various high-dimensional inference tasks have been investigated in the literature: (i) Unsupervised learning tasks include mean estimation [DKKLMS16; LRV16; KSS18; CFB19; DKP20; HLZ20; CTBJ22; HR21], sparse estimation [BDLS17; DKKPP22; DKLP22; DKKPS19], covariance estimation [CDGW19; LY20], and PCA [XCM13; KSKO20; JLT20], and (ii) Supervised learning tasks include linear regression [DKS19; KKM18; CATJFB20; PJL20; Dep20; BP21], and general stochastic optimization tasks [PBR19; DKKLSS19].

Within algorithmic robust statistics, our work is related to a line of work that develops novel algorithmic approaches to robust inference. For example, approaches based on gradient-descent and matrix multiplicative weights update  [CDGS20; ZJS22; HLZ20; CDKGGS22]. In particular, several recent works have developed fast algorithms, often nearly linear time as opposed to simply polynomial time algorithms, for robust inference [CDG19; CDGW19; LY20; ZJS22; CDKGGS22; DHL19; CMY20; DKKLT21; DKKLT22].

# B Additional Preliminaries

In this section, we state the results that will be used in the proofs later. Appendices B.1 and B.2 contain facts pertaining to linear algebra and concentration of measure, respectively. Appendix B.3 records some results from both folklore and prior works in robust statistics that are needed in this paper. Appendix B.4 focuses on the goodness condition and proves the samples complexity and key properties of good sets. Appendix B.5 formally states the guarantee of the multi-directional filtering procedure. Finally, Appendix B.6 presents the improved runtime guarantee of the algorithm in [DKKLMS18].

## B.1 Linear Algebraic Facts

**Fact B.1.** *Let $\mathbf{A}$ be a PSD matrix. Then for any unit vector $x$ and $m \geq 1$, $x^\top \mathbf{A}^m x \geq (x^\top \mathbf{A} x)^m$.*

**Fact B.2.** *If $\mathbf{A} \in \mathbb{R}^{d \times d}$ is symmetric and $p \geq 1$, the Schatten norms of $\mathbf{A}$ satisfy the following: $\|\mathbf{A}\|_{p+1} \leq \|\mathbf{A}\|_p \leq \|\mathbf{A}\|_{p+1} d^{\frac{1}{p(p+1)}}$.*

In the following we let $\mathbb{C}$ denote the set of complex numbers, though throughout the paper we will only work with real matrices.

**Definition B.3** (Gershgorin Discs). *For any complex $n \times n$ matrix $\mathbf{A}$, for $i \in [n]$, let $R_i'(\mathbf{A}) = \sum_{j \neq i} |a_{ij}|$ and let $G(\mathbf{A}) = \bigcup_{i=1}^n \{z \in \mathbb{C} : |z - a_{ii}| \leq R_i'(\mathbf{A})\}$. Each disc $\{z \in \mathbb{C} : |z - a_{ii}| \leq R_i'(\mathbf{A})\}$ is called Gershgorin disc and their union $G(\mathbf{A})$ is called the Gershgorin domain.*

**Fact B.4** (Gershgorin's Disc Theorem). *For any complex $n \times n$ matrix $\mathbf{A}$, all the eigenvalues of $\mathbf{A}$ belong to the Gershgorin domain $G(\mathbf{A})$.*

**Fact B.5.** *If $\mathbf{A}, \mathbf{B}, \mathbf{C}$ are symmetric $d \times d$ matrices with $\mathbf{A} \succeq 0$ and $\mathbf{B} \preceq \mathbf{C}$ then $\operatorname{tr}(\mathbf{AB}) \leq \operatorname{tr}(\mathbf{AC})$.*

The following two linear algebra results follow from the Lieb-Thirring inequality.

**Fact B.6** (Lemma 7 in [JLT20]). *Let $\mathbf{A}, \mathbf{B}$ be positive semidefinite matrices with $\mathbf{B} \preceq \mathbf{A}$ and $p \in \mathbb{N}$. Then, $\operatorname{tr}\left(\mathbf{B}^p\right) \leq \operatorname{tr}\left(\mathbf{A}^{p-1}\mathbf{B}\right)$.*

**Fact B.7** (Lieb-Thirring Inequality; see, e.g., Problem IX.8.1 in [Bha13]). *Let $\mathbf{A}, \mathbf{B}$ be positive semidefinite matrices. For all $k \in \mathbb{N}$, $\operatorname{tr}((\mathbf{AB})^k) \leq \operatorname{tr}(\mathbf{A}^k \mathbf{B}^k)$.*

**Fact B.8.** *Let $\mathbf{A}$ be a PSD matrix. Then, $\|\mathbf{A}\|_F \leq \sqrt{\|\mathbf{A}\|_{\mathrm{op}} \|\mathbf{A}\|_1}$. In particular, if $\|\mathbf{A}\|_{\mathrm{op}} \leq \|\mathbf{A}\|_1/t$, then $\|\mathbf{A}\|_F^2 \leq \|\mathbf{A}\|_1^2/t$.*

**Fact B.9** (See, e.g., [Ber18, Proposition 11.10.34]). *Let $\mathbf{R}$ and $\mathbf{M}$ be two square matrices, then $\|\mathbf{RM}\|_F \leq \min\left((\|\mathbf{R}\|_F \|\mathbf{M}\|_{\mathrm{op}}, \|\mathbf{R}\|_{\mathrm{op}} \|\mathbf{M}\|_F\right)$ and $\|\mathbf{RM}\|_{\mathrm{op}} \leq \|\mathbf{R}\|_{\mathrm{op}} \|\mathbf{M}\|_{\mathrm{op}}$.*

## B.2 Concentration of Measure Facts

**Fact B.10** (Power iteration). *For any positive semidefinite matrix $\mathbf{A} \in \mathbb{R}^{d \times d}$ and $\eta, \delta \in (0, 1)$, if $p > \frac{C}{\eta} \log(d/(\eta\delta))$ for a sufficiently large constant $C$, and $u := \mathbf{A}^p z$ for $z \sim \mathcal{N}(0, \mathbf{I})$, then*

$$\mathbf{Pr}\left[u^\top \mathbf{A} u / \|u\|_2^2 \geq (1 - \eta)\|\mathbf{A}\|_{\mathrm{op}}\right] \geq 1 - \delta.$$

**Definition B.11** (Sub-Gaussian and Sub-exponential Random Variables). *A one-dimensional random variable $Y$ is sub-Gaussian if $\|Y\|_{\psi_2} := \sup_{p \geq 1} p^{-1/2} \mathbf{E}\left[|Y|^p\right]$ is finite. We say that $\|Y\|_{\psi_2}$ is the sub-Gaussian norm of $Y$. A random vector $X$ in $\mathbb{R}^d$ is sub-Gaussian if for every $v \in \mathcal{S}^{d-1}$, $\|v^\top X\|_{\psi_2}$ is finite. The sub-Gaussian norm of the vector is defined to be*

$$\|X\|_{\psi_2} := \sup_{v \in \mathcal{S}^{d-1}} \|v^\top X\|_{\psi_2}.$$

*We call a centered one-dimensional random variable $Y$ a $(\nu, \alpha)_+$ sub-exponential if $\mathbf{E}[\exp(\lambda Y)] \leq \nu^2 \lambda^2 / 2$ for all $\lambda \leq 1/\alpha$. We call $\|Y\|_{\psi_1} := \sup_{p \geq 1} p^{-1} \mathbf{E}[|Y|^p]$ the sub-exponential norm of $Y$. We extend our definition to vectors as before.*

**Lemma B.12** (Properties of Sub-exponential Random Variables [Wai19; BLM13]). *The class of sub-exponential random variables satisfy the following:*

1. *[Wai19, Proposition 2.9] If $Y$ is a centered $(\nu, \alpha)_+$ sub-exponential random variable, then with probability $1 - \delta$, $Y \lesssim \nu\sqrt{\log(1/\delta)} + \alpha \log(1/\delta)$.*

2. *[BLM13, Theorem 2.3] If $Y$ is a centered random variable satisfying that for all $\delta \in (0, 1)$, $Y \le \nu\sqrt{\log(1/\delta)} + \alpha \log(1/\delta)$, then $Y$ is $(\nu', \alpha')_+$ sub-exponential with $\nu' \lesssim \nu + \alpha$ and $\alpha' \lesssim \alpha$.*

3. *[Wai19, Section 2.1.3] Let $Y_1, \ldots, Y_k$ be $k$ centered independent $(\nu, \alpha)_+$ sub-exponential random variables. Then $\sum_{i=1}^{k} Y_i$ is a $(\nu\sqrt{k}, \alpha)_+$ sub-exponential random variable.*

**Fact B.13** (Hanson-Wright Inequality). *Let $X \sim \mathcal{N}(0, \mathbf{I})$ Then, for every $t \ge 0$:*

$$\mathbf{Pr}[|X^\top \mathbf{A} X - \mathbf{E}[X^\top \mathbf{A} X]| > t] \le 2 \exp\left(-0.1 \min\left(\frac{t^2}{K^4 \|\mathbf{A}\|_{\mathrm{F}}^2}, \frac{t}{K^2 \|\mathbf{A}\|_{\mathrm{op}}}\right)\right) \ .$$

**Fact B.14** (Johnson-Lindenstrauss Sketch). *Fix a set of $n$ points $x_1, \ldots, x_n \in \mathbb{R}^d$. Let $g(x) := \|\mathbf{A}(x - b)\|_2^2$ be a polynomial for some $\mathbf{A} \in \mathbb{R}^{d \times d}$ and $b \in \mathbb{R}^d$, and let $\mathbf{U}$ be the (random) matrix having as rows the vectors $u_i = \mathbf{A} z_i / \sqrt{k}$ for $z_i \sim \mathcal{N}(0, \mathbf{I}_d)$, $i = 1, \ldots, k$. Define $\tilde{g}(x) := \|\mathbf{U}(x - b)\|_2^2$. If $C$ is a sufficiently large constant and $k > C \log((n + d)/\delta)$, then with probability at least $1 - \delta$, the following holds:*

1. *$0.8 g(x_i) \le \tilde{g}(x_i) \le 1.2 g(x_i)$ for every $i \in [n]$,*

2. *$0.8 \|\mathbf{A}\|_{\mathrm{F}}^2 \le \|\mathbf{U}\|_{\mathrm{F}}^2 \le 1.2 \|\mathbf{A}\|_{\mathrm{F}}^2$.*

**Fact B.15.** *For any $d \times d$ symmetric matrix $\mathbf{A}$, we have that $\mathrm{Var}_{z, z' \sim \mathcal{N}(0, \mathbf{I}_d)}[\langle \mathbf{A} z, \mathbf{A} z' \rangle^2] = \|\mathbf{A}^2\|_{\mathrm{F}}^2$. If $\mathbf{A}$ is a PSD matrix, then for any $\beta > 0$, it holds $\mathbf{Pr}_{z \sim \mathcal{N}(0, \mathbf{I})}[z^\top \mathbf{A} z \ge \beta \mathrm{tr}(\mathbf{A})] \ge 1 - \sqrt{e\beta}$.*

**Fact B.16** (VC inequality). *Let $\mathcal{F}$ be a class of Boolean functions with finite VC dimension $\mathrm{VC}(\mathcal{F})$ and let a probability distribution $D$ over the domain of these functions. For a set $S$ of $n$ independent samples from $D$*

$$\sup_{f \in \mathcal{F}} \left| \mathbf{Pr}_{X \sim S}[f(X)] - \mathbf{Pr}_{X \sim D}[f(X)] \right| \lesssim \sqrt{\frac{\mathrm{VC}(\mathcal{F})}{n}} + \sqrt{\frac{\log(1/\tau)}{n}} \ ,$$

*with probability at least $1 - \tau$.*

## B.3 Existing Robust Algorithms: Trimmed Mean and [DKPP22]

In this section, we state the guarantees of the trimmed mean and the fast robust mean estimation algorithm from [DKPP22].

We begin with the guarantee of the trimmed mean. The following holds even in the strong contamination model where an adversary can edit $\epsilon$-fraction of the samples arbitrarily. We will need it to estimate the (unknown) variance of the labels in our linear regression algorithm.

**Fact B.17** (Univariate Trimmed Mean, see, e.g., [LM21]). *Let $\epsilon_0$ be a sufficiently small absolute constant. There is an algorithm (trimmed mean) that, for every $\epsilon \in (0, \epsilon_0)$ and a univariate distribution $D$ that has $\psi_1$-norm at most $\sigma^2$ (cf. Definition B.11), given a set of $n \gg \log(1/\delta)/(\epsilon \log^2(1/\epsilon))$ samples from $D$ with corruption at rate $\epsilon$, outputs a $\widehat{\mu}$ such that $|\widehat{\mu} - \mathbf{E}_{X \sim D}[X]| \lesssim \sigma^2 \epsilon \log(1/\epsilon)$ with probability at least $1 - \delta$.*

The following result is implicit in [DKPP22]. It will serve as a preprocessing step to ensure small operator norm of the covariance matrix in our robust mean estimation algorithm.

**Lemma B.18.** *There exists an algorithm for which the following is true. If $S$ is a set of $n$ points in $\mathbb{R}^d$ containing a $G \subset S$ such that the uniform distribution on $G$ satisfies conditions (2.a) and (2.b) of Definition 2.1 with $\alpha = \epsilon/\log(1/\epsilon)$, then the algorithm run on input $S$ and $\epsilon$, outputs weights $w(x) \in [0, 1]$ for each $x \in S$ such that the following hold with probability at least $0.999$:*

1. *$\frac{1}{|G|} \sum_{x \in G} w(x) \ge 1 - \frac{\epsilon}{\log(1/\epsilon)}$.*

2. *Let $Q$ be the discrete distribution on $S$ that assigns probability mass $w(x)/\sum_{x \in S} w(x)$ to each $x \in S$. Then, the top eigenvalue of the covariance matrix of $Q$ is at most $1 + O(\epsilon \log^2(1/\epsilon))$.*

*Moreover, the algorithm runs in time $O(nd\text{polylog}(d/\epsilon))$.*

*Proof.* The algorithm that achieves the guarantee is a slightly modified version of Algorithm 1 in [DKPP22] where: (i) instead of outputting an estimate of the mean, the algorithm outputs the underlying weights that it has been updating and (ii) the "DownweightingFilter" procedure is replaced by Algorithm 3 with $\beta = \log(1/\epsilon)$, i.e., the only difference is on the threshold that is used to decide when to stop filtering. Algorithm 3 increases the threshold by the factor $\beta = \log(1/\epsilon)$, which as we show in Lemma B.27 results in removing $\log(1/\epsilon)$ times more mass from the outliers than inliers. The increase in the filtering threshold translates to having an extra $\log(1/\epsilon)$ factor in Item 2 (without this increased threshold, the variance along every direction would have been $\epsilon \log(1/\epsilon)$), and the fact that this filter now removes $\log(1/\epsilon)$ times more mass from the outliers than inliers translates to having $\epsilon$ divided by $\log(1/\epsilon)$ in Item 1. $\square$

## B.4   Goodness Condition and Its Sample Complexity

We first restate the definition of the conditions. Recall that for a weight function $w : \mathbb{R}^d \to [0,1]$ we let $G_w$ denote weighted by $w$ version of the distribution $G$. The matrix $\overline{\Sigma}_{G_w}$ below denotes $\mathbf{E}_{X \sim G_w}[(X - \mu)(X - \mu)^\top]$ (as opposed to the covariance matrix $\Sigma_{G_w} := \mathbf{E}_{X \sim G_w}[(X - \mu_{G_w})(X - \mu_{G_w})^\top]$).

**Definition 2.1** ( $(\epsilon, \alpha, k)$-Goodness)**.** *We say that a distribution $G$ on $\mathbb{R}^d$ is $(\epsilon, \alpha, k)$-good with respect to $\mu \in \mathbb{R}^d$, if the following conditions are satisfied:*

*(1) (Median) For all $v \in \mathcal{S}^{d-1}$, $\mathbf{Pr}_{X \sim G}[|v^\top(X - \mu)| \gtrsim \epsilon] < 1/2$.*

*(2) For every weight function $w$ with $\mathbf{E}_{X \sim G}[w(X)] \geq 1 - \alpha$, the following hold:*

> *(2.a) (Mean) $\|\mu_{G_w} - \mu\|_2 \lesssim \alpha \sqrt{\log(1/\alpha)}$,*
> *(2.b) (Covariance) $\|\overline{\Sigma}_{G_w} - \mathbf{I}_d\|_{\text{op}} \lesssim \alpha \log(1/\alpha)$,*
> *(2.c) (Concentration along nearly-orthogonal vectors) For any $\mathbf{U} \in \mathbb{R}^{k \times d}$ with $\text{tr}(\mathbf{U}^\top \mathbf{U}) = k$ and $\|\mathbf{U}^\top \mathbf{U}\|_{\text{op}} \leq 2k/\log(k/\epsilon)$, the degree-2 polynomial $p(x) := \|\mathbf{U}(x - \mu)\|_2^2$ satisfies $\mathbf{E}_{X \sim G_w}[p(X)\mathbb{1}(p(x) > 100k)] \leq \epsilon/\log(1/\epsilon)$.*

**Remark B.19.** We note that once the conditions of Definition 2.1 are established, then they also hold for projections of the data on a subspace, something that we will need later on in our proofs. Concretely, let $\mathcal{V}$ be any subspace of $\mathbb{R}^d$. Then Conditions (1), (2.a) and (2.b) continue to hold for the projected points $\text{Proj}_{\mathcal{V}}(x)$. This is because all of these conditions concern properties of one-dimensional projections of the data. Regarding the third condition, if the rows of $\mathbf{U}$ belong in the subspace $\mathcal{V}$ (which will be the case in our analysis later on), the last condition also continues to hold for the points $\text{Proj}_{\mathcal{V}}(x)$ since the $\|\mathbf{U}\text{Proj}_{\mathcal{V}}(x)\|_2^2 = \|\mathbf{U}\Pi_{\mathcal{V}}x\|_2^2 = \|\mathbf{U}x\|_2^2$, where we used that $\mathbf{U}\Pi_{\mathcal{V}} = \mathbf{U}$ because the rows of $\mathbf{U}$ already live in the space $\mathcal{V}$.

### B.4.1   Sample Complexity

In our results, we use the goodness condition from Definition 2.1 with parameters $\alpha = \epsilon/\log(1/\epsilon)$ and $k = \text{polylog}(d/\epsilon)$. The following result implies that a set of $\frac{(d+\log(1/\tau))\text{polylog}(d/\epsilon)}{\epsilon^2}$ i.i.d. samples from the inliers satisfies the desired stability condition.

**Lemma B.20.** *Let $\epsilon_0$ be a small enough positive constant, let $\epsilon \in (0, \epsilon_0)$, and let $\tau \in (0, 1)$. Let $S \subset \mathbb{R}^d$ be a set of $n$ independent samples from $\mathcal{N}(\mu, \mathbf{I}_d)$ for $n \gg \frac{k^3(d \log(d/\epsilon) + \log(1/\tau))}{\min(\alpha^2, \epsilon^2/\log^2(1/\epsilon))}$. If $G$ denotes the uniform distribution over $S$, then with probability at least $1 - \tau$, $G$ is $(\epsilon, \alpha, k)$-good.*

*Proof.* We prove the results for each condition separately.

**Condition (1)**   As the VC dimension of Linear Threshold Functions (LTFs) is $d$, the VC inequality (Theorem B.16) implies that there exists a sufficiently large universal constant $C$ such that for any unit vector $v$ and $\gamma$, $|\mathbf{Pr}_{X \sim S}[v^\top(X - \mu) > \gamma] - \mathbf{Pr}_{X \sim \mathcal{N}(0,1)}[X > \gamma]| \leq C(\sqrt{d/n} + \sqrt{\log(1/\tau)/n})$ with probability $1 - \tau$.

We next apply Taylor's Theorem to the complementary cumulative function of the Gaussian, $\bar{\Phi}(t) := \mathbf{Pr}_{X \sim \mathcal{N}(0,1)}[X > t] = (1/\sqrt{2\pi}) \int_t^\infty e^{-x^2/2}\mathrm{d}x$. The first two derivatives are $\bar{\Phi}'(t) =$

$-e^{-t^2/2}/\sqrt{2\pi}$ and $\bar{\Phi}''(t) = te^{-t^2/2}/\sqrt{2\pi}$ and thus there exists a $\xi$ between $0$ and $t$ such that $\bar{\Phi}(t) = 1/2 - t/\sqrt{2\pi} + t^2\xi e^{-\xi^2/2}/(2\sqrt{2\pi}) \leq 1/2 - t/\sqrt{2\pi} + t^3/(2\sqrt{2\pi})$. Combining with the VC inequality, with probability $1 - \tau$, we have that

$$\Pr_{X \sim G}[v^\top(X - \mu) > \gamma] \leq 1/2 - \gamma/\sqrt{2\pi} + \gamma^3/(2\sqrt{2\pi}) + C(\sqrt{d/n} + \sqrt{\log(1/\tau)/n}) \,. \quad (2)$$

By our assumption that $\sqrt{d/n} + \sqrt{\log(1/\tau)/n}$ is $O(\epsilon)$ and thus taking $\gamma = \Theta(\epsilon)$, we obtain that the right hand side is less than $1/2$.

**Conditions (2.a) and (2.b)**   Let $G$ be the uniform distribution over a set of $n$ samples drawn from $\mathcal{N}(0, \mathbf{I})$. We recall the following result from the recent robust statistics literature:

**Lemma B.21** ([DK19; DKKLMS16]). *Let $\alpha \in (0, \epsilon_0)$ for a small enough positive constant $\epsilon_0$. Let $P$ be an $O(1)$-subgaussian distribution with mean $\mu$ and covariance $\Sigma$, where $\|\Sigma - \mathbf{I}\|_{\mathrm{op}} \leq \alpha\log(1/\alpha)$. The uniform distribution over a set of $n \gg (d + \log(1/\delta))/\alpha^2$ points drawn i.i.d. from $P$ satisfies Conditions (2.a) and (2.b) from Definition 2.1 with probability $1 - \delta$.*

**Condition (2.c)**   First we argue that $\mathbf{E}_{X \sim \mathcal{N}(\mu, \mathbf{I})}[\mathbb{1}(p(X) > 100\mathrm{tr}(\mathbf{U}^\top\mathbf{U}))] < \epsilon/\log(1/\epsilon)$ for all the polynomials $p(x)$ described in Condition (2.c). Fix such a $p(x)$ and observe that $\mathbf{E}[p(x)] = \mathrm{tr}(\mathbf{U}^\top\mathbf{U})$. Under the Gaussian distribution, by Hanson-Wright inequality (Fact B.13), we have that

$$\Pr_{X \sim \mathcal{N}(\mu, \mathbf{I})}\left[\left|(X - \mu)^\top\mathbf{U}^\top\mathbf{U}(X - \mu) - \mathrm{tr}(\mathbf{U}^\top\mathbf{U})\right| > t\right] \leq 2\exp\left(-0.1\min\left(\frac{t^2}{\|\mathbf{U}^\top\mathbf{U}\|_F^2}, \frac{t}{\|\mathbf{U}^\top\mathbf{U}\|_{\mathrm{op}}}\right)\right) \,.$$
$$(3)$$

Let $k' := \log(k/\epsilon)/2$. Using our assumption $\|\mathbf{U}^\top\mathbf{U}\|_{\mathrm{op}} \leq \mathrm{tr}(\mathbf{U}^\top\mathbf{U})/k'$, Fact B.8 implies that $\|\mathbf{U}^\top\mathbf{U}\|_F^2 \leq \mathrm{tr}(\mathbf{U}^\top\mathbf{U})^2/k'$. Plugging in $t = 99\mathrm{tr}(\mathbf{U}^\top\mathbf{U})$ above, we obtain that

$$\Pr_{X \sim \mathcal{N}(\mu, \mathbf{I})}\left[\left|(X - \mu)^\top\mathbf{U}^\top\mathbf{U}(X - \mu) \geq 100\mathrm{tr}(\mathbf{U}^\top\mathbf{U})\right|\right] \leq 2\exp(-48k') = 2\epsilon^{24}/k^{24} \,, \quad (4)$$

where we use the definition of $k'$. Moreover, integrating the tail inequality Equation (3) implies that $\mathbf{E}[p^2(X)] \lesssim (\mathbf{E}[p(X)])^2 + \|\mathbf{U}^\top\mathbf{U}\|_F^2 \lesssim \mathrm{tr}(\mathbf{U}^\top\mathbf{U})^2$. Alternatively, Equation (3) implies that with probability $1 - \delta$,

$$p_{\mathbf{U}}(x)\mathbb{1}(p_{\mathbf{U}}(x) \geq 2\mathbf{E}[p_{\mathbf{U}}(X)]) \leq 2(p_{\mathbf{U}}(x) - \mathbf{E}[p_{\mathbf{U}}(X)])\mathbb{1}(p_{\mathbf{U}}(x) \geq 2\mathbf{E}[p_{\mathbf{U}}(X)])$$
$$\lesssim \sqrt{k^2/k'}\sqrt{\log(1/\delta)} + (k/k')\log(1/\delta)$$

The covering argument is based on the following discretization of the unit sphere.

**Fact B.22** (Cover of the sphere). *Let $r > 0$. There exists a set $\mathcal{C}$ of unit vectors of $\mathbb{R}^d$, such that $|\mathcal{C}| \leq (1 + 2/r)^d$ and for every $v \in \mathcal{S}^{d-1}$ we have that $\min_{y \in \mathcal{C}} \|y - v\|_2 \leq r$.*

Let $\mathcal{V}$ be the set of all nearly orthogonal matrices:

$$\mathcal{V} := \{\mathbf{U} \in \mathbb{R}^{k \times d} : \|\mathbf{U}^\top\mathbf{U}\|_{\mathrm{op}} \leq 2\mathrm{tr}(\mathbf{U}^\top\mathbf{U})/k', \mathrm{tr}(\mathbf{U}^\top\mathbf{U}) = k\} \,.$$

Let $\mathcal{V}_\eta \subset \mathcal{V}$ be the $\eta$-cover of $\mathcal{V}$ for some $\eta \in (0, 1)$ such that for any $\mathbf{U} \in \mathcal{V}$, we have a matrix $\mathbf{U}'$ such that $\|\mathbf{U} - \mathbf{U}'\|_F \leq \eta$. We now bound the cardinality of $\eta$-cover $\mathcal{V}_\eta$. Looking it as a vector in $\mathbb{R}^{dk}$ as flattened vector, there exists an $\eta$-cover of size $2(3/\eta)^{dk}$, where we also use that we can ensure $\mathcal{V}_\eta \subset \mathcal{V}$ using [Ver18, Exercise 4.2.9].

For any $\mathbf{U} \in \mathcal{V}$, define $p_{\mathbf{U}}(x) = \|\mathbf{U}x\|_2^2$ and $\tau_{\mathbf{U}}(x) = p_{\mathbf{U}}(x)\mathbb{1}(p_{\mathbf{U}}(x) \geq 100\mathrm{tr}(\mathbf{U}^\top\mathbf{U}))$. Let $X_1, \ldots, X_n$ be $n$ i.i.d. random variables from $\mathcal{N}(0, \mathbf{I})$. We first claim that for $X \sim \mathcal{N}(0, \mathbf{I})$, $\tau_{\mathbf{U}}(X) - \mathbf{E}[\tau_{\mathbf{U}}(X)]$ is a $(O(k^2/k'), O(k/k'))$-subexponential random variable and $\mathbf{E}[\tau_{\mathbf{U}}(X)] \lesssim \epsilon^3$. For the latter, the Cauchy-Schwarz inequality along with Equation (3) applies that

$$\mathbf{E}[\tau_{\mathbf{U}}(X)] = \mathbf{E}[p_{\mathbf{U}}(X)\mathbb{1}(p_{\mathbf{U}}(x) \geq 100\mathrm{tr}(\mathbf{U}^\top\mathbf{U}))] \leq \sqrt{\mathbf{E}[p_{\mathbf{U}}^2(X)]}\sqrt{\Pr(p_{\mathbf{U}}(X) \geq 100\mathrm{tr}(\mathbf{U}^\top\mathbf{U}))}$$
$$= O(k)O(\epsilon^{12}/k^{12}) \,,$$

which is less than $\epsilon^3$. For the former, we have that with probability $1 - \delta$.

$$\tau_{\mathbf{U}}(x) - \mathbf{E}[\tau_{\mathbf{U}}(X)] \leq p_{\mathbf{U}}(x)\mathbb{1}(p_{\mathbf{U}}(x) \geq 2\,\mathbf{E}[p_{\mathbf{U}}(X)]) \lesssim \sqrt{k^2/k'}\log(1/\delta) + (k/k')\log(1/\delta).$$

Thus, Lemma B.12 implies that $\tau_{\mathbf{U}}(x) - \mathbf{E}[\tau_{\mathbf{U}}(X)]$ is a $(O(k^2/k'), O(k/k'))$ subexponential random variable. Applying the concentration properties of the mean of independent subexponenial random variables (Lemma B.12), we obtain that for any fixed $\mathbf{U} \in \mathcal{V}_\eta$, with probability $1 - \tau$,

$$\sum_{i=1}^{n} \frac{1}{n}\tau_{\mathbf{U}}(x_i) \lesssim \epsilon^3 + \sqrt{\frac{k^2\log(1/\tau)}{k'n}} + \frac{k\log(1/\tau)}{k'n}. \tag{5}$$

Taking a union bound over every $\mathbf{U} \in \mathcal{V}_\eta$, we obtain the following: with probability $1 - \tau$,

$$\forall U' \in \mathcal{V}_\eta : \quad \sum_{i=1}^{n} \frac{1}{n}\tau_{\mathbf{U}'}(x_i) \lesssim \epsilon^3 + \sqrt{\frac{k^2\log(1/\tau)}{k'n}} + \frac{k\log(1/\tau)}{k'n} + \sqrt{\frac{dk^3\log(1/\eta)}{k'n}} + \frac{dk^2\log(1/\eta)}{k'n}. \tag{6}$$

Let $\mathbf{U} \in \mathcal{V}$ and $\mathbf{U}' \in \mathcal{V}_\eta$ be arbitrary such that $\|\mathbf{U} - \mathbf{U}'\|_F \leq \eta$. The goal now is to show that $\sum_{i=1}^{n} \frac{1}{n}\tau_{\mathbf{U}}(x_i)$ and $\sum_{i=1}^{n} \frac{1}{n}\tau_{\mathbf{U}'}(x_i)$ are close to each other and thus the former is bounded similarly to (6). Define the difference of the polynomials $\Delta p(x) := p_{\mathbf{U}}(x) - p_{\mathbf{U}'}(x)$. We will also use the notation $i \sim [n]$ to denote the averaging over $[n]$ samples. First, we have that

$$\mathbf{E}_{i\sim[n]}[\tau_{\mathbf{U}}(x_i)] = \mathbf{E}_{i\sim[n]}[p_{\mathbf{U}}(x_i)\mathbb{1}(p_{\mathbf{U}}(x_i) > 100k)]$$

$$= \mathbf{E}_{i\sim[n]}[p_{\mathbf{U}'}(x_i)\mathbb{1}(p_{\mathbf{U}'}(x_i) > 100k - \Delta p(x_i))] + \mathbf{E}_{i\sim[n]}[\Delta p(x_i)\mathbb{1}(p_{\mathbf{U}}(x_i) > 100k)]$$

$$= \mathbf{E}_{i\sim[n]}[p_{\mathbf{U}'}(x_i)\mathbb{1}(p_{\mathbf{U}'}(x_i) > 100k - \Delta p(x_i))] + \mathbf{E}_{i\sim[n]}[|\Delta p(x_i)|]. \tag{7}$$

We start with the second term of (7).

$$\mathbf{E}_{i\sim[n]}[|\Delta p(x_i)|] = \mathbf{E}_{i\sim[n]}[|\langle \mathbf{U}^\top\mathbf{U} - \mathbf{U}'^\top\mathbf{U}', x_i x_i^\top \rangle|]$$

$$\leq \|\mathbf{U}^\top\mathbf{U} - \mathbf{U}'^\top\mathbf{U}'\|_F \,\mathbf{E}_{i\sim[n]}[\|x_i\|_2^2]$$

$$\lesssim d^2\eta,$$

where we used two things: First that by Gaussian norm concentration (e.g., Hanson-Wright inequality with $A = I$ combined with Lemma B.12):

$$\frac{1}{n}\sum_{i=1}^{n}\|x_i\|_2^2 \leq d + O\left(\sqrt{d}\sqrt{\frac{\log(1/\delta)}{n}} + \frac{\log(1/\delta)}{n}\right) = O(d), \tag{8}$$

if $n \gg \log(1/\delta)$ and, second, we used that

$$\left\|\mathbf{U}^\top\mathbf{U} - \mathbf{U}'^\top\mathbf{U}'\right\|_F = \left\|\mathbf{U}^\top(\mathbf{U} - \mathbf{U}') + (\mathbf{U} - \mathbf{U}')^\top\mathbf{U}'\right\|_F$$

$$\leq 2\max\left(\|\mathbf{U}\|_{op}, \|\mathbf{U}'\|_{op}\right)\|\mathbf{U} - \mathbf{U}'\|_F \leq 2d\eta.$$

We now bound the first term in (7),

$$\mathbf{E}_{i\sim[n]}[p_{\mathbf{U}'}(x_i)\mathbb{1}(p_{\mathbf{U}'}(x_i) > 100k - \Delta p(x_i))]$$

$$= \mathbf{E}_{i\sim[n]}[p_{\mathbf{U}'}(x_i)\mathbb{1}(p_{\mathbf{U}'}(x_i) > 100k - \Delta p(x_i))\mathbb{1}(\Delta p(x_i) \leq k)]$$

$$+ \mathbf{E}_{i\sim[n]}[p_{\mathbf{U}'}(x_i)\mathbb{1}(p_{\mathbf{U}'}(x_i) > 100k - \Delta p(x_i))\mathbb{1}(\Delta p(x_i) > k)]$$

$$\leq \mathbf{E}_{i\sim[n]}[p_{\mathbf{U}'}(x_i)\mathbb{1}(p_{\mathbf{U}'}(x_i) > 99k)] + \mathbf{E}_{i\sim[n]}[|p_{\mathbf{U}'}(x_i)|\mathbb{1}(\Delta p(x_i) > k)] \tag{9}$$

The first term is bounded by (6)[8].

---

[8]There is a small difference in the constant in front of $k$, but the same proof gives similar quantitative bounds.

Choose $\eta = 0.001\epsilon^3/d^2$. Then second term in (9), $\mathbf{E}_{i\sim[n]}[|p_{\mathbf{U}'}(x_i)|\mathbb{1}(\Delta p(x_i) > k)]$ is zero unless there exists an $x_i$ in the sample set with $\|x_i\| > 10\sqrt{d}$. This relies on the fact that $\Delta p(x) \leq \|x\|_2^2\|\mathbf{U}^\top\mathbf{U} - \mathbf{U}'^\top\mathbf{U}\|_{\mathrm{F}} \leq 2d\eta\|x\|_2^2$ which becomes less than $k$ if $\|x\|_2^2 \leq 100d$ and $\eta = 0.001\epsilon^3/d^2$. But the probability of the event $\|x_i\| > 10\sqrt{d}$ is exponentially small:

$$\Pr_{X_1,\ldots,X_n\sim\mathcal{N}(0,I)}[\exists \in S : \|X_i\|_2 > 10\sqrt{d}] \leq ne^{-d/100} , \tag{10}$$

by Gaussian norm concentration and a union bound.

Therefore taking $\eta = 0.001\epsilon^3/d^2$, we have that with high probability the scores from $\mathcal{V}$ and $\mathcal{V}_\eta$ are close up to $\epsilon^3$. Combining this with Equation (6) and the union bound over $\mathcal{E}$ when $n \gg d + \log(1/\delta)$, we obtain that that with probability at least $1 - \delta/2 - ne^{-d/100}$, for all $\mathbf{U} \in \mathcal{V}$:

$$\sum_{i=1}^n \frac{1}{n}\tau_{\mathbf{U}}(x) \lesssim \epsilon^3 + \sqrt{\frac{k^2\log(1/\tau)}{k'n}} + \frac{k\log(1/\tau)}{k'n} + \sqrt{\frac{k^3d\log(d/\epsilon)}{k'n}} + \frac{dk^2\log(d/\epsilon)}{k'n} .$$

Thus, as soon as $n = C\frac{k^3(d\log(d/\epsilon)+\log(1/\tau))}{k'\epsilon^2/\log^2(1/\epsilon)}$ for a sufficiently constant $C$, and $\epsilon \in (0,\epsilon_0)$ a sufficiently small constant $\epsilon_0$, all of the terms above become less than $\epsilon/\log(1/\epsilon)$. We also need $ne^{-d/100}$ to be at most $\delta/2$. Using the same $n$ as before, the quantity $ne^{-d/100}$ is at most $\delta/2$ if $d \gg \mathrm{poly}\log(\epsilon^{-1}\delta^{-1})$. We can assume without loss of generality that the last condition is always true as we can artificially augment the dimension by padding the data with Gaussian coordinates until the resulting dimension becomes $d' \gg \mathrm{poly}\log(\epsilon^{-1}\delta^{-1})$ and it is easy to see that once the goodness conditions get established for the augmented data, they continue to hold for the original data. $\square$

### B.4.2 Helper Lemmata Related to Goodness Condition

We now record implications of the goodness conditions that we will use in our analysis of the main algorithm. The first lemma below provides a certification that when the operator norm of the empirical covariance is suitably bounded, the empirical mean closely approximates the true mean. This allows us to halt the outlier filtering process and return the resulting vector.

**Lemma B.23** (Certificate Lemma (restated)). *Let $0 < \alpha < \epsilon < 1/4$ and $\delta \in (0,1)$. Let $P = (1 - \epsilon)G + \epsilon B$ be a mixture of distributions, where $G$ satisfies Conditions (2.a) and (2.b) of Definition 2.1 with respect to $\mu \in \mathbb{R}^d$. Let $w(x)$ be such that $\mathbf{E}_{X\sim G}[w(X)] > 1 - \alpha$. Assume that the the top eigenvalue of $\boldsymbol{\Sigma}_{P_w}$ is less than $1 + \lambda$. Then,*

$$\|\mu_{P_w} - \mu\|_2 \lesssim \alpha\sqrt{\log(1/\alpha)} + \sqrt{\lambda\epsilon} + \epsilon + \sqrt{\alpha\epsilon\log(1/\alpha)} .$$

*Proof.* Let $\rho = \epsilon\mathbf{E}_{X\sim B}[w(X)]/\mathbf{E}_{X\sim P}[w(X)]$. Denote $P_w(x) = w(x)P(x)/\mathbf{E}_{X\sim P}[w(X)], B_w(x) = w(x)B(x)/\mathbf{E}_{X\sim B}[w(X)], G_w(x) = w(x)G(x)/\mathbf{E}_{X\sim G}[w(X)]$ the weighted by $w(x)$ versions of the distributions $P, B, G$ and denote by $\boldsymbol{\Sigma}_{P_w}, \boldsymbol{\Sigma}_{B_w}, \boldsymbol{\Sigma}_{G_w}$ their covariance matrices. We can write:

$$\boldsymbol{\Sigma}_{P_w} = \rho\boldsymbol{\Sigma}_{B_w} + (1-\rho)\boldsymbol{\Sigma}_{G_w} + \rho(1-\rho)(\mu_{G_w} - \mu_{B_w})(\mu_{G_w} - \mu_{B_w})^\top . \tag{11}$$

Let $v$ be the vector in the direction of $\mu_{G_w} - \mu_{B_w}$. Since the top eigenvalue of $\boldsymbol{\Sigma}_{P_w}$ is less than $1 + \lambda$, we obtain the following:

$$1 + \lambda \geq v^\top\boldsymbol{\Sigma}_{P_w}v \geq (1-\rho)v^\top\boldsymbol{\Sigma}_{G_w}v + \rho(1-\rho)(v^\top(\mu_{B_w} - \mu_{G_w}))^2$$
$$\geq (1-\rho)(1 - \alpha\log(1/\alpha)) + \rho(1-\rho)(v^\top(\mu_{B_w} - \mu_{G_w}))_2^2 ,$$

where the second step uses Equation (11) and the last step uses that $G$ satisfies Conditions (2.a) and (2.b) of Definition 2.1. The expression above implies the following:

$$(v^\top(\mu_{B_w} - \mu_{G_w}))^2 \leq \frac{\lambda + \rho + \alpha\log(1/\alpha)}{\rho(1-\rho)} .$$

We can now bound the error $\|\mu_{P_w} - \mu\|_2$ as follows:

$$\|\mu_{P_w} - \mu\|_2 = |v^\top(\mu_{G_w} - \mu) + \rho(\mu_{B_w} - \mu_{G_w})|$$

$$\leq |v^\top(\mu_{G_w} - \mu)| + \rho|v^\top(\mu_{B_w} - \mu_{G_w})|$$
$$\leq \|\mu_{G_w} - \mu\|_2 + \rho|v^\top(\mu_{B_w} - \mu_{G_w})|$$
$$\leq \alpha\sqrt{\log(1/\alpha)} + \sqrt{\rho}\sqrt{\frac{\lambda + \rho + \alpha\log(1/\alpha)}{1 - \rho}},$$

where the last inequality uses that $G$ satisfies Conditions (2.a) and (2.b). We now use bounds on $\rho$ to simplify the terms. Recall that $\rho = \epsilon\,\mathbf{E}_{X\sim B}[w(X)]/\mathbf{E}_{X\sim P}[w(X)] \leq \epsilon/(1-\alpha)$. As $\alpha < 1/2$, we get that $\rho < 2\epsilon$. In addition, note that $\rho < 1/2$

$$\|\mu_{P_w} - \mu\|_2 \lesssim \alpha\sqrt{\log(1/\alpha)} + \sqrt{\lambda\epsilon} + \epsilon + \sqrt{\alpha\epsilon\log(1/\alpha)}\,.$$

$\square$

Recall the the goodness conditions are phrased in terms of the deviation from the true mean. However, the true mean is unknown to the algorithm and the algorithm will use the empirical mean as an approximation, which introduces. additional errors in our bounds. To analyze these errors, we require the following lemmata.

**Lemma B.24** (see, e.g., Lemma 2.13 in [DKPP22]). *Let $w : \mathbb{R}^d \to [0,1]$ such that $\mathbf{E}_{X\sim G}[w(X)] \geq 1 - \alpha$ and let $G$ be distribution satisfying Conditions (2.a) and (2.b) of Definition 2.1 with respect to $\mu \in \mathbb{R}^d$ with parameters $(\epsilon, \alpha, k)$.[9] For any matrix $\mathbf{U} \in \mathbb{R}^{m\times d}$ and any vector $b \in \mathbb{R}^d$, we have that*

$$\mathop{\mathbf{E}}_{X\sim G_w}\left[\|\mathbf{U}(X - b)\|_2^2\right] = \|\mathbf{U}\|_F^2(1 \pm \alpha\log(1/\alpha)) + \|\mathbf{U}(\mu - b)\|_2^2 \pm 2\alpha\sqrt{\log(1/\alpha)}\,\|\mathbf{U}\|_F^2\|\mu - b\|_2\,.$$

**Lemma B.25.** *Let $w : \mathbb{R}^d \to [0,1]$ such that $\mathbf{E}_{X\sim G}[w(X)] \geq 1 - \alpha$ and let $G$ be an $(\epsilon, \alpha, k)$-good distribution with respect to $\mu \in \mathbb{R}^d$ as in Definition 2.1. For any matrix $\mathbf{U} \in \mathbb{R}^{k\times d}$, if we define the polynomial $\tau(x) := \|\mathbf{U}(x - b)\|_2^2 \mathbb{1}(\|\mathbf{U}(x - b)\|_2^2 > 100\mathrm{tr}(\mathbf{U}^\top\mathbf{U}))$ and assume that $\|\mathbf{U}(\mu - b)\|_2^2 \leq \mathrm{tr}(\mathbf{U}^\top\mathbf{U})$, then it holds*

$$\mathop{\mathbf{E}}_{X\sim G_w}[\tau(X)] \leq \frac{2\epsilon}{\log(1/\epsilon)}\mathrm{tr}(\mathbf{U}^\top\mathbf{U}) + 2\|\mathbf{U}(\mu - b)\|_2^2\,.$$

*Proof.* We have that

$$\|\mathbf{U}(x - b)\|_2^2 \leq 2\|\mathbf{U}(x - \mu)\|_2^2 + 2\|\mathbf{U}(\mu - b)\|_2^2$$
$$\leq 2\|\mathbf{U}(x - \mu)\|_2^2 + 2\mathrm{tr}(\mathbf{U}^\top\mathbf{U})\,,$$

where the first step uses the triangle inequality combined with the inequality $(a + b)^2 \leq 2a^2 + 2b^2$. Using the above, we can write

$$\mathop{\mathbf{E}}_{X\sim G_w}[\tau(X)] \leq 2\mathop{\mathbf{E}}_{X\sim G_w}[\|\mathbf{U}(x - \mu)\|_2^2\mathbb{1}(\|\mathbf{U}(x - b)\|_2^2 > 49\mathrm{tr}(\mathbf{U}^\top\mathbf{U})] + 2\|\mathbf{U}(\mu - b)\|_2^2$$
$$\leq 2\epsilon/\log(1/\epsilon) + 2\|\mathbf{U}(\mu - b)\|_2^2\,,$$

where the last line uses the goodness condition for $G$. $\square$

Lastly, the following lemma shows that if the matrix $\mathbf{U}$ used in the last goodness condition consists of nearly orthogonal vectors, then the matrix satisfies the near-orthogonality condition mentioned there.

**Lemma B.26.** *Let $u_i$ for $i \in [k]$ be vectors in $\mathbb{R}^d$ and define $\mathbf{U} = [u_1, \ldots, u_k]$. If for every $i, j \in [k]$ with $i \neq j$, it holds $|\langle u_i, u_j\rangle| \leq \|u_i\|_2\|u_j\|_2/k^2$, and $\max_i \|u_i\|_2^2 \leq \frac{\log k}{k}\mathrm{tr}(\mathbf{U}^\top\mathbf{U})$ then $\|\mathbf{U}^\top\mathbf{U}\|_{\mathrm{op}} \leq 2\mathrm{tr}(\mathbf{U}^\top\mathbf{U})/(k/\log k)$.*

*Proof.* We note that by the Gershgorin circle theorem (Fact B.4),

$$\|\mathbf{U}^\top\mathbf{U}\|_{\mathrm{op}} \leq \max_{i\in[k]}\|u_i\|_2^2 + \sum_{i\neq j}|\langle u_i, u_j\rangle|$$

---

[9]This lemma does not use the last goodness condition; thus the parameter $k$ does not appear in the conclusion.

$$\leq \max_{i \in [k]} \|u_i\|_2^2 + k^2 \frac{\|u_i\|_2 \|u_j\|_2}{k^2}$$

$$\leq 2 \max_{i \in [k]} \|u_i\|_2^2 \leq 2 \frac{\operatorname{tr}(\mathbf{U}^\top \mathbf{U})}{k/\log(k)} \ .$$

$\square$

## B.5  Filtering Procedure

In this section, we formally describe the filtering procedure (Algorithm 3) and prove its formal guarantee (Lemma 2.5).

---

**Algorithm 3** Down-weighting Filter

---

1: **Input**: Distribution $P$ on $n$ points, weight $w(x)$ for each point $x$, score $\tilde{\tau}(x)$ for every point $x$ with the guarantee $\tilde{\tau}(x) < r$, threshold $T > 0$, parameters $\beta > 0, \ell_{\max} \in \mathbb{Z}_+$.
2: **Output**: New weights $w'(x)$.

3: Initialize $w'(x) = w(x)$.
4: $\ell_{\max} \leftarrow r/(eT)$.
5: **for** $i = 1, \ldots, \ell_{\max}$ **do**      ▷ Can be implemented in $O(\log \ell_{\max})$ with Binary Search
6:     **if** $\mathbf{E}_{X \sim P}[w'(X)\tilde{\tau}(X)] > T\beta$ **then**
7:         $w'(x) \leftarrow w(x)(1 - \tilde{\tau}(x)/r)$.
8:     **else return** $w'$.
9: **return** $w'$.

---

**Lemma B.27** (Filtering Guarantee). *Let $P = (1-\epsilon)G + \epsilon B$ be a mixture of distributions supported on $n$ points and $\beta > 1$. If $(1-\epsilon)\mathbf{E}_{X \sim G}[w(X)\tilde{\tau}(X)] < T$, $\|\tilde{\tau}\|_\infty \leq r$ and $\ell_{\max} > r/T$ then the new weights $w'(x)$ satisfy:*

1. $(1-\epsilon)\mathbf{E}_{X \sim G}[w(X) - w'(X)] < \frac{\epsilon}{\beta - 1}\mathbf{E}_{X \sim B}[w(X) - w'(X)]$

2. $\mathbf{E}_{X \sim P}[w(X)\tilde{\tau}(X)] \leq T\beta.$

*The algorithm can be implemented in $O(n \log(r/T)))$ time.*

*Proof.* The weight removed from inliers is:

$$(1-\epsilon)\mathop{\mathbf{E}}_{X \sim G}[w(X) - w'(X)] = \frac{1-\epsilon}{r}\mathop{\mathbf{E}}_{X \sim G}[w(X)\tilde{\tau}(X)] \leq T/r$$

The weight removed from outliers is:

$$\epsilon \mathop{\mathbf{E}}_{X \sim B}[w(X) - w'(X)] = \frac{\epsilon}{r}\mathop{\mathbf{E}}_{X \sim B}[w(X)\tilde{\tau}(X)]$$
$$= \frac{1}{r}\left(\mathop{\mathbf{E}}_{X \sim P}[w(X)\tilde{\tau}(X)] - (1-\epsilon)\mathop{\mathbf{E}}_{X \sim G}[w(X)\tilde{\tau}(X)]\right)$$
$$\geq \frac{T(\beta - 1)}{r}$$

Regarding runtime, for any $\ell > r/(eT)$ we have that $\mathbf{E}_{X \sim P}[w_\ell(X)\tilde{\tau}(X)] \leq \mathbf{E}_{X \sim P}[w_\ell(X)\tilde{\tau}(X)\exp(-\ell\tilde{\tau}(X)/r)] \leq \frac{r}{e\ell} \leq T$, where we used the inequality $xe^{-\alpha x} \leq 1/(e \cdot \alpha)$.

$\square$

**Lemma 2.5** (Multi-Directional Filtering). *Consider Setting 2.3. Given a nearly-orthogonal matrix $\mathbf{U} \in \mathbb{R}^{k \times d}$ satisfying $\|\mathbf{U}^\top \mathbf{U}\|_{\mathrm{op}} \leq 2\operatorname{tr}(\mathbf{U}^\top \mathbf{U})/\log(1/\epsilon)$, there is an algorithm that reads $\epsilon$, the $n$ points, their weights $w(x)$, and returns weights $w'$ in time $ndk + \operatorname{polylog}(d/\epsilon, \|\mathbf{U}\|_{\mathrm{F}}^2)$ such that*

(i) $\mathbf{E}_{X \sim G}[w(X) - w'(X)] < (\epsilon/\log(1/\epsilon))\mathbf{E}_{X \sim B}[w(X) - w'(X)]$.

(ii) $\epsilon \mathbf{E}_{X \sim B}[w'(X)\|\mathbf{U}(X - \mu_{P_w})\|_2^2] \lesssim \epsilon \operatorname{tr}(\mathbf{U}^\top \mathbf{U})$.

*Proof.* Let $g(x) := \|\mathbf{U}(x - \mu_{P_w})\|_2^2$ and $\tau(x) := g(x)\mathbb{1}(p(x) > 100\mathrm{tr}(\mathbf{U}^\top \mathbf{U}))$. Denote $\mathbf{B}_{P_w} := \boldsymbol{\Sigma}_{P_w} - \mathbf{I}$. Using the goodness conditions and the certificate lemma:

$$\mathop{\mathbf{E}}_{X \sim G}[w(X)\tau(X)] \leq \frac{2\epsilon}{\log(1/\epsilon)}\|\mathbf{U}\|_{\mathrm{F}}^2 + \|\mathbf{U}(\mu - \mu_{P_w})\|_2^2 \tag{12}$$

$$\leq \left(\frac{2\epsilon}{\log(1/\epsilon)} + \|\mu - \mu_{P_w}\|_2^2\right)\|\mathbf{U}\|_{\mathrm{F}}^2 \qquad \text{(by Lemma B.25)}$$

$$\leq \left(\frac{2\epsilon}{\log(1/\epsilon)} + \epsilon^2 + O\left(\|\mathbf{B}_{P_w}\|_{\mathrm{op}}\epsilon\right)\right)\|\mathbf{U}\|_{\mathrm{F}}^2 \tag{13}$$

$$\lesssim \left(\frac{\epsilon}{\log(1/\epsilon)} + \epsilon\|\mathbf{B}_{P_w}\|_{\mathrm{op}}\right)\|\mathbf{U}\|_{\mathrm{F}}^2 \tag{14}$$

$$\lesssim \left(\frac{\epsilon}{\log(1/\epsilon)} + \epsilon^2\mathrm{polylog}(1/\epsilon)\right)\|\mathbf{U}\|_{\mathrm{F}}^2 \qquad \text{(By Setting 2.3)}$$

$$\lesssim \left(\frac{\epsilon}{\log(1/\epsilon)}\right)\|\mathbf{U}\|_{\mathrm{F}}^2 \, , \tag{15}$$

where the first inequality uses Lemma B.25 (this uses that $\|\mathbf{U}(\mu - \mu_{P_w})\|_2^2 \leq \mathrm{tr}(\mathbf{U}^\top \mathbf{U})$ which follows by the certificate lemma and that $\|\mathbf{B}_{P_w}\| < 1$), the second inequality uses $\|\mathbf{U}(x - \mu)\|_2^2 = \mathrm{tr}(\mathbf{U}^\top \mathbf{U}(x - \mu)(x - \mu)^\top)$ and then the fact $\mathrm{tr}(\mathbf{AB}) \leq \mathrm{tr}(\mathbf{A})\|\mathbf{B}\|_{\mathrm{op}}$ for $\mathbf{A} = \mathbf{U}^\top \mathbf{U}, \mathbf{B} = (x - \mu)(x - \mu)^\top$, the third line uses the certificate lemma (Lemma 2.4).

We can thus apply the filtering lemma (Lemma B.27) with $T$ equal to the right hand side of Equation (15), and $\beta = \log(1/\epsilon)$ to obtain that $\mathbf{E}_{X \sim P}[w'(X)\tau(X)] \leq T\beta \lesssim \epsilon\|\mathbf{U}\|_{\mathrm{F}}^2$. In more detail,

$$\epsilon\mathop{\mathbf{E}}_{X \sim B}[w'(X)g_t(X)] \leq \epsilon\left(\mathop{\mathbf{E}}_{X \sim B}[w'(X)\tau_t(X)] + 100\mathrm{tr}(\mathbf{U}^\top \mathbf{U})\right)$$

$$\lesssim \epsilon\mathop{\mathbf{E}}_{X \sim B}[w'(X)\tau_t(X)] + \epsilon\mathrm{tr}(\mathbf{U}^\top \mathbf{U})$$

$$\lesssim T\log(1/\epsilon) + \epsilon\mathrm{tr}(\mathbf{U}^\top \mathbf{U}) \qquad \text{(by Lemma B.27)}$$

$$\lesssim \epsilon\mathrm{tr}(\mathbf{U}^\top \mathbf{U})$$

This completes the proof of part (ii). Part (i) follows directly from Lemma B.27. $\qquad\square$

## B.6 Algorithm from [DKKLMS18]

We now describe (a simplified version of) the algorithm from [DKKLMS18] and state its improved runtime.

---

**Algorithm 4** Optimal Error Robust Mean Estimation Under Huber Contamination

---

1: **Input**: Parameters $\epsilon, r$, and sets $S_1, S_2 \subset \mathbb{R}^d$, where for each $i = 1, 2$ the uniform distribution on $S_i$ is of the form $(1 - \epsilon)G_i + \epsilon B_i$ for some $G_i$ satisfying Definition 2.1 with appropriate parameters.
2: **Output**: A vector in $\mathbb{R}^d$.

3: Let $k := r \log(1/\epsilon)$, and $P$ be the uniform distribution on $S_1$.
4: Calculate naïve estimate $\widehat{\mu}$ s.t. $\|\widehat{\mu} - \mu\|_2 = R$ for $R = O(\sqrt{d \log(1/\epsilon)})$.    $\triangleright$ e.g., geometric mean
5: Initialize $t \leftarrow 0$ and $w_t(x) = \mathbb{1}(\|x - \widehat{\mu}\|_2 \leq 2R)$ for all $x \in \mathbb{R}^d$
6: Denote by $P_t$ the weighted by $w$ version of $P$, by $\boldsymbol{\Sigma}_t$ the covariance of $P_t$.
7: Let $v_{t,1}, \ldots, v_{t,d}$ be the eigenvector decomposition of $\boldsymbol{\Sigma}_t$ and $\lambda_{t,1} \geq \ldots \geq \lambda_{t,d}$ the corresponding eigenvalues.
8: **while** $\sum_{i=1}^{k} \frac{1}{k}\lambda_{t,k} \geq 1 + \lambda$ for $\lambda = \frac{C}{r}\epsilon$ **do**
9:     Define the polynomial $g_t(x) = \|\mathbf{U}_t(x - \mu_t)\|_2^2$, where $\mathbf{U}_t = [v_{t,1}/\sqrt{k}, \ldots, v_{t,k}/\sqrt{k}]^\top$.
10:     Define the score function $\tau_t(x) = g_t(x)\mathbb{1}(g_t(x) > 100/r)$
11:     $w_{t+1} \leftarrow \text{Filter}(P, \epsilon, w_t, \tau_t)$    $\triangleright$ Algorithm 3
12:     $t \leftarrow t + 1$, and update vectors $v_{t,1}, \ldots, v_{t,k}$.
13: Let $\mathcal{V} = \text{span}\{v_1, \ldots, v_k\}$ and $\mathcal{V}^\perp = \text{span}\{v_{k+1}, \ldots, v_d\}$.
14: $\mu_1 \leftarrow \text{Proj}_{\mathcal{V}^\perp}(\mu_t)$.
15: $\mu_2 \leftarrow \text{BruteForce}(S_2, \epsilon, \mathcal{V})$.
16: **return** $\mu_1 + \mu_2$.

---

**Lemma 2.6** (Adapted From [DKKLMS18]). *Let $T$ be a set of $n$ i.i.d. samples from an $\epsilon$-corrupted version of $\mathcal{N}(0, \mathbf{I}_d)$ and let $r, \delta \in (0, 1)$ be parameters. If $n \gg (d + \log(1/\delta))/\epsilon^2)\text{polylog}(d/\epsilon)$, then there is an algorithm that, when having as input $T$, $\epsilon$, and $r$, after time $\tilde{O}(nd^2 + \frac{1}{\epsilon^{2+r}})$, it outputs a vector $\widehat{\mu} \in \mathbb{R}^d$ such that $\|\widehat{\mu} - \mu\|_2 \lesssim \epsilon/r$ with probability $1 - \delta$.*

*Proof Sketch.* The algorithm consists of two parts. The first part (while loop of Line 8) finds weights for the points $w_t(x)$ and subspace $\mathcal{V} = \text{span}\{v_1, \ldots, v_k\}$ such that the weighted empirical mean approximates the true mean in that subspace within $\ell_2$-distance $O(\epsilon/r)$, and the second part runs a brute force procedure (Lemma B.28) to approximate the true mean in the remaining $k$-dimensional subspace. The brute force procedure runs in time exponential to the dimension of that subspace, but since $k = r \log(1/\epsilon)$, it becomes polynomial in $1/\epsilon$. Concretely, the runtime of the brute force step is given in Lemma B.28. The dataset $S_2$ that we use for this brute force step is of smaller cardinality: since the dimension of the subspace is $k$, $n' = O(k/\epsilon^2)$ samples in $S_2$ are enough to satisfy the goodness conditions. Thus, the runtime from Lemma B.28 with $n$ replaced by $n'$ is $O((d + \epsilon^{-2-r})\text{polylog}(d/\epsilon))$.

In the remainder, we sketch the runtime bound for the first stage of the algorithm, by arguing that the number of iterations is $\tilde{O}(d)$. If we let our algorithm compute exact eigenvalues and eigenvectors, each iteration needs $nd^3$ time, resulting in a total of $\tilde{O}(nd^4)$ for the runtime of the while loop of Line 8. This can improved to $\tilde{O}(nd^2)$ by calculating approximate top $k$-eigenvectors (see, e.g., [MM15]).

To show that the number of iterations is $\tilde{O}(d)$, we will show that every time that the filtering procedure of Line 11 is used, it removes at least $\tilde{\Omega}(\epsilon/d)$ mass from the points' weights. Then, given that the outliers themselves have mass at most $O(\epsilon)$, the algorithm would stop after $\tilde{O}(d)$ iterations. The mass removed per iteration is (by definition of the filtering procedure)

$$\mathop{\mathbf{E}}_{X \sim P}[w_t(X) - w_{t+1}(X)] = \frac{\mathbf{E}_{X \sim P}[w_t(X)\tau_t(X)]}{\max_{x:w_t(x)>0} \tau_t(x)} .$$

The denominator is upper bounded by $\max_{x:w_t(x)>0} \tau_t(x) \leq \tilde{O}(d)$ by the prepossessing of Line 3.[10] For the numerator,

$$\mathop{\mathbf{E}}_{X \sim P}[w_t(X)\tau_t(X)] = \mathop{\mathbf{E}}_{X \sim P}[w_t(X)g_t(X)] - \mathop{\mathbf{E}}_{X \sim P}[w_t(X)g_t(X)\mathbb{1}(g_t(X) \leq 100/r)] .$$

---

[10] Since the norm of the samples from the Gaussian distribution is tightly concentrated, the filtering procedure removes $o(\epsilon/ \log(1/\epsilon))$ fraction of inliers.

The first term is at least $(1 + \lambda)$ because of the condition in Line 8. We also claim that the second term is at most $(1 + \lambda/2)$. These two would imply that $\mathbf{E}_{X \sim P}[w_t(X) - w_{t+1}(X)] = \Omega(\lambda) = \Omega(\epsilon)$, which implies that each iteration removes $\tilde{\Omega}(\epsilon/d)$ fraction of points, completing the proof. The remaining claim for bounding $\mathbf{E}_{X \sim P}[w_t(X)g_t(X)\mathbb{1}(g_t(X) \le 100/r)]$ is similar to [DKPP23, Lemma 2.14 (iv)]: We first note that $\mathbf{E}_{X \sim G}[w_t(X)\mathbb{1}(g_t(X) \le 100/r)] = 1 - O(\epsilon/\log(1/\epsilon))$, because at most $O(\epsilon/\log(1/\epsilon))$-fraction of inliers have $g_t(x) > 100/r$ (cf. goodness condition (2.c) in Definition 2.1). Then,

$$\left| \mathop{\mathbf{E}}_{X \sim P}[w_t(X)\mathbb{1}(g_t(X) \le 100/r)] - 1 \right| \le (1 - \epsilon) \left| \mathop{\mathbf{E}}_{X \sim G}[w_t(X)g_t(X)\mathbb{1}(g_t(X) \le 100/r)] - 1 \right|$$
$$+ \epsilon \left| \mathop{\mathbf{E}}_{X \sim B}[w_t(X)g_t(X)\mathbb{1}(g_t(X) \le 100/r)] - 1 \right| .$$

The first term is upper bounded by $\lambda/4$ by the goodness condition (2.b) along with the certificate lemma (Lemma B.23). For the second term we can use the bound

$$\epsilon \left| \mathop{\mathbf{E}}_{X \sim B}[w_t(X)g_t(X)\mathbb{1}(g_t(X) \le 100/r)] - 1 \right| \le \epsilon (100/r + 1) = O(\epsilon/r) < \lambda/4 .$$

$\square$

We will use the following brute force algorithm to estimate the mean in a low-dimensional subspace.

**Lemma B.28.** *There exists an algorithm,* BRUTEFORCE, *that given a set of $n$ samples in $\mathbb{R}^d$ from an $\epsilon$-corrupted version of a distribution $G$ satisfying the first condition in Definition 2.1 and a subspace $\mathcal{V}$ of dimension $k$, runs in time $O\left(dnk + (n \log n + \mathrm{poly}(k))2^{O(k)}\right)$ and finds a vector $\tilde{\mu} \in \mathbb{R}^d$ such that $\|\mathrm{Proj}_{\mathcal{V}}(\mu - \tilde{\mu})\|_2 = O(\epsilon)$.*

*Proof.* The algorithm first projects the points to the $k$-dimensional subspace. It then computes an $\eta$-cover $\mathcal{C}$ of the $k$-dimensional unit sphere, which by Fact B.22 consists of at most $(4/r)^k$ vectors, and for every vector $u \in \mathcal{C}$, it finds an estimate $\mu_u$ for the $u$-projection of the mean satisfying $|u^\top \mu - \mu_u| \le \gamma$ for $\gamma = O(\epsilon)$ being the same as in the right hand side of the first goodness condition in Definition 2.1 (because of that goodness condition, letting $\mu_u$ be just the median in the $u$-direction achieves $\gamma$-error). It then solves the linear program

$$\text{find } x \text{ s.t. } |u^\top x - \mu_u| \le 2\gamma \ \ \forall u \in \mathcal{C}.$$

Let $v \in \mathcal{S}^{d-1}$ be in the direction of $x - \mu$ and $u = \arg\min_{y \in \mathcal{C}} \|v - y\|_2$. Then,

$$\begin{aligned}
\|x - \mu\|_2 = |v^\top(x - \mu)| &\le |u^\top(x - \mu)| + |(v - u)^\top(x - \mu)| \\
&\le |u^\top(x - \mu_u)| + |u^\top(\mu_u - \mu)| + \|v - u\|_2 \|x - \mu\|_2 \\
&\le 4\gamma + \eta \|x - \mu\|_2 ,
\end{aligned} \tag{16}$$

which establishes that $\|x - \mu\|_2 \le 4\gamma/(1 - \eta)$.

Regarding runtime, the projection of the data takes $O(dnk)$ time, solving the median in each direction takes $O((4/\eta)^k n \log n)$ time. For the LP, the separation oracle that checks all the constraints exhaustively needs time $(4/\eta)^k k$ and the LP can be solved by $O(\mathrm{poly}(k) \log(R/r))$ calls to that separation oracle, where $R$, $r$ are bounds on the volume of the feasible set $\eta \le \mathrm{vol}(\{x \in \mathbb{R}^k : |u^\top x - \mu_u| \le \gamma_1 \ \forall u \in \mathcal{C}\}) \le R$. For the lower bound we have that the volume of the feasible set is at least the volume of a $k$-dimensional ball of radius $\gamma$ because $\mu$ is a feasible solution and $\mu + \gamma v$ remains feasible for any unit vector $v$. For the upper bound, we have that the feasible set is included in a $k$-dimensional ball of radius $4\gamma/(1 - \eta)$ by Equation (16). Thus the runtime for the LP is $O(\mathrm{poly}(k) \log(R/r)) = O(\mathrm{poly}(k))$. Finally, choosing $\eta = 1/4$, simplifies the total runtime to $O\left(dnk + (n \log n + \mathrm{poly}(k))2^{O(k)}\right)$. $\square$

## C   Omitted Proofs from Section 3

We begin by stating additional conditions that must hold throughout the algorithm's execution. Subsequently, in Appendices C.2 and C.3, we present a comprehensive analysis of the two cases in our main algorithm. Finally, in Appendix C.4, we combine all the components to conclude the proof of Theorem 1.3.

## C.1 Deterministic Conditions for Algorithm 1

**Condition C.1** (Deterministic Conditions for Algorithm 1). Consider the notation in Algorithm 1. For all $t \in [t_{\max}]$, the following hold:

(i) Spectral norm of $\mathbf{B}_t^\perp$: $\widehat{\lambda}_t \in [0.1\|\mathbf{B}_t^\perp\|_{\mathrm{op}}, 10\|\mathbf{B}_t^\perp\|_{\mathrm{op}}]$.

(ii) Frobenius norm $\|\mathbf{U}_t\|_{\mathrm{F}}^2 \in [0.8k\|\mathbf{M}_t^\perp\|_{\mathrm{F}}^2, 1.2k\|\mathbf{M}_t^\perp\|_{\mathrm{F}}^2]$.

(iii) Scores: For all $x \in \mathrm{support}(P)$, $\|\mathbf{U}_t(x - \mu_t)\|_2^2 / \|\mathbf{M}_t^\perp(x - \mu_t)\|_2^2 \in [0.8k, 1.2k]$.

(iv) Maximum norm: $\max_{j \in [k]} \|v_{t,j}\|_2^2 \leq \frac{\log k}{k}\mathrm{tr}(\mathbf{U}_t^\top \mathbf{U}_t)$.

(v) Probability estimate: $|\widehat{q}_t - q_t| \leq 0.1/(k^2 t_{\max})$.

(vi) $\mathbf{U}_t$ almost orthogonal: For every pair $u_i, u_j$ of distinct rows of $\mathbf{U}_t$ it holds $|\langle u_i, u_j \rangle| \leq \|u_i\|_2 \|u_j\|_2 / k^2$.

(vii) Approximate eigenvector: $u_t$ from Line 19 satisfies $u_t^\perp \mathbf{B}_t^\perp u_t \geq (1 - 1/p)\|\mathbf{B}_t^\perp\|_{\mathrm{op}}$.

**Lemma C.2.** *Condition C.1 is satisfied with probability at least* 0.9.

*Proof.* The spectral norm condition holds by using Fact B.10 with $\eta = 0.9$, $\delta = 0.001/t_{\max}$, and a union bound over the $t_{\max}$ iterations of the algorithm.

The next two conditions hold with probability 0.999 by using Fact B.14 with $\mathbf{A} = \mathbf{M}_t^\perp \sqrt{k}$ and a similar union bound.

Regarding condition (iv), with probability 0.999 we have the following regarding the vectors $v_{t,j}$ from line 15

$$\|v_{t,j}\|_2^2 \lesssim \sqrt{\log(kt_{\max})}\|(\mathbf{M}_t^\perp)^2\|_{\mathrm{F}} + \log(kt_{\max})\|(\mathbf{M}_t^\perp)^2\|_{\mathrm{op}}$$

$$\lesssim \log(kt_{\max})\|\mathbf{M}_t^\perp\|_{\mathrm{F}}^2 \lesssim \frac{\log(kt_{\max})}{k}\|\mathbf{U}_t^\perp\|_{\mathrm{F}}^2 \lesssim \frac{\log(k)}{k}\|\mathbf{U}_t^\perp\|_{\mathrm{F}}^2 ,$$

where the first step uses Hanson-Wright inequality with probability of failure $1/(1000kt_{\max})$, the second step uses the fact $\|\mathbf{A}\|_{\mathrm{F}}^2 \leq \mathrm{tr}(\mathbf{A})\|\mathbf{A}\|_{\mathrm{op}} \leq \mathrm{tr}(\mathbf{A})^2$ applied with $\mathbf{A} = (\mathbf{M}_t^\perp)^2$, the third step is due to condition (ii), and the last one is because $t_{\max} \lesssim \mathrm{poly}(k)$ for the choice of values that we have made for $t_{\max}$ and $k$.

For Condition (v), if $\widehat{q}_t$ is the empirical average of $\mathbb{1}(|\langle \mathbf{M}_t^\perp z, \mathbf{M}_t^\perp z'\rangle| > \|\mathbf{M}_t^\perp z\|_2 \|\mathbf{M}_t^\perp z'\|_2 / k^2)$ over $\Theta(k^4 t_{\max}^2)$ instantiations of $z, z' \sim \mathcal{N}(0, \mathbf{I})$, then the condition holds by a standard Chernoff bound.

Condition (vi) holds by the fact that the algorithm uses $\mathbf{U}_t$ only in the case that $\widehat{q}_t \leq 1/(k^2 t_{\max})$. By condition (v), in that case we have that $\mathbf{Pr}_{z,z' \sim \mathcal{N}(0,\mathbf{I})}[|\langle \mathbf{M}_t^\perp z, \mathbf{M}_t^\perp z'\rangle| > \|\mathbf{M}_t^\perp z\|_2 \|\mathbf{M}_t^\perp z'\|_2 / k^2] \geq 0.9/(k^2 t_{\max})$. Thus, by a union bound, with high constant probability, for every iteration $t$, all pairs of rows of $\mathbf{U}_t$ will have angle with cosine in $[-1/k^2, 1/k^2]$.

Regarding Condition (vii): By Fact B.10 with $\eta = 1/p = 1/\log(d)$ and $\delta = 1/t_{\max}$ (so that we can do a union bound over all iterations of the algorithm), we have that if $p' \gg \log^2(dt_{\max})$ then $u_t^\top \mathbf{B}_t^\perp u_t \geq (1 - 1/p)\|\mathbf{B}_t^\perp\|_{\mathrm{op}}$ with high constant probability throughout the algorithm. $\qquad\square$

## C.2 Case 1

This section contains the details missing from Section 3.1. We restate and prove the following:

**Claim C.3** (Claim 3.2 restated). *Assuming that Condition C.1 is satisfied, for every iteration $t$ that the condition of Line 12 is satisfied, $\phi_{t+1} \leq 0.99\phi_t$. Moreover, $\mathbf{E}_{X \sim G}[w_t(X)] \geq 1 - \epsilon/\log(1/\epsilon)$ and $\mathbf{B}_t^\perp \succeq 0$ throughout the algorithm's execution.*

The reminder of the subsection is dedicated to the proof of the claim. In particular, we focus on the claim $\phi_{t+1} \leq 0.99\phi_t$, since the invariant $\mathbf{E}_{X \sim G}[w_t(X)] \geq 1 - \epsilon/\log(1/\epsilon)$ follows by the guarantee of the filter (Lemma 2.5) rather directly: First, we note that by Lemma B.26, the deterministic

conditions (vi) and (iv) from Condition C.1 imply that the goodness condition (2.c) is applicable to the polynomial $\|\mathbf{U}_t(x-\mu)\|_2^2$ where $\mathbf{U}_t$ is the matrix used in Line 15; thus, Lemma 2.5 is applicable. Second, by that lemma, we have that every time that $w_t(x)$ changes to $w_{t+1}(x)$ it holds that $\mathbf{E}_{X\sim G}[w_t(X)-w_{t+1}(X)] < (\epsilon/\log(1/\epsilon))\,\mathbf{E}_{X\sim B}[w_t(X)-w_{t+1}(X)]$. If $t^*$ denotes the final iteration, this means that $\mathbf{E}_{X\sim G}[w_0(X)-w_{t^*}(X)] < (\epsilon/\log(1/\epsilon))\,\mathbf{E}_{X\sim B}[w_0(X)-w_{t^*}(X)] \le \epsilon/\log(1/\epsilon)$, thus $\mathbf{E}_{X\sim G}[w_t(X)] \ge 1 - \epsilon/\log(1/\epsilon)$ throughout the execution.

We also note, that $\mathbf{E}_{X\sim G}[w_t(X)] \ge 1 - \epsilon/\log(1/\epsilon)$ implies that $\mathbf{B}_t \succeq 0$, and $\mathbf{B}_t^\perp \succeq 0$. This is because, by condition (2.b) of the goodness condition, it holds $\mathbf{\Sigma}_t \succeq (1 - O(\epsilon))\mathbf{I}$. Also, we have already shown that $\mathbf{E}_{X\sim G}[w_t(X)] \ge 1 - \epsilon/\log(1/\epsilon)$. Thus $\mathbf{B}_t := (\mathbf{E}_{X\sim G}[w_t(X)])^2\mathbf{\Sigma}_t - (1 - C_1\epsilon)\mathbf{I} \succeq (1 - O(\epsilon))^2(1 - O(\epsilon))\mathbf{I} - (1 - C_1\epsilon)\mathbf{I} \succeq 0$, if $C_1$ is large enough.

Regarding the potential function $\phi_t$, we will establish the following series of inequalities, for $c$ being some constant of the form $c = O(1/C)$, where $C$ is the constant used in Algorithm 1 (we can assume $c < 0.0001$ by selecting $C$ appropriately large in the algorithm):

$$
\begin{aligned}
\phi_{t+1} = \mathrm{tr}((\mathbf{M}_{t+1}^\perp)^2) &= \|\mathbf{M}_{t+1}^\perp\|_{2p}^{2p} \\
&\le \left(d^{\frac{1}{2p(2p+1)}}\|\mathbf{M}_{t+1}^\perp\|_{2p+1}\right)^{2p} && \text{(by Fact B.2)} \\
&= d^{\frac{1}{2p+1}}\left(\|\mathbf{M}_{t+1}^\perp\|_{2p+1}^{2p+1}\right)^{\frac{2p}{2p+1}} \\
&= d^{\frac{1}{2p+1}}\left(\mathrm{tr}((\mathbf{B}_{t+1}^\perp)^{2p+1})\right)^{\frac{2p}{2p+1}} \\
&\le d^{\frac{1}{2p}}\left(\mathrm{tr}(\mathbf{M}_t^\perp \mathbf{B}_{t+1}^\perp \mathbf{M}_t^\perp)\right)^{\frac{2p}{2p+1}} && \text{(by Fact B.6)} \\
&\le d^{\frac{1}{2p+1}}(c\|\mathbf{B}_t^\perp\|_{\mathrm{op}}\|\mathbf{B}_t^\perp\|_{2p}^{2p})^{\frac{2p}{2p+1}} && \text{(by Lemma C.4 below)} \\
&\le d^{\frac{1}{2p}}c^{\frac{2p}{2p+1}}(\|\mathbf{B}_t^\perp\|_{2p}\|\mathbf{B}_t^\perp\|_{2p}^{2p})^{\frac{2p}{2p+1}} && \text{(by Fact B.2)} \\
&\le 0.99\|\mathbf{B}_t^\perp\|_{2p}^{2p} = 0.99\phi_t . && \text{(since } p \gg \log(d) \text{ and } c < 0.0001\text{)}
\end{aligned}
$$

Where the application of Fact B.6 uses that $\mathbf{B}_{t+1} \preceq \mathbf{B}_t$: this is by definition of the covariance matrix as $\frac{1}{2(\mathbf{E}_{X\sim P}[w_t(X)])^2}\mathbf{E}_{X,Y\sim P}[w_t(X)w_t(Y)(X-Y)(X-Y)^\top]$ which shows that downweighting points can only make $\mathbf{B}_t$ the matrix smaller in PSD order (note that $\mathbf{B}_t$ has the covariance matrix multiplied by $(\mathbf{E}_{X\sim P}[w_t(X)])^2$).

It remains to establish the main inequality that we used above:

**Lemma C.4** (Filtering Implication (Lemma C.4 restated)). *In the context of Algorithm 1, and assuming that Condition C.1 holds, for every round $t \in [t_{\max}]$ that the algorithm enters Line 13, it holds $\mathrm{tr}(\mathbf{M}_t^\perp \mathbf{B}_{t+1}^\perp \mathbf{M}_t^\perp) \le c'\|\mathbf{B}_t^\perp\|_{\mathrm{op}}\|\mathbf{M}_t^\perp\|_F^2$ for some $c' < O(1/C)$ (where $C$ is the constant appearing in the algorithm's pseudocode).*

*Proof.* First, we note that

$$
\begin{aligned}
\mathbf{\Sigma}_{t+1}^\perp &= \mathop{\mathbf{E}}_{X\sim P_{t+1}}\left[(\mathrm{Proj}_{\mathcal{V}_{t+1}^\perp}(X) - \mu_{t+1}^\perp)(\mathrm{Proj}_{\mathcal{V}_{t+1}^\perp}(X) - \mu_{t+1}^\perp)^\top\right] \\
&\preceq \mathop{\mathbf{E}}_{X\sim P_{t+1}}\left[(\mathrm{Proj}_{\mathcal{V}_{t+1}^\perp}(X) - \mu_t^\perp)(\mathrm{Proj}_{\mathcal{V}_{t+1}^\perp}(X) - \mu_t^\perp)^\top\right] \\
&= \frac{1}{\mathbf{E}_{X\sim P}[w_{t+1}(X)]}\mathop{\mathbf{E}}_{X\sim P}[w_{t+1}(X)(\mathrm{Proj}_{\mathcal{V}_{t+1}^\perp}(X) - \mu_t^\perp)(\mathrm{Proj}_{\mathcal{V}_{t+1}^\perp}(X) - \mu_t^\perp)^\top] . \quad (17)
\end{aligned}
$$

To avoid the repetitive notation of $\mathrm{Proj}_{\mathcal{V}_{t+1}^\perp}(X)$ in the remainder of the proof, we introduce the shorthand notation $X^\perp$ to denote the projection of the random variable $X$ to $\mathcal{V}_{t+1}^\perp$. To avoid confusion for later on, we also note that since we are in the case that the algorithm enters Line 13, $\mathcal{V}_{t+1}^\perp = \mathcal{V}_t^\perp$, i.e., no change is done to the subspace and in this subsection it does not matter which of the two subspaces we use in our notation.

We decompose $\mathrm{tr}(\mathbf{M}_t^\perp \mathbf{B}_{t+1}^\perp \mathbf{M}_t^\perp)$ into the contribution from inliers and outliers: Using the definition of $\mathbf{B}_{t+1}^\perp = \mathbf{E}_{X\sim P}[w_{t+1}(X)]^2\mathbf{\Sigma}_{t+1}^\perp - (1 - C_1\epsilon)\mathbf{\Pi}_{\mathcal{V}_t^\perp}$ and Equation (17), we have that

$$\mathrm{tr}(\mathbf{M}_t^\perp \mathbf{B}_{t+1}^\perp \mathbf{M}_t^\perp) \tag{18}$$

$$\leq \operatorname*{\mathbf{E}}_{X\sim P}[w_{t+1}(X)] \operatorname*{\mathbf{E}}_{X\sim P}[w_{t+1}(X)\|\mathbf{M}_t^\perp(X^\perp - \mu_t^\perp)\|_2^2] - (1 - C_1\epsilon)\mathrm{tr}(\mathbf{M}_t^\perp \mathbf{\Pi}_{\mathcal{V}_t^\perp} \mathbf{M}_t^\perp) \tag{19}$$

$$\leq (1-\epsilon) \operatorname*{\mathbf{E}}_{X\sim G}[w_{t+1}(X)\|\mathbf{M}_t^\perp(X^\perp - \mu_t^\perp)\|_2^2] + \epsilon \operatorname*{\mathbf{E}}_{X\sim B}[w_{t+1}(X)\|\mathbf{M}_t^\perp(X^\perp - \mu_t^\perp)\|_2^2] \tag{20}$$

$$- (1 - C_1\epsilon)\mathrm{tr}((\mathbf{M}_t^\perp)^2)$$

$$\leq \operatorname*{\mathbf{E}}_{X\sim G}[w_{t+1}(X)\|\mathbf{U}_t(X^\perp - \mu_t^\perp)\|_2^2] + \frac{1.25\epsilon}{k} \operatorname*{\mathbf{E}}_{X\sim B}[w_{t+1}(X)\|\mathbf{U}_t(X^\perp - \mu_t^\perp)\|_2^2] \tag{21}$$

$$- (1 - C_1\epsilon)\|\mathbf{M}_t^\perp\|_{\mathrm{F}}^2 \ , \tag{22}$$

where the first inequality uses the definition of $\mathbf{B}_t$, Fact B.5 and Equation (17), for the last term in Equation (19) we used that $\mathbf{\Pi}_{\mathcal{V}_t^\perp}\mathbf{M}_t^\perp = \mathbf{M}_t^\perp$ which can be shown as follows: recall that using our definitions $\mathbf{M}_t^\perp = (\mathbf{B}_t^\perp)^p = \left(\mathbf{\Pi}_{\mathcal{V}_t^\perp}\mathbf{B}_t\mathbf{\Pi}_{\mathcal{V}_t^\perp}\right)^p$, then since $\mathbf{\Pi}_{\mathcal{V}_t^\perp}$ is idempotent, multiplying that expression from any side does not change it. For Equation (21) we used $1 - \epsilon < 1$ and the Johnson-Lindenstrauss sketch guarantee (condition (iii) of Condition C.1).

We start by upper bounding the combined contribution of the first and last term. We will need our assumption that $G$ (the uniform distribution on inlier samples) satisfies conditions (2.b) and (2.a) from Definition 2.1 with $\alpha = \epsilon/\log(1/\epsilon)$, in order to apply Lemma B.24 with $\mathbf{U} = \mathbf{M}_t^\perp$ and $b = \mu_t^\perp$:[11]

$$\operatorname*{\mathbf{E}}_{X\sim G}[w_{t+1}(X)\|\mathbf{M}_t^\perp(X^\perp - \mu_t^\perp)\|_2^2] - (1 - C_1\epsilon)\|\mathbf{M}_t^\perp\|_{\mathrm{F}}^2 \tag{23}$$

$$\leq (1+O(\epsilon))\|\mathbf{M}_t^\perp\|_{\mathrm{F}}^2 + \|\mathbf{M}_t^\perp(\mu^\perp - \mu_t^\perp)\|_2^2 + O(\epsilon)\|\mathbf{M}_t^\perp\|_{\mathrm{F}}^2\|\mu^\perp - \mu_t^\perp\|_2 - (1 - C_1\epsilon)\|\mathbf{M}_t^\perp\|_{\mathrm{F}}^2 \tag{24}$$

$$\leq (1+O(\epsilon))\|\mathbf{M}_t\|_{\mathrm{F}}^2 + \|\mathbf{M}_t\|_{\mathrm{F}}^2 O(\|\mathbf{B}_t^\perp\|_{\mathrm{op}}\epsilon) + O(\epsilon)\|\mathbf{M}_t\|_{\mathrm{F}}^2 O\left(\sqrt{\|\mathbf{B}_t^\perp\|_{\mathrm{op}}\epsilon}\right) - (1 - C_1\epsilon)\|\mathbf{M}_t^\perp\|_{\mathrm{F}}^2 \tag{25}$$

$$\leq O(1/C)\|\mathbf{B}_t^\perp\|_{\mathrm{op}}\|\mathbf{M}_t\|_{\mathrm{F}}^2 \ , \tag{26}$$

where Equation (24) uses Lemma B.24, Equation (25) uses $\|\mathbf{M}_t(x^\perp - \mu_t^\perp)\|_2^2 = \mathrm{tr}(\mathbf{M}_t^\top\mathbf{M}_t(x^\perp - \mu_t^\perp)(x - \mu_t^\perp)^\top)$ and then the fact $\mathrm{tr}(\mathbf{A}\mathbf{B}) \leq \mathrm{tr}(\mathbf{A})\|\mathbf{B}\|_{\mathrm{op}}$ for $\mathbf{A} = \mathbf{M}_t^\top\mathbf{M}_t, \mathbf{B} = (x^\perp - \mu_t^\perp)(x - \mu_t^\perp)^\top$ as well as the certificate lemma (Lemma 2.4) which gives a bound where it can be checked that the dominant term is $O(\sqrt{\epsilon\|\mathbf{B}_t^\perp\|_{\mathrm{op}}})$, and Equation (26) uses that $\epsilon < O(1/C)\|\mathbf{B}_t^\perp\|_{\mathrm{op}}$ due to Line 9 of Algorithm 1 and condition (i) of Condition C.1.

We now focus on upper bounding the middle term of Equation (21), where we will use the guarantee of filtering, in particular the second part of Lemma 2.5. By Lemma B.26, the deterministic conditions (vi) (pairwise angles close to 90 degrees) and (iv) from Condition C.1 imply that the goodness condition (2.c) is applicable to the polynomial $\|\mathbf{U}_t(x - \mu)\|_2^2$ where $\mathbf{U}_t$ is the matrix used in Line 15. In other words, the requirement of Lemma 2.5 is satisfied and the lemma is applicable to our case. We thus obtain that

$$\frac{\epsilon}{k} \operatorname*{\mathbf{E}}_{X\sim B}[w_{t+1}(X)\|\mathbf{U}_t(X^\perp - \mu^\perp)\|_2^2] \lesssim \frac{\epsilon}{k}\|\mathbf{U}_t\|_{\mathrm{F}}^2 \lesssim \epsilon\|\mathbf{M}_t\|_{\mathrm{F}}^2 \lesssim \frac{1}{C}\|\mathbf{B}_t^\perp\|_{\mathrm{op}}\|\mathbf{M}_t\|_{\mathrm{F}}^2 \ ,$$

where the second inequality uses the Johnson-Lindenstrauss guarantee (condition (iii) of Condition C.1) and the last inequality uses that $\epsilon < O(1/C)\|\mathbf{B}_t^\perp\|_{\mathrm{op}}$, which is due to Line 9 of Algorithm 1 and Condition (i) of Condition C.1.

$\square$

## C.3  Case 2

This section contains the details missing from Section 3.2. We begin by proving the formal version of Claim 3.4 below:

---

[11]Since we are working with the projection of the data to $\mathcal{V}_t^\perp$, this lemma requires that the goodness conditions hold for the projected data, which as we have noted before holds (see Remark B.19).

**Claim C.5** (Formal version of Claim 3.4). *If* $\mathbf{Pr}_{z,z'\sim\mathcal{N}(0,\mathbf{I})}\left[\frac{|\langle\mathbf{M}_t^\perp z,\mathbf{M}_t^\perp z'\rangle|}{\|\mathbf{M}_t^\perp z\|_2\|\mathbf{M}_t^\perp z'\|_2} > \gamma\right] \geq \alpha$, *then* $\frac{\|\mathbf{M}_t^\perp\|_{\mathrm{op}}^2}{\mathrm{tr}((\mathbf{M}_t^\perp)^2)} \gtrsim \gamma^2\alpha^5$.

*Proof.* For some parameter $\delta > 0$, define the following events for random variables $z, z'$: $\mathcal{E}_1 = \{z : \|\mathbf{M}_t^\perp z\|_2^2 > \mathrm{tr}((\mathbf{M}_t^\perp)^2)\delta\}$, and $\mathcal{E}_2 = \{z : \|\mathbf{M}_t^\perp z'\|_2^2 > \mathrm{tr}((\mathbf{M}_t^\perp)^2)\delta\}$. We have the following series of inequalities:

$$
\begin{aligned}
\alpha &\leq \mathop{\mathbf{Pr}}_{z,z'\sim\mathcal{N}(0,\mathbf{I})}\left[\left|\frac{\langle\mathbf{M}_t^\perp z,\mathbf{M}_t^\perp z'\rangle}{\|\mathbf{M}_t^\perp z\|_2\|\mathbf{M}_t^\perp z'\|_2}\right| > \gamma\right] \\
&\leq \mathop{\mathbf{Pr}}_{z,z'\sim\mathcal{N}(0,\mathbf{I})}\left[\left|\frac{\langle\mathbf{M}_t^\perp z,\mathbf{M}_t^\perp z'\rangle}{\|\mathbf{M}_t^\perp z\|_2\|\mathbf{M}_t^\perp z'\|_2}\right| > \gamma \wedge \mathcal{E}_1 \wedge \mathcal{E}_2\right] + \mathbf{Pr}[\overline{\mathcal{E}_1} \vee \overline{\mathcal{E}_2}] \\
&\leq \mathop{\mathbf{Pr}}_{z,z'\sim\mathcal{N}(0,\mathbf{I})}\left[\left|\frac{\langle\mathbf{M}_t^\perp z,\mathbf{M}_t^\perp z'\rangle}{\mathrm{tr}((\mathbf{M}_t^\perp)^2)\delta}\right| > \gamma \wedge \mathcal{E}_1 \wedge \mathcal{E}_2\right] + 2\sqrt{e}\delta &&(27) \\
&\leq \mathop{\mathbf{Pr}}_{z,z'\sim\mathcal{N}(0,\mathbf{I})}\left[\left|\frac{\langle\mathbf{M}_t^\perp z,\mathbf{M}_t^\perp z'\rangle}{\mathrm{tr}((\mathbf{M}_t^\perp)^2)\delta}\right| > \gamma\right] + 2\sqrt{e}\delta \\
&\leq \frac{1}{\gamma^2}\frac{\mathbf{E}_{z,z'\sim\mathcal{N}(0,\mathbf{I})}[\langle\mathbf{M}_t^\perp z,\mathbf{M}_t^\perp z'\rangle^2]}{\mathrm{tr}((\mathbf{M}_t^\perp)^2)^2\delta^2} + 2\sqrt{e}\delta &&(28) \\
&\leq \frac{1}{\gamma^2}\frac{\|(\mathbf{M}_t^\perp)^2\|_{\mathrm{F}}^2}{\mathrm{tr}((\mathbf{M}_t^\perp)^2)^2\delta^2} + 2\sqrt{e}\delta \;, &&(29)
\end{aligned}
$$

where Equation (27) is obtained by a union bound along with the second part of Fact B.15, Equation (28) is obtained by squaring and using Markov's inequality, and Equation (29) follows by using the first part of Fact B.15.

Picking the value $\delta = \alpha^2/1000$ and reorganizing Equation (29) implies the lower bound $\|(\mathbf{M}_t^\perp)^2\|_{\mathrm{F}}/\mathrm{tr}((\mathbf{M}_t^\perp)^2) \gtrsim \alpha^5\gamma^2$. Combining that with the upper bound $\|\mathbf{A}\|_{\mathrm{F}}^2 \leq \mathrm{tr}(\mathbf{A})\|\mathbf{A}\|_{\mathrm{op}}$ for any PSD matrix $\mathbf{A}$ (Fact B.8), we obtain:

$$
\alpha^5\gamma^2 \lesssim \frac{\|(\mathbf{M}_t^\perp)^2\|_{\mathrm{F}}}{\mathrm{tr}((\mathbf{M}_t^\perp)^2)} \leq \sqrt{\frac{\|(\mathbf{M}_t^\perp)^2\|_{\mathrm{op}}}{\mathrm{tr}((\mathbf{M}_t^\perp)^2)}} \;. \tag{30}
$$

$\square$

We now turn to the second claim, using the same notation as in the pseudocode. The requirement that $\mathbf{B}_t$ is positive semi-definite (PSD) has already been established. As shown in Claim C.3, whenever the weights are changed, we have proven that $\mathbf{B}_t$ remains PSD.

**Claim C.6** (Claim 3.5 restated). *Assume that $\mathbf{B}_t$ is PSD. Let $V_{t+1} = V_t \cup \{u_t\}$ for some unit vector $u_t \in V_t^\perp$, then $\phi_{t+1} \leq \phi_t - u_t^\top(\mathbf{M}_t^\perp)^2 u_t$.*

*Proof.* We first examine the effect that projecting everything to the space perpendicular to $u_t$ has on the matrix $\mathbf{B}_t^\perp$.

**Claim C.7.** *Define $\boldsymbol{\Delta}_t := \mathbf{I} - u_t u_t^\top$. We have the following:*

(i) $\boldsymbol{\Pi}_{t+1}^\perp = \boldsymbol{\Delta}_t \boldsymbol{\Pi}_t^\perp \boldsymbol{\Delta}_t$.

(ii) $\boldsymbol{\Sigma}_{t+1}^\perp = \boldsymbol{\Delta}_t \boldsymbol{\Sigma}_t^\perp \boldsymbol{\Delta}_t$.

(iii) $\mathbf{B}_{t+1}^\perp = \boldsymbol{\Delta}_t \mathbf{B}_t^\perp \boldsymbol{\Delta}_t$.

*Proof.* We prove each of these separately.

**Proof of (i)** We first recall the definition of the orthogonal projection matrix: Given any basis $v_{t,1}, \ldots, v_{t,r} \in \mathbb{R}^d$ of the subspace $\mathcal{V}_t^\perp$ ($r$ is the dimension of the subspace), if $\mathbf{A}_t$ is the $d \times r$ matrix that has the $v_{t,i}$'s as its columns, then the matrix that performs the orthogonal projection to $\mathcal{V}_t^\perp$ is $\mathbf{\Pi}_t^\perp := \mathbf{A}_t(\mathbf{A}_t^\top \mathbf{A}_t)^{-1} \mathbf{A}_t^\top$. Consider the basis that is orthonormal and starts with $v_{t,1} = u_t$ (the vector mentioned in the claim's statement; such a basis exists since $u_t \in \mathcal{V}_t^\perp$). Then, $\mathbf{\Pi}_t^\perp = \mathbf{A}_t \mathbf{A}_t^\top = u_t u_t^\top + \sum_{i=2}^{r} v_{t,i} v_{t,i}^\top$. Now, let $\mathbf{\Pi}_{t+1}^\perp$ be the matrix that projects to the subspace $\mathcal{V}_{t+1}^\perp := \{x \in \mathcal{V}_t^\perp : x \perp u_t\}$. Since $u_t$ itself belongs to $\mathcal{V}_t^\perp$, an orthonormal basis of $\mathcal{V}_{t+1}^\perp$ is $v_{t,2}, \ldots, v_{t,r}$, and thus if we define $\mathbf{A}_{t+1} := [v_{t,2}, \ldots, v_{t,r}]$ then the orthogonal projection matrix for $\mathcal{V}_{t+1}^\perp$ is $\mathbf{\Pi}_{t+1}^\perp = \mathbf{A}_{t+1} \mathbf{A}_{t+1}^\top = \sum_{i=2}^{r} v_{t,i} v_{t,i}^\top$. We claim that this is the same as $\mathbf{\Delta}_t \mathbf{\Pi}_t^\perp \mathbf{\Delta}_t$: Indeed, $\mathbf{\Delta}_t \mathbf{\Pi}_t^\perp \mathbf{\Delta}_t = (\mathbf{\Delta}_t \mathbf{A}_t)(\mathbf{\Delta}_t \mathbf{A}_t)^\top$, and $\mathbf{\Delta}_t \mathbf{A}_t = (\mathbf{I} - u_t u_t^\top) \mathbf{A}_t$ is the matrix with zeroes in the first column and $v_{t,2}, \ldots, v_{t,r}$ in the rest of them, thus $(\mathbf{\Delta}_t \mathbf{A}_t)(\mathbf{\Delta}_t \mathbf{A}_t)^\top = \sum_{i=2}^{r} v_{t,i} v_{t,i}^\top$.

**Proof of (ii)** Let $\mathbf{A}_t$ be exactly as in the previous paragraph. For the specific orthonormal basis $v_{t,1}, \ldots, v_{t,r} \in \mathbb{R}^d$ of $\mathcal{V}_t^\perp$ (where $r$ is the dimension of that subspace) where the first vector has been chosen to be $v_{t,1} = u_t$, we define the $d \times r$ matrix $\mathbf{A}_t = [v_{t,1}, v_{t,2}, \ldots, v_{t,r}] = [u_t, v_{t,2}, \ldots, v_{t,r}]$. Also, let $\mu_t$ be the mean of $P_t$. Then,

$$
\begin{aligned}
\mathbf{\Delta}_t \mathbf{\Sigma}_t^\perp \mathbf{\Delta}_t &= \mathop{\mathbf{E}}_{X \sim P_t} \left[ \mathbf{\Delta}_t (\mathrm{Proj}_{\mathcal{V}_t^\perp}(X - \mu_t))(\mathrm{Proj}_{\mathcal{V}_t^\perp}(X - \mu_t))^\top \mathbf{\Delta}_t \right] \\
&= \mathop{\mathbf{E}}_{X \sim P_t} \left[ \mathbf{\Delta}_t \mathbf{\Pi}_t^\perp (X - \mu_t)(X - \mu_t)^\top \mathbf{\Pi}_t^\perp \mathbf{\Delta}_t \right] \\
&= \mathop{\mathbf{E}}_{X \sim P_t} \left[ \mathbf{\Delta}_t \mathbf{A}_t \mathbf{A}_t^\top (X - \mu_t)(X - \mu_t)^\top \mathbf{A}_t \mathbf{A}_t^\top \mathbf{\Delta}_t \right] && (31) \\
&= \mathop{\mathbf{E}}_{X \sim P_t} \left[ \mathbf{A}_{t+1} \mathbf{A}_{t+1}^\top (X - \mu_t)(X - \mu_t)^\top \mathbf{A}_{t+1} \mathbf{A}_{t+1}^\top \right] \\
&= \mathop{\mathbf{E}}_{X \sim P_t} \left[ (\mathrm{Proj}_{\mathcal{V}_{t+1}^\perp}(X - \mu_t))(\mathrm{Proj}_{\mathcal{V}_{t+1}^\perp}(X - \mu_t))^\top \right] && (32)
\end{aligned}
$$

for the last line we used the expressions for $\mathbf{A}_t, \mathbf{A}_{t+1}$ that we gave in the previous paragraph to conclude that $\mathbf{A}_{t+1} \mathbf{A}_{t+1}^\top = \mathbf{\Pi}_{\mathcal{V}_{t+1}^\perp}$.

**Proof of (iii)** The proof follows by the definition of $\mathbf{B}_{t+1}^\perp = \left(\mathbf{E}_{X \sim P}[w_t(X)]\right)^2 \mathbf{\Sigma}_{t+1}^\perp - (1 - C_1 \epsilon) \mathbf{\Pi}_{\mathcal{V}_{t+1}^\perp}$, the previous two claims, and linearity.

$\square$

We now show how to complete the proof given Claim C.7.

$$
\begin{aligned}
\phi_{t+1} := \mathrm{tr}\left((\mathbf{B}_{t+1}^\perp)^{2p}\right) & \\
= \mathrm{tr}\left((\mathbf{\Delta}_t \mathbf{B}_t^\perp \mathbf{\Delta}_t)^{2p}\right) && \text{(by Claim C.7)} \\
= \mathrm{tr}\left((\mathbf{B}_t^\perp \mathbf{\Delta}_t)^{2p}\right) && \text{(by properties of } \mathbf{\Delta}_t \text{ and trace; see below)} \\
\leq \mathrm{tr}\left((\mathbf{B}_t^\perp)^{2p} \mathbf{\Delta}_t^{2p}\right) && \text{(by Lieb-Thirring inequality (Fact B.7))} \\
= \mathrm{tr}\left((\mathbf{B}_t^\perp)^{2p} \mathbf{\Delta}_t\right) && (\mathbf{\Delta}_t := \mathbf{I} - u_t u_t^\top \text{ is idempotent}) \\
= \mathrm{tr}\left((\mathbf{B}_t^\perp)^{2p} \left(\mathbf{I} - u_t u_t^\top\right)\right) && \text{(by definition of } \mathbf{\Delta}_t) \\
= \mathrm{tr}\left((\mathbf{B}_t^\perp)^{2p}\right) - \mathrm{tr}\left((\mathbf{B}_t^\perp)^{2p} u_t u_t^\top\right) & \\
= \mathrm{tr}\left((\mathbf{B}_t^\perp)^{2p}\right) - u_t^\top (\mathbf{B}_t^\perp)^{2p} u_t\,, && \text{(by cyclic property of trace)}
\end{aligned}
$$

where for the third step one can expand out $(\mathbf{\Delta}_t \mathbf{B}_t^\perp \mathbf{\Delta}_t)^{2p} = \mathbf{\Delta}_t \mathbf{B}_t^\perp \mathbf{\Delta}_t^2 \mathbf{B}_t^\perp \mathbf{\Delta}_t^2 \cdots \mathbf{\Delta}_t^2 \mathbf{B}_t^\perp \mathbf{\Delta}_t$ and observe that: (i) $\mathbf{\Delta}_t^2 = \mathbf{\Delta}_t$ since $\mathbf{\Delta}_t := \mathbf{I} - u_t u_t^\top$, and (ii) using the cyclic property of trace, after applying trace to both sides we can move the first $\mathbf{\Delta}_t$ to the end. $\square$

Finally, we show how the above two claims imply that $\phi_{t+1} \leq (1 - \mathrm{polylog}(\epsilon/d))\phi_t$, for every iteration $t$ that Line 19 of the algorithm is executed:

$$
\begin{aligned}
\phi_{t+1} - \phi_t &\geq u_t^\top (\mathbf{B}_t^\perp)^{2p} u_t && \text{(by Claim C.6)} \\
&\geq (u_t^\top \mathbf{B}_t^\perp u_t)^{2p} && \text{(by Fact B.1)} \\
&= \left( \left(1 - \frac{1}{p}\right) \|\mathbf{B}_t^\perp\|_{\mathrm{op}} \right)^{2p} && \text{(by condition (vii) of Condition C.1)} \\
&\gtrsim \|\mathbf{B}_t^\perp\|_{\mathrm{op}}^{2p} = \|\mathbf{M}_t^\perp\|_{\mathrm{op}}^2 \\
&\gtrsim \frac{\mathrm{tr}((\mathbf{M}_t^\perp)^2)}{\mathrm{polylog}(d/\epsilon)} = \frac{\phi_t}{\mathrm{polylog}(d/\epsilon)} . && \text{(by Claim 3.4 with } \alpha, \gamma = \mathrm{polylog}(d/\epsilon))
\end{aligned}
$$

## C.4 Combining Everything Together

We now describe in more detail the algorithm that achieves Theorem 1.3 and put together the previous parts to derive the guarantee and runtime of the main theorem.

The final algorithm is outlined in Algorithm 5.

---

**Algorithm 5** Full Algorithm

---

1: **Input**: $\epsilon \in (0, 1/2)$, sets $S_1, S_2 \subset \mathbb{R}^d$ datasets, where for each $i = 1, 2$ the uniform distribution on $S_i$ is of the form $(1 - \epsilon)G_i + \epsilon B_i$ for some $G_i$ satisfying Definition 2.1 with appropriate parameters.
2: **Output**: A vector in $\mathbb{R}^d$.

3: Run the algorithm from Lemma B.18 on $S_1$ and let $w(x)$ for $x \in S_1$ be the weights that it outputs. Let $P$ be the distribution assigning mass $w(x)/\sum_{x \in S_1} w(x)$ to every $x \in S_1$.
4: Run Algorithm 1 on input $P$. Let $\mathcal{V}_1$ and $\mu_1$ be the subspace and the vector that it outputs.
5: Project all points from $S_2$ to $\mathcal{V}_1$, i.e., $S_2 \leftarrow \{\mathrm{Proj}_{\mathcal{V}_1}(x) : x \in S_2\}$.
6: Run Algorithm 4 on $S_2$. Let $\mu_2$ be the vector that it outputs.
7: **return** $\mu_1 + \mu_2$.

---

*Proof of Theorem 1.3.* The set $S_1$ is comprised of i.i.d. points from the $\epsilon$-corrupted version of $\mathcal{N}(\mu, \mathbf{I})$ and its size is chosen to be a large enough multiple of $\epsilon^{-2}(d + \log(1/\delta))\mathrm{polylog}(d/\epsilon)$, so that by Lemma B.20 the uniform distribution on $S_1$ is $(\epsilon, \alpha, k)$-good with $\alpha = \epsilon/\log(1/\epsilon)$ and $k = \mathrm{polylog}(d/\epsilon)$. The preprocessing step of Line 3 ensures that the eigenvalues of the covariance matrix are at most $1 + \epsilon \mathrm{polylog}(1/\epsilon)$, which is part of Setting 2.3 and utilized in the analysis of the next steps. The preprocessing step of Line 3 runs in nearly-linear time $O(nd\mathrm{polylog}(d/\epsilon))$.

The step of Line 4 has been analyzed in Section 3 and Appendix C using the potential argument. We have demonstrated that in each iteration of the algorithm, the potential function $\phi_t := \mathrm{tr}((\mathbf{M}_t^\perp)^2)$ undergoes multiplicative reduction: $\phi_{t+1} \leq (1 - \mathrm{polylog}(\epsilon/d))\phi_t$. Initially, the potential is naïvely bounded by $\phi_0 = \mathrm{poly}(d/\epsilon)^{\log d}$. With just $\mathrm{polylog}(d/\epsilon)$ iterations, the potential decreases to $\phi_t \leq \epsilon^{\log d}$, resulting in $\|\mathbf{B}_t^\perp\|_{\mathrm{op}} = O(\epsilon)$ and leading to the termination of Algorithm 1 with $\|\mathrm{Proj}_{\mathcal{V}_1^\perp}(\mu_1 - \mu)\|_2 = O(\epsilon)$.

Each step of the algorithm can be implemented in $O(nd\mathrm{polylog}(d/\epsilon))$ time, for example, computing $(\mathbf{B}^\perp)^{\log d}z$ by iteratively multiplying $z$ by $\mathbf{B}^\perp$. Utilizing the special form of covariance matrices enables each multiplication to be calculated efficiently in $O(nd)$ time.

Since Algorithm 1 only adds one direction per iteration to the subspace $\mathcal{V}_1$, the dimension of $\mathcal{V}_1$ will be $d' = O(\mathrm{polylog}(d/\epsilon))$. We now use a smaller sized second dataset $S_2$ that consists of $\epsilon^{-2}(d' + \log(1/\delta))\mathrm{polylog}(d/\epsilon)$ samples and run Algorithm 4 on $S_2$. The runtime of that algorithm is given in Lemma 2.6, and since we are using $d' = O(\mathrm{polylog}(d/\epsilon))$ for the dimension, the runtime of that algorithm is $O(\epsilon^{-2-r}\mathrm{polylog}(d/\epsilon))$. Finally, since the output $\mu_2$ approximates $\mu$ in the subspace $\mathcal{V}_1$ within error $O(\epsilon)$, combining $\mu_1$ and $\mu_2$ yields an estimate with $\|(\mu_1 + \mu_2) - \mu\|_2 \leq \|\mathrm{Proj}_{\mathcal{V}_1^\perp}(\mu_1 - \mu)\|_2 + \|\mathrm{Proj}_{\mathcal{V}_1}(\mu_2 - \mu)\|_2 = O(\epsilon)$. $\square$

# D Omitted Proofs from Section 4

In this section, we provide the details omitted from Section 4. Appendix D.1 proves the properties of the conditional distribution of the covariates conditioned on the responses being in an interval. Appendix D.2 then uses these properties to show that the samples from this distribution satisfy the goodness condition. Finally, we provide the details missing from the proof sketch in the main body in Appendix D.3.

## D.1 Conditional Distribution of Covariates

We first begin by proving Claim 4.1.

**Claim 4.1** (Conditional Distribution). *Let $(X, y) \in \mathbb{R}^{d+1}$ follow Definition 1.2. Given $a \in \mathbb{R}$, denote by $G_a$ the distribution of $X$ given $y = a$. Similarly, given an interval $I \subset \mathbb{R}$, let $G_I$ represent the conditional distribution of $X$ given $y \in I$. Define $\sigma_y^2 := \sigma^2 + \|\beta\|_2^2$. Then, $G_a = \mathcal{N}\left(\frac{a}{\sigma_y^2}\beta, \mathbf{I}_d - \frac{1}{\sigma_y^2}\beta\beta^\top\right)$ and $G_I = \frac{1}{\mathbf{Pr}[y \in I]} \int_I \phi(a'; 0, \sigma_y^2)\mathcal{N}\left(\frac{a'}{\sigma_y^2}\beta, I_d - \frac{1}{\sigma_y^2}\beta\beta^\top\right)\mathrm{d}a'$, where $\phi(z; 0, \nu^2)$ denotes the pdf of the $\mathcal{N}(0, \nu^2)$ at $z \in \mathbb{R}$.*

*Proof.* Using Fact D.1 with $y_1 = X, y_2 = y, \mu_1 = \mu_2 = 0, \mathbf{\Sigma}_{11} = \mathbf{I}_d, \mathbf{\Sigma}_{12} = \beta, \mathbf{\Sigma}_{21} = \beta^\top, \mathbf{\Sigma}_{22} = \sigma_y^2 = \sigma^2 + \|\beta\|_2^2$.

**Fact D.1.** *If $\begin{bmatrix} y_1 \\ y_2 \end{bmatrix} \sim \mathcal{N}\left(\begin{bmatrix} \mu_1 \\ \mu_2 \end{bmatrix}, \begin{bmatrix} \mathbf{\Sigma}_{11} & \mathbf{\Sigma}_{12} \\ \mathbf{\Sigma}_{21} & \mathbf{\Sigma}_{22} \end{bmatrix}\right)$, then $y_1|y_2 \sim \mathcal{N}(\bar{\mu}, \bar{\mathbf{\Sigma}})$, with $\bar{\mu} = \mu_1 + \mathbf{\Sigma}_{12}\mathbf{\Sigma}_{22}^{-1}(y_2 - \mu_2)$ and $\mathbf{\Sigma}_{11} - \mathbf{\Sigma}_{12}\mathbf{\Sigma}_{22}^{-1}\mathbf{\Sigma}_{21}$.*

For the distribution $G_I$, the claim follows by definition of conditional probability and law of total probability

$$p_{X|y \in I}(x) = \frac{\int_{a' \in I} p_{X|y=a'}(x)p_y(a')\mathrm{d}a'}{\mathbf{Pr}[y \in I]} \; ,$$

and then use the expression for the pdf of $G_{a'}$ that we showed earlier. $\qquad\square$

### D.1.1 Properties of Conditional Distribution

In Section 3, we mainly focused on the case where the underlying distribution is $\mathcal{N}(\mu, \mathbf{I}_d)$. For the purposes of our robust regression algorithm, we show in this section that the goodness conditions are satisfied by the distribution $G_I$ and get as a corollary a robust mean estimator for $\mu_{G_I}$.

**Lemma D.2** (Properties of $G_I$). *Let $I = [a, b]$ be an interval of length $\ell$. Further suppose that $\max(|a|, |b|) \leq 2\sigma_y$. Let $X \sim G_I$ and let $\mu$ and $\mathbf{\Sigma}$ be the mean and covariance of $X$. Then:*

1. *$\left\|\mu - a\beta/\sigma_y^2\right\|_2 \leq \ell\|\beta\|_2/\sigma_y^2$ and $\|\mu\|_2 \leq 4\|\beta\|_2/\sigma_y$*

2. *$\|\mathbf{\Sigma} - \mathbf{I}\|_{\mathrm{op}} \leq 9\|\beta\|_2^2/\sigma_y^2$.*

3. *$\|X - \mu\|_{\psi_2} \lesssim 1 + \|\beta\|_2/\sigma_y$.*

*Proof.* For any $z \in I$, we know that the mean of $G_z$ is equal to $z\beta/\sigma_y^2$ by Claim 4.1. Since $G_I$ is a convex combination of $G_z$ for $z \in I$, it follows that $\mu$ is a convex combination of $a\beta/\sigma_y^2$ and $b\beta/\sigma_y^2$. Therefore, the mean $\mu$ satisfies $\|\mu - a\beta/\sigma_y^2\| \leq |b - a|\|\beta\|_2/\sigma_y^2$.

Similarly, for any $z \in I$, $\mathbf{E}_{W \sim G_z}[WW^\top] = \mathbf{I}_d - \frac{1}{\sigma_y^2}\beta\beta^\top + \frac{z^2}{\sigma_y^4}\beta\beta^\top$ by Claim 4.1. Thus the second moment of $X \sim G_I$ is equal to $\mathbf{E}_{X \sim G_I}[XX^\top] = \mathbf{I}_d - \frac{1}{\sigma_y^2}\beta\beta^\top + \frac{\alpha a^2 + (1-\alpha)b^2}{\sigma_y^4}\beta\beta^\top$ for $\alpha \in [0, 1]$. Thus, we have that

$$\|\mathbf{\Sigma} - \mathbf{I}_d\|_{\mathrm{op}} = \left\|\mathop{\mathbf{E}}_{X \sim G_I}[XX^\top] - \mu\mu^\top - \mathbf{I}_d\right\|_{\mathrm{op}}$$

$$\leq \|\mu\|_2^2 + \frac{1}{\sigma_y^2}\|\beta\|_2^2 + \max(a^2, b^2)\|\beta\|_2^2/\sigma_y^4$$

$$\leq 2\max(a^2, b^2)\|\beta\|_2^2/\sigma_y^4 + \frac{\|\beta\|_2^2}{\sigma_y^2} \leq 9\|\beta\|_2^2/\sigma_y^2.$$

The proof of the last claim is based on the fact that each component of the mixture $G_I$ is Gaussian (and thus sub-Gaussian). For a single one-dimensional Gaussian $Y \sim \mathcal{N}(0, \sigma^2)$, the sub-Gaussian norm is constant, in particular $\|Y\|_{\psi_2} = \sqrt{2/\pi}$.

$$\mathop{\mathbf{E}}_{X \sim G_I} \left[|v^\top(x - \mu_{G_I})|^p\right]^{1/p} = \left(\mathop{\mathbf{E}}_{a \sim I} \mathop{\mathbf{E}}_{X \sim G_a} \left[|v^\top(x - \mu_{G_I})|^p\right]\right)^{1/p}$$

$$= \max_{a \in I} \mathop{\mathbf{E}}_{X \sim G_a} \left[|v^\top(x - \mu_{G_I})|^p\right]^{1/p}$$

$$\leq \max_{a \in I} \mathop{\mathbf{E}}_{X \sim G_a} \left[|v^\top(x - \mu_{G_a})|^p\right]^{1/p} + \|\mu_{G_a} - \mu_{G_I}\|_2$$

$$\leq \sqrt{2p/\pi} + \ell\|\beta\|_2/\sigma_y^2 .$$

Hence, $\left\|v^\top(X - \mu_{G_I})\right\|_{\psi_2} \leq \sqrt{2/\pi} + \ell\|\beta\|_2/\sigma_y^2$ for every direction $v \in \mathcal{S}^{d-1}$.  $\square$

## D.2  Robustness of Goodness Conditions to Spectral Noise

We now show that a large set of i.i.d. samples from the conditional distribution $G_I$ satisfies the goodness condition provided that the interval is close to the origin and the norm of $\beta$ is small.

**Lemma D.3.** *Let $I = [a, b]$ be an interval of length $\ell$. Let $\epsilon_0$ be a small enough constant. Let $\epsilon \in (0, \epsilon_0)$. Further suppose that $\max(|a|, |b|) \leq 2\sigma_y$, $\|\beta\|_2 \lesssim \sigma\epsilon\log(1/\epsilon)$, and $\alpha \geq \epsilon^2$.*

*Let $S$ be a set of $n$ i.i.d. points from the distribution $G_I$. If $n \gg \frac{k^3(d\log(k/\epsilon) + \log(1/\tau))}{\min(\alpha^2, \epsilon^2/\log^2(1/\epsilon))}$ then with probability $1 - \tau$, the uniform distribution on $S$ satisfies $(\epsilon, \alpha, k)$-goodness condition ([Definition 2.1](#)) with respect to $\mu_{G_I}$.*

*Proof.* We show this for each one of the conditions separately.

**Condition [(1)](#)**  We first calculate the true probability of $v^\top(X - \mu_{G_I})$ being larger than $\gamma$. The distribution of $v^\top(X - \mu_{G_a})$ is $\mathcal{N}(0, \tilde{\sigma}_v)$ where $\tilde{\sigma}_v := 1 - (v^\top\beta)^2/\sigma_y^2$. Using Taylor approximation for the cdf of the projected points,

$$\mathop{\mathbf{Pr}}_{X \sim G_a}[v^\top(X - \mu_{G_a}) > \tilde{\sigma}_v t] \leq \frac{1}{2} - \frac{t}{\sqrt{2\pi}} + \frac{t^3}{2\sqrt{2\pi}} .$$

Thus,

$$\max_{v \in \mathcal{S}^{d-1}} \mathop{\mathbf{Pr}}_{X \sim G_I}[v^\top(X - \mu_{G_I}) > t\tilde{\sigma}_v + \max_{a \in I}\|\mu_{G_a} - \mu_{G_I}\|_2]$$

$$\leq \max_{a \in I} \mathop{\mathbf{Pr}}_{X \sim G_a}[v^\top(X - \mu_{G_I}) > t\tilde{\sigma} + \max_{a \in I}\|\mu_{G_a} - \mu_{G_I}\|_2]$$

$$\leq \max_{a \in I} \mathop{\mathbf{Pr}}_{X \sim G_a}[v^\top(X - \mu_{G_a}) > t\tilde{\sigma}]$$

$$\leq \frac{1}{2} - \frac{t}{\sqrt{2\pi}} + \frac{t^3}{2\sqrt{2\pi}} .$$

Applying VC inequality as in [Lemma B.20](#), we obtain that the empirical probability over a set of size $n \gtrsim (d + \log(1/\tau))/\epsilon^2$ samples will satisfy that with probability $1 - \tau$,

$$\max_{v \in \mathcal{S}^{d-1}} \mathop{\mathbf{Pr}}_{X \sim S}[v^\top(X - \mu_{G_I}) > t \max_{v \in \mathcal{S}^{d-1}} \tilde{\sigma}_v + \max_{a \in I}\|\mu_{G_a} - \mu_{G_I}\|] \leq \frac{1}{2} - \frac{t}{\sqrt{2\pi}} + \frac{t^3}{2\sqrt{2\pi}} + \epsilon .$$

The expression on the right is less than $1/2$ for $t \gtrsim \epsilon$. Thus, we obtain that Condition [(1)](#) is satisfied with $\gamma_1 \lesssim \epsilon + \max_{a \in I}\|\mu_{G_a} - \mu_{G_I}\|_2 \lesssim \epsilon + \ell\beta/\sigma_y^2$.

**Conditions (2.a) and (2.b)** The proof of these two conditions for the Gaussian case only uses its sub-Gaussian concentration. From Lemma D.2, we have that $\|X - \mu_{G_I}\|_{\psi_2} \leq \sqrt{2/\pi} + \ell \|\beta\|_2 / \sigma_y^2$ which is less than a constant when $\ell \lesssim \sigma$ and $\|\beta\|_2 \lesssim \sigma$.

Hence, identically to Lemma B.21, we have that if $n \gg (d + \log(1/\tau))/\alpha^2$, for any set $S' \subset S$ with $|S'| \geq (1 - \alpha)|S|$, the sample mean $\mu_{S'}$ and the sample second moment $\overline{\Sigma}_{S'}$ satisfy that $\|\mu_{S'} - \mu_{G_I}\|_2 \lesssim \alpha\sqrt{\log(1/\alpha)}$ and $\|\overline{\Sigma}_{S'} - \Sigma_{G_I}\|_{\mathrm{op}} \lesssim \alpha \log(1/\alpha)$.

Using the triangle inequality, we have that $\|\overline{\Sigma}_{S'} - \mathbf{I}\|_{\mathrm{op}} \leq \|\overline{\Sigma}_{S'} - \mathbf{I}\|_{\mathrm{op}} + \|\overline{\Sigma}_{S'} - \Sigma_{G_I}\|_{\mathrm{op}} \lesssim \alpha \log(1/\alpha) + \epsilon^2 \log^2(1/\epsilon) \lesssim \alpha \log(1/\alpha)$, where we used Lemma D.2 and the assumptions on $\|\beta\|_2$ and $\alpha$. Thus, Conditions (2.a) and (2.b) are satisfied.

**Condition (2.c)** We will follow the same notation as in Lemma B.20 and claim 4.1. Observe that the proof of Lemma B.20 relied on Equation (4). Our goal will be to show that a similar right tail concentration inequality is satisfied for the distribution $G_I$. Since $G_I$ is a convex combination of $G_z$ for $z \in [a, b]$, it suffices to show that each $G_z$ satisfies Equation (4) after small changes.

Let $X \sim G_z$. Then $x = \Sigma_z^{1/2} y + \mu_z$ for $y \sim \mathcal{N}(0, \mathbf{I})$ and $\mu_z = z\beta/\sigma_y^2$ and $\Sigma_z = \mathbf{I} - \beta\beta^\top/\sigma_y^2$. We will show that the polynomial $p_{\mathbf{U}}(x) = \|\mathbf{U}(x - \mu_I)\|_2^2$ satisfies similar right tail concentration inequality as $\|\mathbf{U}y\|_2^2$ in Lemma B.20. Applying triangle inequality,

$$p_{\mathbf{U}}(x) = \|\mathbf{U}(x - \mu_I)\|_2^2 \leq 2\|\mathbf{U}\Sigma_z^{1/2}y\|_2^2 + 2\|\mathbf{U}(\mu_z - \mu_I)\|_2^2$$
$$\leq 2\|\mathbf{U}\Sigma_z^{1/2}y\|_2^2 + 2\|\mathbf{U}\|_{\mathrm{op}}^2\|\mu_z - \mu_I\|_2^2 \,.$$

By Lemma D.2, $\|\mu_z - \mu_I\|_2^2$, is at most $\ell^2\|\beta\|_2^2/\sigma_y^4 \leq 1$ under the assumptions. Thus, we obtain the following deterministic expression:

$$p_{\mathbf{U}}(x) \leq 2\|\mathbf{U}\Sigma_z^{1/2}y\|_2^2 + 2\mathrm{tr}(\mathbf{U}^\top\mathbf{U}). \tag{33}$$

The expectation of $\|\mathbf{U}\Sigma_z^{1/2}y\|_2^2$ is $\mathrm{tr}(\Sigma_z^{1/2}\mathbf{U}^\top\mathbf{U}\Sigma_z^{1/2}) = \mathrm{tr}(\mathbf{U}^\top\mathbf{U}\Sigma_z)$, which is less than $\mathrm{tr}(\mathbf{U}^\top\mathbf{U})$ since $\Sigma \preceq \mathbf{I}$. Combining this with Equation (33), we obtain

$$\mathbb{P}(p_{\mathbf{U}}(x) \geq 100\mathrm{tr}(\mathbf{U}^\top\mathbf{U})) \leq \mathbb{P}(\|\mathbf{U}\Sigma_z^{1/2}y\|_2^2 - \mathbb{E}[\|\mathbf{U}\Sigma_z^{1/2}y\|_2^2] \geq 48\mathrm{tr}(\mathbf{U}^\top\mathbf{U})) \,. \tag{34}$$

Finally, the right tail of $\|\mathbf{U}\Sigma_z^{1/2}y\|_2^2 - \mathbb{E}[\|\mathbf{U}\Sigma_z^{1/2}y\|_2^2]$ depends on $\|\Sigma_z^{1/2}\mathbf{U}^\top\mathbf{U}\Sigma_z^{1/2}\|_{\mathrm{F}}$ and $\|\Sigma_z^{1/2}\mathbf{U}^\top\mathbf{U}\Sigma_z^{1/2}\|_{\mathrm{op}}$ by the Hanson Wright inequality. Both of these expressions are again bounded by $\|\mathbf{U}^\top\mathbf{U}\|_{\mathrm{F}}$ and $\|\mathbf{U}^\top\mathbf{U}\|_{\mathrm{op}}$, respectively as follows:

$$\|\Sigma_z^{1/2}\mathbf{U}^\top\mathbf{U}\Sigma_z^{1/2}\|_{\mathrm{F}} \leq \|\Sigma_z^{1/2}\|_{\mathrm{op}}\|\mathbf{U}^\top\mathbf{U}\|_{\mathrm{F}}\|\Sigma_z^{1/2}\|_{\mathrm{op}} \leq \|\mathbf{U}^\top\mathbf{U}\|_{\mathrm{F}}$$

and

$$\|\Sigma_z^{1/2}\mathbf{U}^\top\mathbf{U}\Sigma_z^{1/2}\|_{\mathrm{op}} \leq \|\Sigma_z^{1/2}\|_{\mathrm{op}}\|\mathbf{U}^\top\mathbf{U}\|_{\mathrm{op}}\|\Sigma_z^{1/2}\|_{\mathrm{op}} \leq \|\mathbf{U}^\top\mathbf{U}\|_{\mathrm{op}} \,,$$

where we use $\|\Sigma_z\|_{\mathrm{op}} \leq 1$ and Fact B.9. Hence, applying Hanson-Wright inequality to $\|\mathbf{U}\Sigma_z^{1/2}y\|_2^2$ along with the above relations, Equation (34) implies that $p_{\mathbf{U}}(x)$ satisfies Equation (4) with $(\epsilon/k)^5$ in the right hand side. The rest of the proof goes through similar to Lemma B.20.

$\square$

Thus, Theorem 3.1 is applicable and we obtain the following algorithm.

**Corollary D.4.** *Let $\epsilon_0$ be a small enough constant. Let $\epsilon \in (0, \epsilon_0), c \in (0, 1), \delta \in (0, 1)$. Let $S \in \mathbb{R}^d$ be an $\epsilon$-corrupted set of $n \gg \frac{d + \log(1/\tau)}{\epsilon^2}\mathrm{polylog}(d/\epsilon)$ points from the distribution $G_I$ of Claim 4.1 with $\ell \leq \sigma/\log(1/\epsilon)$ and $\|\beta\|_2 \leq \sigma\epsilon\log(1/\epsilon)$. Then, the Algorithm 5 produces an estimate $\widehat{\mu}$ such that $\|\widehat{\mu} - \mu_{G_I}\|_2 \leq \epsilon$ with probability at least $1 - \tau$ in time $(nd + 1/\epsilon^{2+c})\mathrm{polylog}(n, d, 1/\epsilon, 1/\tau)$.*

### D.3 Proof of Theorem 1.4

We now provide the details missing from Section 4. The proof sketch there relied heavily on the events defined in Equation (1), restated below:

$$\text{(i) } |\widehat{\sigma}_y^2 - \sigma_y^2| \lesssim \sigma_y^2\epsilon\log(1/\epsilon) \qquad \text{(ii) } |a| > 0.0001\widehat{\sigma}_y \qquad \text{(iii) } \|\widehat{\beta}_I - \mu_{G_I}\|_2 \lesssim \epsilon/c. \tag{35}$$

The details missing from the proof sketch in Section 4 are: (i) the events in Equation (35) hold, the events in Equation (35) suffice for the algorithm's output to be correct, and the proof of Lemma 4.2. We already explained why (i) and (ii) hold. The third holds by Corollary D.4.

We begin by formally showing that if the events in Equation (1) hold, then the algorithm's output is correct.

**Claim D.5.** *Consider the setting and notation of Theorem 1.4 and Section 4. Let $c \in (0, 1)$. Assuming that the following three events hold*

(i) $|\widehat{\sigma}_y^2 - \sigma_y^2| \lesssim \sigma_y^2 \epsilon \log(1/\epsilon)$.

(ii) $|a| > 0.0001 \widehat{\sigma}_y$.

(iii) $\|\widehat{\beta}_I - \mu_{G_I}\|_2 \lesssim \epsilon/c$.

*Then,* $\|\widehat{\beta} - \beta\|_2 \lesssim \sigma\epsilon/c$.

*Proof.* We want to show that given Items (i) to (iii), we have that $\|\widehat{\beta} - \beta\|_2 \lesssim \sigma\epsilon/c$, where $\widehat{\beta} := (\widehat{\sigma}_y^2/a)\widehat{\beta}_I$ is the algorithm's output vector. Using the triangle inequality,

$$\left\|\widehat{\beta} - \beta\right\|_2 = \left\|\frac{\widehat{\sigma}_y^2}{a}\widehat{\beta}_I - \beta\right\|_2 \leq \left\|\frac{\widehat{\sigma}_y^2}{a}\widehat{\beta}_I - \frac{\widehat{\sigma}_y^2}{a}\mu_{G_I}\right\|_2 + \left\|\frac{\widehat{\sigma}_y^2}{a}\mu_{G_I} - \beta\right\|_2. \tag{36}$$

We upper bound each term by $O(\sigma\epsilon/c)$ separately. For the first term we have that

$$\left\|\frac{\widehat{\sigma}_y^2}{a}\widehat{\beta}_I - \frac{\widehat{\sigma}_y^2}{a}\mu_{G_I}\right\|_2 = \frac{\widehat{\sigma}_y^2}{|a|}\left\|\widehat{\beta}_I - \mu_{G_I}\right\|_2$$

$$\lesssim \widehat{\sigma}_y \left\|\widehat{\beta}_I - \mu_{G_I}\right\|_2 \qquad\qquad \text{(by Item (ii) above)}$$

$$\lesssim \widehat{\sigma}_y \epsilon/c \qquad\qquad\qquad\qquad \text{(by Item (iii))}$$

$$\lesssim \sigma_y \epsilon/c \qquad\qquad\qquad \text{(by Item (i) and } \epsilon < 1/2)$$

$$\leq (\sigma + \|\beta\|_2)\,\epsilon/c \qquad\qquad (\sigma_y^2 := \sigma^2 + \|\beta\|_2^2)$$

$$\lesssim (\sigma + \sigma\epsilon\log(1/\epsilon))\,\epsilon/c \qquad \text{(by assumption } \|\beta\|_2 \lesssim \sigma\epsilon\log(1/\epsilon))$$

$$\lesssim \sigma\epsilon/c. \qquad\qquad\qquad\qquad\qquad (\epsilon < 1/2)$$

For the second term of Equation (36), the triangle inequality implies that

$$\left\|\frac{\widehat{\sigma}_y^2}{a}\mu_{G_I} - \beta\right\|_2 \leq \left\|\frac{\sigma_y^2}{a}\mu_{G_I} - \beta\right\|_2 + \frac{|\widehat{\sigma}_y^2 - \sigma_y^2|}{|a|}\|\mu_{G_I}\|_2.$$

We bound the first term above as follows:

$$\left\|\frac{\sigma_y^2}{a}\mu_{G_I} - \beta\right\|_2 = \frac{\sigma_y^2}{|a|}\left\|\mu_{G_I} - \frac{a}{\sigma_y^2}\beta\right\|_2 \lesssim \sigma_y\left\|\mu_{G_I} - \frac{a}{\sigma_y^2}\beta\right\|_2 \qquad \text{(by Item (i))}$$

$$\leq \sigma_y \frac{\ell\|\beta\|_2}{\sigma_y^2} \qquad\qquad\qquad\qquad \text{(by Lemma D.2)}$$

$$\leq \frac{\ell\|\beta\|_2}{\sigma} \lesssim \sigma\epsilon.$$
$$\text{(by definition of } \ell, \text{ Item (i), and assumption } \|\beta\|_2 \lesssim \sigma\epsilon\log(1/\epsilon))$$

The second term can be controlled as follows:

$$\frac{|\widehat{\sigma}_y^2 - \sigma_y^2|}{|a|}\|\mu_{G_I}\|_2 \lesssim \sigma_y\epsilon\log(1/\epsilon)\|\mu_{G_I}\|_2 \qquad\qquad \text{(Items (i) and (ii))}$$

$$\lesssim \sigma_y\epsilon\log(1/\epsilon)\frac{(a+\ell)\|\beta\|_2}{\sigma_y^2} \qquad\qquad \text{(by Lemma D.2)}$$

$$\lesssim \frac{\sigma_y \epsilon \log(1/\epsilon)\widehat{\sigma}_y \|\beta\|_2}{\sigma_y^2} \qquad\qquad (a + \ell \lesssim \widehat{\sigma}_y \text{ by definition of } a, \ell)$$

$$\lesssim \epsilon \log(1/\epsilon)\|\beta\|_2 \qquad\qquad\qquad\qquad\qquad (\text{by Item (i)})$$

$$\lesssim \sigma\epsilon. \qquad\qquad\qquad (\text{by assumption } \|\beta\|_2 \lesssim \sigma\epsilon \log(1/\epsilon))$$

$$\square$$

Finally, we provide the proof of Lemma 4.2 below.

**Lemma 4.2.** *Consider the context of Theorem 1.4 and Algorithm 2. First, for every possible choice of $I$ that can be made in Line 3, $\mathbf{Pr}_{(X,y)\sim G}[y \in I] \gtrsim \ell/\sigma_y$. Second, $\mathbf{E}_I \left[\mathbf{Pr}_{(X,y)\sim B}[y \in I | I]\right] \lesssim \ell/\sigma_y$, where the outer expectation is taken with respect to the random choice of the center of $I$.*

*Proof.* We start with the first claim.

$$\Pr_{(X,y)\sim G}[y \in I] = \Pr_{y\sim\mathcal{N}(0,\sigma_y^2)}[y \in I]$$

$$= \int_{y\in\mathbb{R}} \mathbb{1}(y \in I)\phi(y; 0, \sigma_y^2)\mathrm{d}y \qquad (\phi(y; 0, \sigma_y^2) \text{ denotes the pdf of } \mathcal{N}(0, \sigma_y^2))$$

$$\geq \int_{y\in[-1.1\sigma_y, 1.1\sigma_y]} \mathbb{1}(y \in I)\phi(y; 0, \sigma_y^2)\mathrm{d}y$$

$$\geq (0.2/\sigma_y) \int_{y\in[-1.1\sigma_y, 1.1\sigma_y]} \mathbb{1}(y \in I)\mathrm{d}y$$

$$\geq 0.2\ell/\sigma_y,$$

where we the second line uses that $I \subset [-1.1\sigma_y, 1.1\sigma_y]$ because its center belongs in $[-\widehat{\sigma}_y, \widehat{\sigma}_y]$ and its length is $\ell = \widehat{\sigma}_y/\log(1/\epsilon)$, and $\widehat{\sigma}_y$ is very close to $\sigma_y$ (Item (i)), and $\epsilon \ll 1$. In the third line we used that the pdf of $\mathcal{N}(0, \sigma_y^2)$ is pointwise bigger than a small multiple of $1/\sigma_y$ in the range $[-1.1\sigma_y^2, 1.1\sigma_y^2]$.

We now prove the second claim. First,

$$\mathbf{Pr}_{(X,y)\sim B}[y \in I] \leq \Pr_{(X,y)\sim B}[y \in I, y \in [-\widehat{\sigma}_y - \ell, \widehat{\sigma}_y + \ell]] + \Pr_{(X,y)\sim B}[y \in I, y \notin [-\widehat{\sigma}_y - \ell, \widehat{\sigma}_y + \ell]] \tag{37}$$

$$\leq \Pr_{(X,y)\sim B}[y \in I, y \in [-\widehat{\sigma}_y - \ell, \widehat{\sigma}_y + \ell]] \tag{38}$$

$$\leq \Pr_{(X,y)\sim B}[y \in I \mid y \in [-\widehat{\sigma}_y - \ell, \widehat{\sigma}_y + \ell]]. \tag{39}$$

Now let $B'$ be the conditional distribution $B$ given $y \in [-\widehat{\sigma}_y - \ell, \widehat{\sigma}_y + \ell]$ for saving space. Recall that the interval $I$ is itself a random interval. If we take expectation with respect to $I$ of both sides of Equation (39) we have that

$$\mathbf{Pr}_{I,(X,y)}[y \in I] \leq \mathbf{E}_I\left[\mathbf{E}_{(x,y)\sim B'}[\mathbb{1}(y \in I)]\right] = \mathbf{E}_{(x,y)\sim B'}\left[\mathbf{E}_I[\mathbb{1}(y \in I)]\right]$$

$$\leq \mathbf{E}_{(x,y)\sim B'}\left[\frac{\ell}{2.2\widehat{\sigma}_y}\right] \leq \frac{\ell}{2.2\widehat{\sigma}_y} \lesssim \frac{\ell}{\sigma_y}.$$

where we changed the order of expectations, and used that given a fixed point $y \in [-\widehat{\sigma}_y - \ell, \widehat{\sigma}_y + \ell]$, the probability that this $y$ is being hit by our random interval $I$ is at most the length $\ell$ of $I$ divided by the the length of $[-\widehat{\sigma}_y - \ell, \widehat{\sigma}_y + \ell]$ (which as we have shown before is subset of $[-1.1\sigma_y, 1.1\sigma_y]$. In the last inequality we relate $\widehat{\sigma}_y$ and $\sigma_y$ once more using Item (i). $\square$

# E   Adaptation of Algorithm 1 to the Streaming Setting

In this section, we give a brief sketch of how the robust mean estimation algorithm of Theorem 1.3 can be adapted to the streaming setting using the techniques of [DKPP22]. The adaptation follows closely [DKPP22] and we thus omit the details.

We recall the data access model for the streaming setting:

**Definition E.1** (Single-Pass Streaming Model). *Let $S$ be a fixed set. In the one-pass streaming model, the elements of $S$ are revealed one at a time to the algorithm, and the algorithm is allowed a single pass over these points.*

The samples that form the set $S$ are still samples from an $\epsilon$-contaminated version of $\mathcal{N}(\mu, \mathbf{I})$ as before. The adaptation of our main result to that model is the following:

**Theorem E.2** (Almost Linear-Time and Streaming Algorithm for Robust Mean Estimation). *Let $\epsilon_0$ be a sufficiently small positive constant. There is an algorithm that, given parameters $\epsilon \in (0, \epsilon_0)$, $c \in (0, 1), \delta \in (0, 1)$, reads a stream of $\mathrm{poly}(d/\epsilon)$ points from an $\epsilon$-corrupted version of $\mathcal{N}(\mu, \mathbf{I}_d)$ (cf. Definition 1.1), computes an estimate $\widehat{\mu}$ such that $\|\widehat{\mu} - \mu\|_2 = O(\epsilon/c)$ with probability at least $1 - \delta$. Moreover, the algorithm runs in time $(nd + 1/\epsilon^{2+c})\mathrm{polylog}(d/\epsilon\delta)$, and uses additional memory $\tilde{O}(d + \mathrm{poly}(1/\epsilon))$.*

We also restrict our attention to Algorithm 1, which is the first stage of the overall algorithm. This is because the second stage, Algorithm 4, is only utilized when the dimension has been reduced to just a $\mathrm{polylog}(d/\epsilon)$, thus one can simulate the offline algorithm in the streaming setting by storing the whole dataset, which has size $O(\mathrm{polylog}(d/\epsilon)/\epsilon^2)$ (by "size" we mean the number of scalar numbers that the dataset consists of).

**Main Differences from [DKPP22]** For the streaming setting, we change our viewpoint and instead of denoting by $P = (1 - \epsilon)G + \epsilon B$ the uniform distribution over a fixed dataset, we now denote by $P$ the underlying data distribution itself (in our case $G = \mathcal{N}(\mu, \mathbf{I})$ or more generally any distribution satisfying the goodness conditions of Definition 2.1). Consequently, the quantities $\mathbf{\Sigma}_t, \mathbf{B}_t, \mathbf{M}_t, \mathbf{\Sigma}_t^\perp, \mathbf{B}_t^\perp, \mathbf{M}_t^\perp$, mentioned in the pseudocode and in our proofs, are all population-level quantities. Instead of having direct access to them (as in Section 3), the algorithm is free to use the stream of fresh i.i.d. samples from $P$ and build estimators to approximate them. The estimators needed for this are the same as in [DKPP22], for which we will give an overview later on. Apart from this difference, the algorithm from Section 3 is easy to implement in the streaming setting with low memory: The operation of multiplying a vector $z$ by an empirical covariance matrix $\sum_x (x - \mu)(x - \mu)^\top$ (which was used frequently in Section 3) can be implemented in the streaming setting without storing the entire covariance matrix, if one calculates first the inner product $(x - \mu)^\top z$ in the sum and then multiply that scalar result with $(x - \mu)$. Second, the weight function $w_t(x)$ that the algorithm maintains for re-weighting points is now defined over the entire $\mathbb{R}^d$ (and not just over a fixed dataset, since there is no such dataset anymore). The algorithm can store a representation of $w_t$ by storing the matrices $\mathbf{U}_t$ and how many times Line 7 of the filter has been executed (having these in memory then given any point $x \in \mathbb{R}^d$ the algorithm can calculate $w_t(x)$). Since the number of iterations $t_{\max} = O(\mathrm{polylog}(d/\epsilon))$ and the size of each $\mathbf{U}_t$ is $O(d\,\mathrm{polylog}(d/\epsilon))$, the memory needed to store the weights is $O(d\,\mathrm{polylog}(d/\epsilon))$ overall.

The proof of correctness of the resulting streaming algorithm follows the same arguments as before, but needs to account extra errors for the approximations taking place. Instead of giving a complete formal proof, we state the new deterministic conditions that are needed in Condition E.3. Then, we discuss why each of them is needed and how it can be obtained.

**Condition E.3** (Deterministic Conditions for Algorithm 6). Let $S_{\mathrm{cover}}$ be the cover set of [DKPP22, Lemma 4.9] For all $t \in [t_{\max}]$, the following hold:

(i) Scores: For every $x \in S_{\mathrm{cover}}$, it holds $\|\mathbf{M}_t^\perp(x - \mu_t)\|_2^2 \gtrsim \|\mathbf{U}_t(x - \mu_t)\|_2^2 - \|\mathbf{M}_t^\perp\|_F^2$.

(ii) Let $T := \widehat{\lambda}_t \|\mathbf{U}_t\|_F^2$. For every $w : \mathbb{R}^d \to [0, 1]$, the algorithm has access to an estimator $f(w)$ for the quantity $\mathbf{E}_{X \sim P}[w(X)\tilde{\tau}_t(X)]$, where $\tilde{\tau}_t(X) = \|\mathbf{U}_t(x - \mu_t)\|_2^2 \mathbb{1}(\|\mathbf{U}_t(x - \mu_t)\|_2^2 > 100\mathrm{tr}(\mathbf{U}_t^\top \mathbf{U}_t))$ such that $f(w) > T_t/2$ whenever $\mathbf{E}_{X \sim P}[w(X)\tau_t(X)] > T_t$.

(iii) Spectral norm of $\mathbf{B}_t^\perp$: $\widehat{\lambda}_t \in [0.1\|\mathbf{B}_t^\perp\|_{\mathrm{op}}, 10\|\mathbf{B}_t^\perp\|_{\mathrm{op}}]$.

(iv) Frobenius norm $\|\mathbf{U}_t\|_F^2 \in [0.8k\|\mathbf{M}_t^\perp\|_F^2, 1.2k\|\mathbf{M}_t^\perp\|_F^2]$.

(v) Maximum norm: $\max_{j \in [k]} \|v_{t,j}\|_2^2 \leq \frac{\log k}{k}\mathrm{tr}(\mathbf{U}_t^\top \mathbf{U}_t)$.

(vi) Probability estimate: $|\widehat{q}_t - q_t| \leq 0.1/(k^2 t_{\max})$.

(vii) $\mathbf{U}_t$ almost orthogonal: For every pair $u_i, u_j$ of distinct rows of $\mathbf{U}_t$ it holds $|\langle u_i, u_j \rangle| \leq \|u_i\|_2 \|u_j\|_2 / k^2$.

(viii) Approximate eigenvector: The vector $u_t$ from Line 19 satisfies $u_t^\top \mathbf{B}_t^\perp u_t \geq (1 - 1/p) \|\mathbf{B}_t^\perp\|_{\mathrm{op}}$.

**Condition (i)** This condition is used instead of Condition (iii) in Condition C.1 to go from Equation (20) to Equation (21). The issue with the old condition condition is that in the streaming setting there is no fixed dataset for which we can require the approximation to hold. Instead, we can use the technique from [DKPP22] and define an appropriate cover $S_{\mathrm{cover}}$ for which $\mathbf{E}_{X \sim B}[w_{t+1}(X) \|\mathbf{M}_t^\perp (X^\perp - \mu^\perp)] \approx \mathbf{E}_{X \sim S}[w_{t+1}(X) \|\mathbf{M}_t^\perp (X^\perp - \mu^\perp)]$ (see Section 4.2.1 of [DKPP22] for more details on the cover argument). Finally, we remark that the additional term $\|\mathbf{M}_t^\perp\|_{\mathrm{F}}^2$ is due to the fact that the matrix $\mathbf{U}_t$ is calculated in Line 17 based on approximations of $\mathbf{M}_t$ and not $\mathbf{M}_t$ itself (see [DKPP22, Lemma 4.15]). That error term can be tolerated in the steps of the correctness proof that we provided in Appendix C.

**Condition (ii)** Since the stopping condition of the filtering procedure (Algorithm 3) depends on population-level quantities, it needs to be estimated within sufficient accuracy. The estimator for that and the correctness of the resulting filter has been analyzed in [DKPP22].

**Condition (iii)** An approximate top eigenvector for Condition (iii) is obtained by multiplying a random Gaussian vector by $\log d$ fresh sample-based estimators of $\mathbf{B}_t^\perp$ (i.e., multiply $z \sim \mathcal{N}(0, \mathbf{I})$ by $\widehat{\mathbf{M}}_t^\perp$, where $\widehat{\mathbf{M}}_t^\perp$ as in Line 14). This estimator has been analyzed in [DKPP22, page 24].

**Conditions (v) and (vi)** These follow similarly to the corresponding conditions of Appendix C. For Condition (vi), the estimator now will use fresh estimates $\widehat{\mathbf{M}}_t^\perp$ of the form of Line 14, instead of the population-level $\mathbf{M}_t^\perp$.

**Condition (viii)** This condition can be obtained by [DKPP23, Lemma 4.3] and [DKPP22, Lemma 4.15].

**Algorithm 6** Robust Mean Estimation Under Huber Contamination In Streaming Model (Stage 1)

---

**Input**: Parameter $\epsilon \in (0, 1/2)$, stream of i.i.d. points from the distribution $P = (1 - \epsilon)G + \epsilon B$, where $G$ satisfies Definition 2.1 with appropriate parameters.

**Output**: An approximation of the mean in a subspace $\mathcal{V}^\perp$ and the orthogonal subspace $\mathcal{V}$.

1: Let $C$ be a sufficiently large constant, $k = C \log^2(n + d)$, $t_{\max} = (\log(d/\epsilon))^C$.
2: Initialize $V_1 \leftarrow \emptyset$ and $w_1(x) = 1$ for all $x \in \mathbb{R}^d$.
3: **for** $t = 1, \ldots, t_{\max}$ **do**
4:      Let $\mathcal{V}_t$ be the subspace spanned by the vectors in $V_t$, and $\mathcal{V}_t^\perp$ be the perpendicular subspace.
5:      Let $P_t$ be the distribution $P$ re-weighted by $w_t$, i.e., $P_t(x) = w_t(x)P(x)/\mathbf{E}_{X \sim P}[w(X)]$.
6:      Let $\mu_t^\perp, \mathbf{\Sigma}_t^\perp$ be the mean and covariance of $\mathrm{Proj}_{\mathcal{V}_t}(X)$ when $X \sim P_t$
7:      Define $\mathbf{B}_t^\perp = (\mathbf{E}_{X \sim P}[w_t(X)])^2 \mathbf{\Sigma}_t^\perp - (1 - C_1\epsilon)\mathbf{\Pi}_{\mathcal{V}_t^\perp}$, where $\mathbf{\Pi}_{\mathcal{V}_t^\perp}$ is the orthogonal projection matrix for $\mathcal{V}_t^\perp$, and $\mathbf{M}_t^\perp := (\mathbf{B}_t^\perp)^p$ for $p = \log(d)$.
8:      Calculate $\widehat{\lambda}_t$ such that $\widehat{\lambda}_t/\|\mathbf{B}_t^\perp\|_{\mathrm{op}} \in [0.1/10]$ using power iteration.        ▷ cf. Appendix B
9:      **If** $\widehat{\lambda}_t \leq C\epsilon$ **then return** $\mu_t$ and $V_t$.
10:      Let $q_t := \mathbf{Pr}_{z,z' \sim \mathcal{N}(0,\mathbf{I})}[|\langle \mathbf{M}_t^\perp z, \mathbf{M}_t^\perp z'\rangle| > \|\mathbf{M}_t^\perp z\|_2 \|\mathbf{M}_t^\perp z'\|_2/k^2]$.
11:      Calculate an estimate $\widehat{q}_t$ such that $|\widehat{q}_t - q_t| \leq \frac{1}{10(k^2 t_{\max})}$.
12:      **if** $\widehat{q}_t \leq 1/(k^2 t_{\max})$ **then**                  ▷ Case 1 (cf. Section 3.1)
13:          **for** $j \in [k]$ **do**
14:              Let $\widehat{\mathbf{B}}_{t,j,\ell}^\perp$ for $\ell \in [\log d]$ sample-based versions of $\mathbf{B}_t^\perp$.
15:              Let $\widehat{\mathbf{M}}_t^\perp := \prod_{\ell=1}^{\log d} \widehat{\mathbf{B}}_{t,j,\ell}^\perp$.
16:              $v_{t,j} \leftarrow \widehat{\mathbf{M}}_t^\perp z_{t,j}$ for $z_{t,j} \sim \mathcal{N}(0, \mathbf{I})$.
17:          $\mathbf{U}_t \leftarrow [v_{t,1}, \ldots, v_{t,k}]^\top$ i.e., the matrix with rows $v_{t,j}$ for $j \in [k]$.
18:          $w_{t+1} \leftarrow$ MULTI-DIRECTIONALFILTER$(P, w, \epsilon, \mathbf{U}_t)$      ▷ cf. [DKPP22, Algorithm 4]
19:      **else**                                                    ▷ Case 2 (cf. Section 3.2)
20:          Find a vector $u_t$ such that $u_t^\top \mathbf{B}_t^\perp u_t \geq (1 - 1/p)\|\mathbf{B}_t^\perp\|_{\mathrm{op}}$      ▷ Power iteration
21:      $V_{t+1} \leftarrow V_t \cup \{u_t\}$.
22: Let $\mu_{\mathcal{V}_t} = \mathbf{E}_{X \sim P_t}[\mathrm{Proj}_{\mathcal{V}_t}(X)]$ be the mean of $P_t$ after projection to $\mathcal{V}_t$.
23: **return** $\mu_{\mathcal{V}_t}$ and $V_t$.

---

