# OpenReview forum: "Near-Optimal Algorithms for Gaussians with Huber Contamination: Mean Estimation and Linear Regression"
_NeurIPS.cc/2023/Conference — NeurIPS 2023 poster_

### Official Review · Reviewer_7D1F · 2023-07-06

**Soundness:** 3 good
**Presentation:** 3 good
**Contribution:** 2 fair
**Rating:** 6
**Confidence:** 3

**Summary:**

In this paper, the authors give an improved algorithm for estimating the mean of a multivarite Gaussian random variable under the Huber contamination model. Diakonikolas et al. Have given the first algorithm in this model that achieves the information theoretical optimal $\ell_2$ loss of $\Theta(\varepsilon)$ using a polynomial number of samples when the noise rate is $\varepsilon$. Pensia et al. have given a fast streaming algorithm for the same problem with an $\ell_2$ loss of $\Theta(\varepsilon\log \varepsilon^{-1})$ with a sample complexity of $d^2\varepsilon^{-2}$ where $d$ is the dimension. The authors carefully combine the ideas in the above two papers to achieve the sample complexity of $d\varepsilon^{-1}$ which is optimal.

At a high level, a mean vector $\mu’$ and a subspace $V$ is first found such that $\mu’$ and $\mu$ are close in $\ell_2$ norm in $V^{\bot}$. Moreover, $V$ has dimension at most $\mathrm{\poly}\log(d\varepsilon^{-1})$ such that a high sample-complexity inspired from DIakonikolas et al. is affordable. The finding of this subspace $V$ uses ideas from the Pensia et al. Paper by looking at a random direction where the eigenvalue is large as opposed to looking at the top k eigenvalues as in the Diakonikolas paper which incurse the worse loss.

The paper also gives a linear-sample algorithm for robust Gaussian linear regression by reducing it to the robust mean estimation problem of a certain conditional distribution of the multivariate Gaussian. This algorithm is slightly surprising to me as any continuous distribution has zero probability of exactly seeing a particular item in the sample space and therefore conditional sampling in infeasible. However, the authors manage to achieve their reduction by carefully considering a large enough ball that allows their reduction to go through.

I have not carefully checked the mathematical details of the paper.





**Strengths:**

The robust statistics in general and the multivariate robust mean estimation, in particular, have received wide attention recently. Therefore, the improved sample-optimal algorithm derived in this paper should be very interesting to the community.

**Weaknesses:**

None.

**Questions:**

A question to the authors: all the papers in this line of work achieves a recovery rate of say $\varepsilon/1000$ which looks quite lossy to me. Univarite robust estimation achives close to $\varepsilon$ rate, or smaller constants as do several other papers in statistics such as: [arxiv:2002.01432]. So, my question to the authors is: is there any hope that the constant can be improved, or this filtering technique has some inherent barrier?


**Limitations:**

None.

---

> ### Author Rebuttal · Authors · 2023-08-09
>
> We thank the reviewer for their positive comments and feedback.
>
> **(Constants in the breakdown point)** We would like to point out that constants could indeed be optimized, but that work falls outside of the primary aim of our paper. The filtering techniques that we use and their variants have been used to get better constants and breakdown points. In particular, [Dia+18] already takes special care to improve the constant in the error guarantee. Moreover, [arxiv:2002.01432] in its core uses similar filtering techniques to obtain better breakdown points. Finally, optimal break-down point can be achieved using filtering-based algorithms, see, for example, [ZJS22, , Exercise 2.10 of DK23, Appendix E of HLZ20]. As the primary focus on this paper is demonstrating that the nearly-linear runtime is possible, we did not optimize constants.

---

### Official Review · Reviewer_qzWP · 2023-07-06

**Soundness:** 2 fair
**Presentation:** 2 fair
**Contribution:** 2 fair
**Rating:** 6
**Confidence:** 3

**Summary:**

The authors study here a classical problem in statistics: the estimation of Gaussian means and linear regression with Gaussian covariates in the presence of Huber contamination.

This paper proposes a novel polylog time algorithm to compute the mean of a Gaussian in an $\epsilon$-corrupted, high-dimensional setting. This algorithm is obtained as a concatenation of two algorithms presented in the literature. They apply this algorithm in Gaussian linear regression and analyze this setting.

**Strengths:**

The paper seems technically sound and addresses some relevant questions in the field. The approach is based on an algorithm previously known in the literature [Dia+18], enhancing it with ideas from [DKPP22]

**Weaknesses:**

While the paper provides a solid theoretical analysis of the time and sample complexity for these algorithms to work, it could be more concerned with seeing it applied. A section could have been dedicated to applying the algorithm in some cases where the parameter values are relevant to some applications.

Additionally, the work is focused on proving that the algorithm performs as claimed, but it needs to address the limitation of the analysis of the algorithm. The conditions and assumptions are not clearly stated in the introduction of the problem.

The first part of the paper is straightforward and gives a clear introduction to understanding the questions addressed by the paper. It is pedagogical to explain the position in the current literature. The technical part of explaining the algorithm (specifically Section 3.2) takes more work to follow.

**Questions:**

I believe actual experimentation is lacking: Is it possible to test the algorithm and see its performance for some distribution choices? While it is clear that the distribution is Gaussian, are there some constraints on the other distribution?


**Limitations:**

The paper is theoretical is nature, so there are no limit

---

> ### Author Rebuttal · Authors · 2023-08-09
>
> We thank the reviewer for their time and feedback. We respond to the individual points below:
>
> * **(Simulations)** We refer to the response given to reviewer rxyU regarding the same matter.
> * **(``While it is clear that the distribution is Gaussian, are there some constraints on the other distribution?’’)** We are not entirely sure which is the other distribution that the reviewer is referring to. If the reviewer asks if there are any constraints on the distribution of the outliers, then the answer is no: The outliers could be sampled from any (arbitrary) distribution. If the the reviewer is asking whether the inlier distribution should necessarily be Gaussian, then we provide the response below:
>     * **(Distributional choices of inliers)** Regarding testing the algorithm for different distribution choices of inliers, we would like to point out that the Gaussianity assumption is somewhat critical for achieving error $O(\epsilon)$. Even for univariate sub-Gaussian distributions, the adversary can corrupt the distribution (in the Huber contamination model) such that every algorithm incurs error $\Omega(\epsilon \sqrt{\log(1/\epsilon)})$; see [DK19, page 4].
>     * **(Possible extensions)**  As outlined above, it is impossible to relax the Gaussianity assumption to general sub-gaussian distributions for inliers. Nonetheless, we can extend our theoretical results to inliers following subgaussian distributions which are centrally symmetric (to ensure correctness of median) and satisfy Hanson-Wright inequality (to ensure that multi-directional filter works).
>
> * **(``The conditions and assumptions are not clearly stated in the introduction of the problem’’)** The introduction provides an informal overview of the problem, and Section 1.1. provides complete formal statements for the two results with all conditions and assumptions stated. We are not sure which are the missing conditions that the reviewer is referring to.
>
> * **(``Position in the current literature’’)** This is mentioned in the related work section; cf. Lines 144-151. Given the extra space in the camera ready version, we could mention it earlier in the paper as well. In summary, the paper is positioned as follows:
>     * For subgaussian distributions, there are existing nearly-linear runtime robust mean estimation algorithms getting the information-theoretic optimal error of $O(\epsilon \sqrt{\log(1/\epsilon)})$ in the strong contamination model.
>     * For Gaussian distributions, these algorithms obviously continue to work, but the information-theoretic optimal error is now $O(\epsilon)$ and existing Statistical Query lower bounds show that achieving that error is computationally hard under the strong (or even TV-distance) contamination model. Thus, one needs to restrict to the Huber contamination model.
>     *  For Gaussians under Huber contamination, [Dia+18] gave the first polynomial time-and-sample algorithm achieving the optimal $O(\epsilon)$ error. In the present paper, we improve the runtime and sample complexity to be nearly linear. Additionally, we also obtain similar guarantees for robust linear-regression, where prior to our results, not even polynomial algorithms were known for $O(\sigma \epsilon)$ error.

---

### Official Review · Reviewer_BEmV · 2023-07-06

**Soundness:** 4 excellent
**Presentation:** 4 excellent
**Contribution:** 4 excellent
**Rating:** 8
**Confidence:** 3

**Summary:**

Gaussian estimation and linear regression are fundamental statistical tasks, and the Huber contamination model is one of the most well-studied models for robust statistics. This paper works to design algorithms for these tasks with near-optimal sample complexity, error, and almost linear running time. The main result is for Gaussian robust mean estimation with near-optimal sample complexity of $\tilde{O}(d/\varepsilon^2)$, almost linear running time, and near-optimal $\ell_2$ error of $O(\varepsilon)$. (Previously there have only been algorithms with (i) near-optimal error but polynomially-suboptimal sample complexity and running time, or (ii) near-linear runtime but suboptimal error.) The paper obtains similar guarantees for robust Gaussian linear regression with Huber contamination: resolving the open problem of whether there exists an algorithm with $O(\sigma \varepsilon)$ error in polynomial time and sample complexity. Applicability to streaming is also discussed.

**Strengths:**

The strengths will read similarly to the summary, in that the obtained results are for fundamental problems that mostly speak for themselves. The tasks of Gaussian mean estimation and linear regression with Huber contamination are core to algorithmic robust statistics, and it is highly desirable to have algorithms with near-optimal error, sample complexity, and near-linear running time simultaneously.

The main result of this paper is for Gaussian mean estimation with Huber contamination. Grossly oversimplifying, the result works by leveraging ideas related to [DKPP22] and [Dia+18]. [DKPP22] obtains near-linear time but suboptimal error guarantees. It obtains a fast running time by techniques with which it filters in random directions of large variance. [Dia+18] obtains near-optimal error with slower running time by employing a stronger filtering technique that deterministically filters with respect to the subspace for the k-largest eigenvalues. These techniques are not obvious how to use together, as trying to sample for random, strong filtering directions may just consistently produce the same directions, and deterministic filtering will naturally incur a slowdown. In summary, this work’s primary technical contribution is showing how to obtain the guarantees of strong filtering techniques without incurring the slower runtime that is naively associated with them.

**Weaknesses:**

One could argue that the very high-level approach for the main result of mean estimation is not too surprising in how it blends the intuitions of [Dia+18] and [DKPP22]. Still, there are significant technical obstacles to this approach and it is perhaps even nicer to see how techniques from this general body of work are flexible enough to coalesce into a new, important result.

Providing some discussion on the relationship to settings with non-identity covariance might be beneficial.

**Questions:**

Could you provide discussion on implications for more general covariances?

"reminder" -> "remainder" line 233

**Limitations:**

The limitations are appropriately discussed.

---

> ### Author Rebuttal · Authors · 2023-08-09
>
> We thank the reviewer for the positive evaluation. We respond to their question about general covariance matrices below.
> First, we can restrict to the case that the covariance matrix is unknown to the algorithm; otherwise, using an appropriate whitening of the data can reduce to the identity covariance case.  For this case, we note the following:
>
> **(Information-theoretic Error)** For unknown covariance Gaussians, the information theoretic optimal for the Euclidean error is $\Theta(\epsilon \sqrt{\lVert \Sigma \rVert_{op}} )$. Moreover, this can be achieved with linear in $d$ sample complexity by (computationally-inefficient) algorithms. See, e.g., [DK23, Lemma 1.9 and Proposition 1.20].
>
>
> **(Computationally-Efficient Error)** However, any computationally-efficient (statistical query) algorithm that achieves error $o( \sqrt{\epsilon} \sqrt{\lVert \Sigma \rVert_{op}} )$ requires $\Omega(d^2)$ samples [Dia+22a, Theorem 6.11 with $k = \sqrt{d}$]. With $d^2$ samples, one can robustly learn the covariance (without the extraneous $\log(1/\epsilon)$ factor) using [Dia+18] and reduce it to the known covariance setting. Crucially, the algorithm in [Dia+18] runs in quasipolynomial time and speeding up this algorithm is an important problem but beyond the scope of our work.

---

> > ### Comment · Reviewer_BEmV · 2023-08-13
> >
> > Thank you for your response, I have no more questions.

---

### Official Review · Reviewer_bgYs · 2023-07-07

**Soundness:** 3 good
**Presentation:** 3 good
**Contribution:** 3 good
**Rating:** 6
**Confidence:** 3

**Summary:**

This paper presents an algorithm to estimate the mean of a Huber contaminated d-dimensional Gaussian with sample complexity of n=O(d/\epsilon^2)  and a run time that is near-linear. They also provide an algorithm for robust linear regression with sample complexity of n=O(d/\epsilon^2) and a run time that is again near-linear.  The results improve upon the existing poly(d/\epsilon) algorithms that achieve \Theta(\epsilon) error.

**Strengths:**

The paper is well-written and the flow is good. Although the paper is highly theoretical, the writing is friendly.

The results of the paper seem to answer an important open question and the paper claims to achieve a sample complexity that is close the the mean estimation problem in the uncontaminated case, which is good.

**Weaknesses:**

The paper seems to combine the techniques present in two existing papers to obtain their result. Although the authors clarify that such a combination is highly non-trivial, the math used in the proof section is very dense for a thorough verification of this claim.



**Questions:**

In Appendix B (which forms the core of the proof for theorem 1.3), what is the core insight that helps obtain the final sample complexity? The paper claims that combining the techniques from [DKPP22] and [Dia+18] is highly non-trivial. However, The core proofs in sections B.4.1 do not contain any novel proof techniques apart from the ones already present in [DK19] and [Dia+16]. If you could offer more insight into this in the main paper, that would be good.

**Limitations:**

This is a theoretical paper with no negative societal impact.

---

> ### Author Rebuttal · Authors · 2023-08-09
>
> We appreciate the positive feedback on the contributions and the writing.
>
> However, it appears that the reviewer touched upon and subtly intertwined two distinct aspects of the algorithm:
> (i) combining the algorithmic techniques used in [DKPP22, Dia+18] that is developed in Section 3, and
> (ii) the sample complexity for ensuring the deterministic conditions (discussed in Appendix B.4.1 and its relation to [DK19, Dia+16]).
>  Specifically, we would like to highlight that (i) is the main contribution of our work and forms the core of Theorem 1.3. Thus, to minimize any potential confusion, we would like to provide clarifications on both of these points below.
>
> **(Algorithmic techniques from [DKPP22, Dia+18])** Our main contribution is achieving the optimal error in nearly-linear runtime, which requires a non-trivial combination of the algorithm components from [DKPP22, Dia+18]. Combining the techniques from [DKPP22, Dia+18] is the main focus of our paper which we argue that is non-trivial, both from an algorithmic and an analytical viewpoint. We highlight the roadblocks in combining these techniques in Lines 116-121. Our solution is then outlined in Lines 122-134, with the details presented in Section 3. We stress again that this solution is the main contribution of our paper (and orthogonal to [Dia+16,DK19]).
>
> **(Sample complexity for deterministic conditions)** The proposed algorithm above succeeds under a set of (novel) deterministic conditions (given in Definition 2.1). The last of these conditions (Condition 2.c), which is the key to obtaining the optimal error, is novel, and thus requires a new proof, provided in Appendix B.4.1 (only the conditions 2.a and 2.b come from [DK19, Dia+16]). In addition, we show that Condition 2.c holds with linear sample complexity, which should be contrasted with the (large) polynomial sample complexity in [Dia+18] for the corresponding condition there. That said, we stress again that this is not the main result/focus of our paper (and is thus deferred to Appendix). The reviewer is also asking about the core insight for linear sample complexity of Definition 2.1, which we summarize next:
>
> **(Core insight for linear sample complexity of Definition 2.1)** We focus our attention to Condition 2.c, which is the novel deterministic condition, and provide a very high level intuition for why its sample complexity should be (nearly) linear. Observe that this condition asks for a uniform concentration along all matrices $U$ of dimension $d \times k$, which are objects of $dk$ parameters. Since $k$ is logarithmic in $d$ and $1/\epsilon$, the total number of parameters is nearly linear in $d$. Informally, as the number of parameters of the objects that we ask uniform concentration for is roughly related to the sample complexity, this eventually results in a nearly linear sample complexity. (More formally, we construct a finite cover for these matrices whose size is exponential with the exponent being nearly linear. Since Gaussians have exponential concentration, the resulting sample complexity is logarithmic in the cover size, and thus nearly linear.)

---

> > ### Comment · Reviewer_bgYs · 2023-08-18
> >
> > Thank you for the clarification, I do not have any further comments.

---

### Official Review · Reviewer_rxyU · 2023-07-07

**Soundness:** 4 excellent
**Presentation:** 2 fair
**Contribution:** 3 good
**Rating:** 5
**Confidence:** 3

**Summary:**

This paper solves an important problem in robust mean estimation and robust linear regression, that is, can we achieve optimal error rate with nearly linear sample size and runtime when the inliers are standard Gaussian. All the previous works that have linear runtime suffer from an additional $\sqrt{\log(1/\epsilon)}$ factor. To derive such results, they develop a new analysis for the classic filtering technique. Moreover, they develop a novel reduction for the Gaussian linear regression to mean estimation problem, which is also of independent interest.

**Strengths:**

- This study solves a long-standing problem in the robust mean estimation and robust linear regression, that is, can we close the gap between the information-theoretic lower bound and the guarantees for algorithms that enjoy linear runtime. This problem, though is nearly the simplest setting, is very fundamental in this field, and I feel like the analysis developed in this paper can be adapted to other tasks.

- The proof technique is very strong. The reduction for Gaussian linear regression is quite novel.

- I like the intuition illustrated in Section 1.2.

**Weaknesses:**

- In most of the literature listed in this paper, people consider the stronger outlier model, i.e., the strong contamination model. My question is, why the authors can only handle Huber's model? What is the main difficulty in extending to the strong contamination model?

- Some notations are not consistent: the pdf function is denoted by both $P(x)$ (line 154) and $p(x)$ (line 173).

- It would be great if there are some preliminary simulations to support the theory, like plotting the runtime v.s. dimension.

**Questions:**

N/A

---

> ### Author Rebuttal · Authors · 2023-08-09
>
> We thank the reviewer for their effort and feedback.
>
> Regarding the summary of the paper, we would like to highlight that the algorithm for linear regression is in fact the *first polynomial algorithm* obtaining error $O(\sigma \epsilon)$, i.e., not only attains nearly-linear runtime and nearly-optimal sample complexity, but prior to this, no polynomial-time and sample complexity algorithm was known to achieve the $O(\sigma \epsilon)$ error (cf. lines 82-83).
>
> We now proceed by addressing the reviewer’s questions below:
>
> **(Huber’s vs strong contamination)** There is an inherent difference in the guarantees that can be obtained between the two contamination models. While the information theoretic optimal error is $\Theta(\epsilon)$ for both contamination models (and achievable by exponential time algorithms with linear sample complexity), there is a statistical query lower bound providing evidence that it is computationally hard to reach error $o(\epsilon\sqrt{\log(1/\epsilon)})$ in the strong contamination model (or even total variation model); see lines 146-148 in the introduction. This computational lower bound necessitates that we consider the simpler Huber contamination.
>
> **(Notation)** Throughout the paper, we have consistently used capital letters (e.g., $P(x)$) to denote probability density functions (pdfs). The instance $p(x)$ mentioned in line 173 does not represent a pdf; instead, it denotes the polynomial defined in the previous line.
>
> **(Simulations)** While we acknowledge the reviewer's interest in experiments and the importance of developing practical algorithms, the primary contribution of our work is to complete the computational and statistical landscape for the problem in terms of error guarantee, sample complexity and runtime. As NeurIPS invites theoretical ML papers (such as ours),  we believe that our work should not be negatively evaluated for the lack of experiments, especially when no other nearly-linear time estimator with optimal guarantees was previously known.
>
>
> In the light of these responses, we kindly request the reviewer to reconsider their score.

---

> > ### Comment · Reviewer_rxyU · 2023-08-19
> > **Thanks for your reply**
> >
> > Thanks for your reply, which addresses my concern.

---

### Author Rebuttal · Authors · 2023-08-09

We thank the reviewers for their time and effort in providing feedback. We are encouraged by the positive comments, and that all the reviewers appreciated the paper for the following: (i) **novelty** (rxyU, BEmV, qzWP), **importance** (rxyU,bgYs,BEmV, 7D1F) (iii) **good** and **friendly** writing (bgYs), and (iv) **surprising results** (7D1F). We address the individual questions and comments by the reviewers separately.

---

### Decision · Program_Chairs · 2023-09-21

**Decision:**

Accept (poster)

**Comment:**

The paper tackles the fundamental problem of Gaussian mean estimation under Huber contamination. The authors propose the first method that linear time method that is sample optimal.

The reviewers were positive about this paper. While there is some variance between the reviewer scores, some of the reasoning for the lower scores (such as lack of experiments for this theory paper) was not defended nor elaborated in the discussion period.

Given the importance of the contributions and the overall scores, I recommend acceptance.